# A dynamical process-based model for quantifying global agricultural ammonia emissions – AMmonia–CLIMate v1.0 (AMCLIM v1.0) – Part 2: livestock farming

Jize Jiang[1,a,b], David S. Stevenson[1], Aimable Uwizeye[2], Giuseppe Tempio[2], Alessandra Falcucci[2], Flavia Casu[2], and Mark A. Sutton[3]

[1]School of GeoSciences, The University of Edinburgh, Crew Building. Alexander Crum Brown Road, Edinburgh, EH9 3FF, UK

[2]Food and Agriculture Organization of the United Nations, Animal Production and Health Division, Viale delle Terme di Caracalla, 00153 Rome, Italy

[3]UK Centre for Ecology and Hydrology, Edinburgh, Bush Estate, Midlothian, Penicuik, EH26 0QB, UK

[a]now at: Institute of Agricultural Sciences/Institute of Biogeochemistry and Pollutant Dynamics, ETH Zurich, 8092 Zurich, Switzerland

[b]now at: Eawag, Swiss Federal Institute of Aquatic Science and Technology, Ueberlandstrasse 133, 8600 Dübendorf, Switzerland

*Correspondence to*: Jize Jiang (jize.jiang@usys.ethz.ch)

**Abstract.** Agricultural ammonia ($NH_3$) emissions are a major pathway of nitrogen loss, which can have significant environmental consequences, such as air and water pollution, ecosystem damage and biodiversity loss. Ammonia emissions related to livestock farming are major sources in the agricultural sector, resulting from animal housing, manure management and land application. This paper is the second part of the description of the AMmonia–CLIMate (AMCLIM) model, presenting the development and application of all three main modules to estimate $NH_3$ emissions from livestock, including pigs, poultry (chicken), cattle, sheep and goats. The AMCLIM model simulates the flows of N species at different stages comprised in livestock agriculture. It incorporates the effects of environmental factors and also provides an adequate level of detail for the representations of human management practices. According to simulations by AMCLIM, it is estimated that $NH_3$ emissions from global livestock farming are about 29.9 Tg N $yr^{-1}$, accounting for around 30 % of total excreted nitrogen. Cattle and buffaloes systems are estimated to be the largest sources of $NH_3$ emissions, contributing over 60 % of total livestock emissions. Both pigs and poultry systems result in more than 15 % of estimated total emissions, while sheep and goats are responsible for the remaining 7 %. High volatilization rates frequently occur in hot regions, indicating the climate-dependence of $NH_3$ volatilization. It is also shown how AMCLIM can simulate the influence of management practices on $NH_3$ volatilization, e.g., illustrating how fully-enclosed animal houses with heating and forced ventilation can result in higher emissions than naturally ventilated barns, while poorly managed manure leads to substantially increased $NH_3$ emissions.

## 1 Introduction

Ammonia ($NH_3$) is the primary form of reduced reactive nitrogen ($N_r$) and it mainly originates from agricultural activities. Excessive $NH_3$ emissions affect air, soil, and water quality, local ecosystems and biodiversity, and can pose serious threats to human society. At the same time, $NH_3$ volatilization is one of the key pathways of N leak from the agricultural systems to the environment, representing a critical nutrient loss and causing unnecessary economic cost.

Livestock farming is an important component of agricultural systems. As the global population grows, livestock numbers have increased dramatically to fulfil the rising demand for animal products such as milk, meat and eggs. Specifically, pigs and poultry are the sectors that recorded the largest increase in livestock population numbers, with pigs having increased by about 140 % and poultry having increased by nearly five-fold over the past 50 years (FAO, 2022). This surge in livestock population has also resulted in a substantial increase of nutrient requirements, particularly in N inputs in animal feed. However, N recycling within livestock farming systems is often poor, resulting in a significant amount of N loss instead of being used by the animals. In particular, $NH_3$ emissions are a major pathway of N loss to the environment and can cause serious environmental problems (Sutton et al., 2011). Therefore, accurate estimation of $NH_3$ emissions is crucial for assessing the environmental impact of livestock farming systems and optimizing resource utilization.

Cattle systems contribute to the largest $NH_3$ emissions among livestock (Uwizeye et al., 2020). Existing studies have reported that over 50 to 60 % of animal–related $NH_3$ originates from cattle agriculture (including buffaloes), while sheep and goat farming together resulted in around 10 % of livestock $NH_3$ emissions (Behera et al., 2013; Bouwman et al., 1997; Dentener and Crutzen, 1994). According to FAO statistical data, ruminant population have increased more than 60 % over the past 50 years (FAO, 2022). Compared with pigs and poultry, $NH_3$ can also originate from excreted nitrogen during ruminant grazing, which is still poorly quantified globally and needs to be investigated.

The most commonly used method for estimating $NH_3$ emissions is using emission factors (EFs), combining with statistical activity data for different source sectors. However, as $NH_3$ emissions are highly sensitive to environmental conditions, such as temperature and water availability, EFs usually only consider the climatic effects to a limited extent, so they may not accurately represent $NH_3$ volatilization. To address this deficiency of EFs, process-based models are developed based on the theoretical understanding of relevant processes (Flechard et al., 2013; Móring et al., 2016; Nemitz et al., 2001; Sutton et al., 1995). A challenge for process-based models is the representation of various management practices existing in livestock agriculture, which can also influence the $NH_3$ volatilization in different ways. Such complications are difficult to parameterize in models and keep the consistency of the model structuring, especially for large-scale simulations. Other barriers for the modelling include the high requirement of input data and the shortage of sufficient quality observations for validation and evaluation.

A process-based, dynamical emission model, AMmonia–CLIMate (AMCLIM) has been specifically designed that incorporates the effect of both environmental conditions and management practice to simulate agricultural $NH_3$ emissions. Compared with existing process-based models, AMCLIM is thought to be the first model that simulates $NH_3$ emission from both synthetic fertilizer use and livestock farming using a consistent process-based modelling approach, with high levels of

detail of the representation of agricultural practices. There are other process-based models, such as the 'Flow of Agricultural Nitrogen model version 2' (FANv2; Vira et al. 2020) that simulates agricultural $NH_3$ emissions interactively within the Community Earth System Model (CESM) with detailed soil processes for land application of fertilizers and ruminant grazing. Another is 'Calculation of AMmonia Emissions in ORCHIDEE' model (CAMEO), which includes several management modules for livestock feed, manure management and agricultural handling practices within a global land surface model (Beaudor et al., 2023). While these models still largely rely on EFs for estimating $NH_3$ emissions from livestock sectors, AMCLIM explicitly models the N flows within the systems and include several major N processes. AMCLIM uses an integrated approach to simulate how various N species are influenced by environmental factors in a sequence of the practices in the livestock sector, from livestock housing to manure management and ultimate application of manure to fields, as well as ruminant grazing. By following this sequence in AMCLIM, changes in emissions at early stage of livestock agriculture influence the simulated N pools, and can thereby affect emission at a later stage of these activities.

The structure and simulations of AMCLIM for global synthetic fertilizer use have been presented in the companion paper (Jiang et al, 2024). In the present paper, the development of the modules, evaluation, and application of the AMCLIM model for simulating $NH_3$ emissions from livestock farming are described. An earlier version of this conceptual approach has already been reported by Jiang et al. (2021) with a focus on chicken farming. The present paper describes the development of the approach for other livestock, as well as describes updates in relation to the treatment for poultry. The AMCLIM model has been tested against measurements at a site scale and then applied to global scale.

## 2 Method and materials

### 2.1 AMCLIM model structure

The design of the AMCLIM model is closely associated with human activities in agriculture systems. The model structure and components are shown in Fig.1 (same as the Fig.1 in the companion paper, Jiang et al., 2024). There are three modules in AMCLIM: a) Housing, b) Manure Management and c) Land. The development and application of the Land module (AMCLIM–Land) for simulating synthetic fertilizer use has been described in detail in Jiang et al. (2024). Therefore, the present paper mainly focuses on the livestock sector, including pigs, poultry (chicken) and major ruminants (cattle, sheep and goats).

Livestock consume N from feed crops, agrifood industry by-products and concentrated feeds for gaining weight, producing meat, milk and eggs. Most N ingested through feeding is excreted through urine and dung, and can be a valuable source of organic fertilizers for grassland and cropland application. The animal excreta collected from animal houses is stored as slurry or solid manure and then applied to arable land during growing seasons. However, the management of livestock manure can vary greatly across regions. For example, some farmers spread manure daily or simply leave it in the yard or holding area without much management or storage. Each of these management practices can result in $NH_3$ emissions.

All three modules in AMCLIM are operated for capturing the activities and practices of livestock farming. The connections between modules reflect typical N flows in the livestock production systems, from animal housing to manure storage/management and then to the ultimate land application, as shown in Fig. 1. As $NH_3$ emissions can be released at all stages, all three modules need to provide robust estimates as previous components can have substantial influences on the following ones, i.e., less emission from housing leaves higher N content in the animal excreta, which can cause larger emissions

in the succeeding practices. The following sections describe the different modules that are used to address these components, as well as explain how the modules are linked. The focus here is on describing the Housing (AMCLIM–Housing) and Manure Management (AMCLIM–MMS) Modules of AMCLIM. This paper highlights the different processes of manure application compared to synthetic fertilizer application and differentiates the processes specific to grazing livestock.

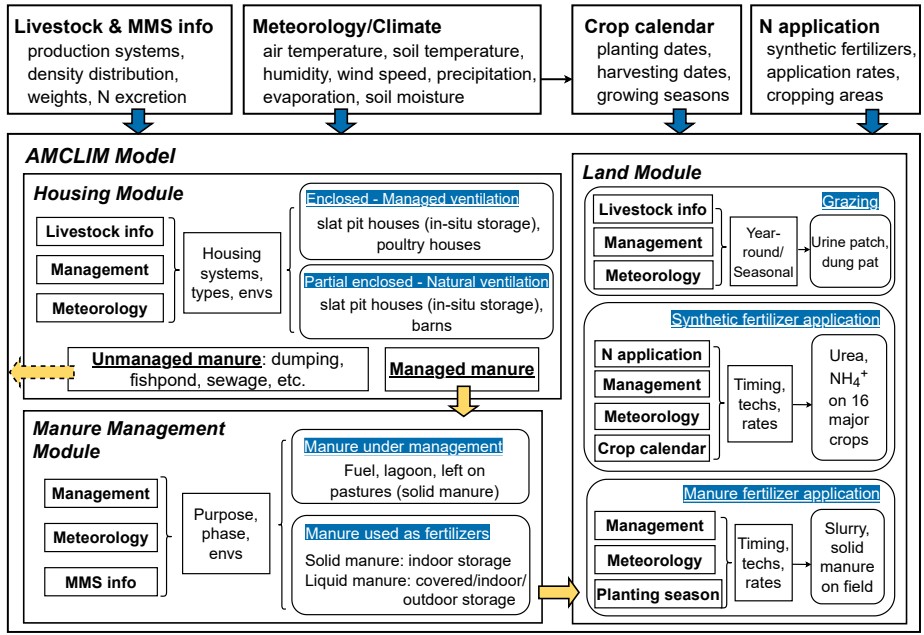

**Figure 1. Components and structure of the AMCLIM model and inputs (blue arrows) used for simulations. The dashed yellow arrows represent a fraction of unmanaged N from housing that is not simulated in the manure management module (AMCLIM–MMS). Solid yellow arrows represent the N flows between modules (MMS: manure management system; Envs: environments; Techs: techniques).**

## 2.2 Housing Module of AMCLIM: AMCLIM–Housing

### 2.2.1 Housing systems and house types

The Housing Module in AMCLIM (AMCLIM–Housing) was designed to estimate the $NH_3$ emissions from livestock housing using principles relevant for different livestock types. Pigs and poultry are mostly kept in buildings, while ruminants like cattle

and sheep may also spend a considerable amount of time in barns or stalls depending on weather and local management. In each case, $NH_3$ emissions are the result of decomposition of excreted N, with negligible amounts by comparison assumed to be emitted through animal breath and sweat.

AMCLIM–Housing includes two housing systems and three house types, depending on the production system of the livestock and management. Two housing systems are distinguished: enclosed housing and partially enclosed housing, which are reflected by different indoor environmental conditions. Enclosed housing is assumed to have forced heating and managed ventilation, which is commonly used for commercial pigs and poultry in order to improve livestock performance (Gyldenkærne, 2005; FAO, 2018). Partially enclosed housing refers to barns or houses that are naturally ventilated, the indoor environment of which is assumed to be close to the natural environment. These two systems are employed to differing degrees by different livestock sectors and production systems. For example, cattle have a higher tolerance to cold weather than pigs and poultry, so are typically kept in naturally ventilated barns (Seedorf et al., 1998).

Three house types in AMCLIM–Housing include: 1) houses with slatted floors and storage pits, 2) normal barns (without slatted floors and pits) and 3) deep litter poultry houses. The first house type with **slatted floors** allows animal excreta to be removed quickly and effectively, so the house can be easily cleaned. The slatted floor is usually concrete or iron, and there are partially slatted compartments. The gap area of the slatted floor usually accounts for approximately 20 % and no more than 50 % of the total floor area (Aarnink et al., 1997). The excreta fall to the pit underneath through the gaps and are stored in situ for a period. Emission of $NH_3$ can be from both the slatted floor area and the storage pit. Such slatted pit houses are prevalent in pig farming, especially for industrial production systems. A two-reservoir emission scheme is used for this type of houses in AMCLIM–Housing, with the pit storage simulated by a two-film model (Liss, 1973; Liss and Slater, 1974). The two-reservoir emission scheme details are given in Sect.2.2.3, and the two-film model is described in Sect.S3.1 in the Supplementary Materials.

The second house type is **barns**. Barns are commonly used facilities in livestock housing because they can be easily set up and require less capital input compared to animal houses with slatted floors and pit storage. Barns are normally naturally ventilated and are not fully enclosed. During cold days, mechanical blocking may be applied to open barns to reduce ventilation (Gyldenkærne, 2005). Excreta and bedding are frequently removed to a separate storage unit to keep the barn clean. In most cases, daily cleaning of barns is necessary.

The third house type is **deep litter poultry houses**. Except for some regions, poultry houses for broiler and layer production systems are mainly enclosed with forced heating and ventilation. Commonly, poultry excreta accumulate and remain in the houses for a long time, e.g., months to years, until it is removed. Bedding materials such as straw are added to absorb moisture as well as to reduce emissions, which is a typical management practice for breeders and broiler system (FAO, 2018).

Across the world, there are many other variants of animal housing systems. However, the major systems that are listed can be considered sufficient for the focus of exploring the sensitivity of climate on $NH_3$ emissions, while providing a modelling approach that addresses the major management opportunities to reduce emissions.

## 2.2.2 Simulated processes in animal houses

Animal housing is one of the primary sources and often the very first place of $NH_3$ emissions in livestock farming systems. Figure 2 depicts the processes through which $NH_3$ emissions originate from excreta in animal houses, ultimately released into the outdoor atmosphere. In general, there are six processes that can be summarized as follows:

- **Process 1: Excretion.** Livestock excreta contain N in the form of urea in pigs' and ruminants' urine (and uric acid in poultry excretion), as well as other organic forms of N in pigs' and ruminants' dung and poultry faeces.

- **Process 2: Conversion of excreted N to ammoniacal N.** Excreted N on the floor surface of the animal house is converted to total ammoniacal nitrogen (TAN) through hydrolysis of urea or uric acid and decomposition of organic N (details are given in Sect. S2 in the Supplement).

- **Process 3: Equilibration of TAN.** The TAN pool partitions into multiple phases; gaseous $NH_3$ is in equilibrium with aqueous TAN.

- **Process 4: Emission from surfaces.** Ammonia volatilizes to the house atmosphere from manure and other surfaces in the building.

- **Process 5: Accumulation of gaseous ammonia.** The indoor $NH_3$ level builds up due to $NH_3$ volatilization according to the limited extent of ventilation.

- **Process 6: Emission to outside the building.** Indoor $NH_3$ is removed from the house to the outside atmosphere through ventilation.

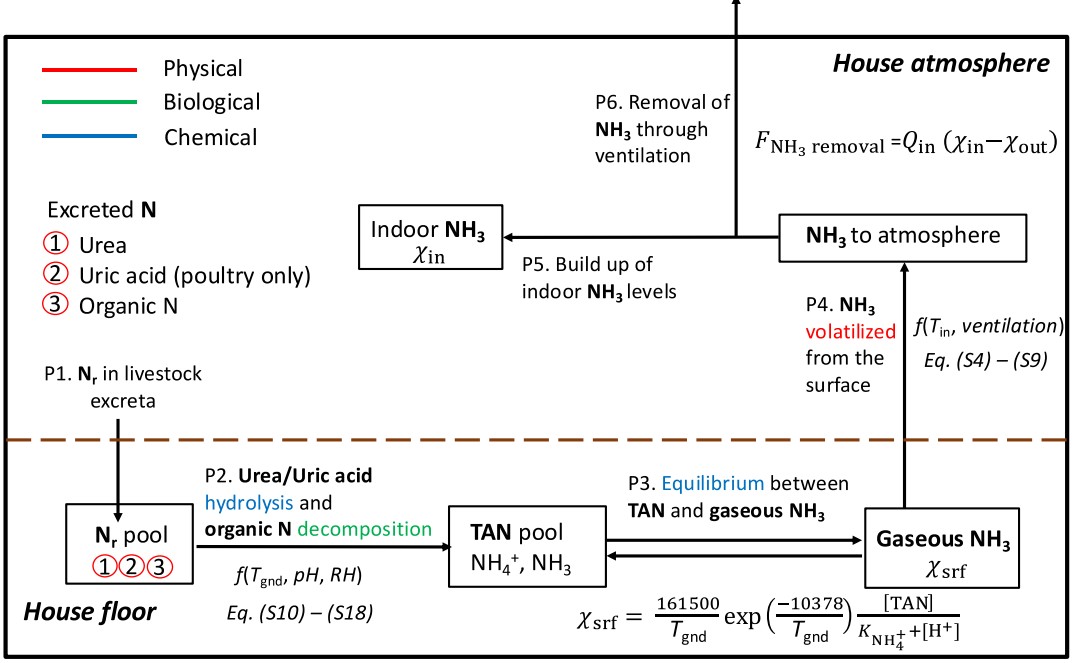

**Figure 2. Schematic of NH₃ volatilization in animal houses (adapted from Elliott and Collins, 1982 and Jiang et al., 2021). Physical, biological and chemical processes are highlighted in red, green and blue, respectively.**

The concentration of $NH_3$ inside the animal house ($\chi_{in}$, g m⁻³) is regulated by the balance between $NH_3$ volatilization from the floor surface ($F_{NH_3}$) and removal of $NH_3$ to the outside atmosphere ($F_{NH_3\text{ removal}}$), which can be expressed by the following equation:

$$\frac{d\chi_{in}}{dt} = F_{NH_3} - F_{NH_3\text{ removal}}, \tag{1}$$

where the fluxes are expressed as the sum of the flux for the whole animal house (g N s⁻¹). The time-dependent concentration of indoor $NH_3$ of the animal house can be represented by the following equation:

$$\begin{cases} V_{house} \frac{d\chi_{in}}{dt} = \frac{(\chi_{srf} - \chi_{in})}{R_{G,house}} \cdot S_{house} - Q_{in}(\chi_{in} - \chi_{out}) \\ F_{NH_3} = \frac{(\chi_{srf} - \chi_{in})}{R_{G,house}} \end{cases}, \tag{2}$$

where $\chi_{in}$ (g m⁻³) represents the indoor $NH_3$ concentration assuming a well-mixed state of air inside the animal house. $\chi_{srf}$ (g m⁻³) is the gaseous $NH_3$ concentration at the emitting surface, and $\chi_{out}$ (g m⁻³) is the free-atmosphere $NH_3$ concentration. $S_{house}$ (m²) and $V_{house}$ (m³) represent the surface area and the volume of the house, respectively. $Q_{in}$ (m³ s⁻¹) is the airflow rate of the house with a unit of cubic meter per second. The resistance for $NH_3$ volatilization in the animal house ($R_{G,house}$, s m⁻¹) is determined by the inverse of an empirically-derived gaseous transfer coefficient for $NH_3$ ($k_{G,housing}$, m s⁻¹), which depends on housing conditions such as temperature and ventilation, as expressed by the following equation:

$$R_{\text{G,house}} = \frac{1}{k_{\text{G,housing}}} \tag{3}$$

Animal houses are cleaned after certain amount of time. The frequency of cleaning varies depending on the housing management. The TAN pool ($M_{\text{TAN}}$; given in per unit area; all masses have units of g m$^{-2}$ if not specifically explained) in the
185 animal house can be determined by the following equation:

$$\frac{dM_{\text{TAN}}}{dt} = F_{\text{TAN}} - F_{\text{NH}_3} - \psi_{\text{cleaning}}(\text{t, TAN}), \tag{4}$$

where $F_{\text{TAN}}$ is the TAN production, i.e., through urea or uric acid hydrolysis and decomposition of organic N for livestock excreta (together with other processes are presented in Sect.S2 in the Supplement). $F_{\text{NH}_3}$ is the flux of NH$_3$ volatilization (all following N fluxes/flows have units of g N m$^{-2}$ s$^{-1}$ if not explained). $\psi_{\text{cleaning}}(\text{t})$ represents the cleaning event of
190 the house, as expressed as follows:

$$\psi_{\text{cleaning}}(\text{t, excreta/N/H}_2\text{O}) = \begin{cases} 0, & \text{if "Not a cleaning day"}, \\ \frac{M_{\text{excreta/N/H}_2\text{O}}}{t_{\text{cleaning}}}, & \text{if "a cleaning day"} \end{cases} \tag{5}$$

The cleaning event refers to the removal of livestock excreta ($M_{\text{excreta}}$), all N species ($M_{\text{N}}$) and water ($M_{\text{H}_2\text{O}}$) from the animal house within an assumed time scale of 24 h ($t_{\text{cleaning}}$).

The removed excreta can either be stored or applied to land as fertilizer, which will be described in the following sections.
The pools for other N species, e.g., urea, in the animal houses can be expressed as follows:

$$\frac{dM_{\text{N}_i}}{dt} = F_{\text{excretN}} f_{\text{N}_i} - K_{\text{N}_i} M_{\text{N}_i} - \psi_{\text{cleaning}}(\text{t, N}_i), \tag{6}$$

where $F_{\text{excretN}}$ is the total N excretion rate from the livestock, and $f_{\text{N}}$ is the fraction of a N form in the excretion. $K_{\text{N}}$ is the conversion rate (s$^{-1}$) at which a N species ($M_{\text{N}}$) decomposes. For pigs and ruminants, nitrogen is excreted in AMCLIM as urinary N and faecal N, with urinary N being in the form of urea and organic forms (Jørgensen et al., 2013; Vu et al., 2009a,
b). For poultry, AMCLIM assumes 60 % of the excreted N is in the form of uric acid, and the remaining 40 % is in organic forms (Nahm, 2003). The excretion pool is determined using the following equation:

$$\frac{dM_{\text{excreta}}}{dt} = F_{\text{excreta}} - \psi_{\text{cleaning}}(\text{t, excreta}), \tag{7}$$

where $F_{\text{excreta}}$ is the excretion rate from the livestock, which is derived from the N excretion rates based on the N content in the excreta. The pH of the livestock excretion is used for determining the decomposition rates of N species and chemical equilibria
in housing simulations. As discussed in the companion paper (Jiang et al., 2024), substrate pH is a critical factor that impacts the NH$_3$ emission. The dynamic equilibrium between gaseous NH$_3$ and aqueous ammonium is dependent on pH. On the other hand, pH affects the rates of uric acid hydrolysis and nitrification, which together control the TAN pool. In AMCLIM, the pH of the livestock excretion is used for determining the decomposition rates of N species and chemical equilibria in housing simulations.

It is worth noting that there are different characteristics for the simulations of animal housing between the studied livestock sectors, which are presented in Sect.S3 in the Supplementary Material.

### 2.2.3 Two-reservoir emission scheme for simulating houses with slats and pits

Houses with slatted floor and pit storage allow animal excreta to be stored in-situ, keeping the floor area clean. For this housing type, a two-source emission scheme is used to model $NH_3$ emissions, as there are two emitting surfaces: the slats and the pit. The two $NH_3$ emission elements are treated as additive, i.e., the total housing emission is the sum of the emissions from the two housing compartments. The pools of N species and other simulated variables are divided into two separate reservoirs to represent the processes on the slats and in the pit. Livestock excreta are split proportionally between the two reservoirs depending on the gap space of the slats. For example, if the gap space is 20 % and the slat space is 80 %, then 20 % of initial pig excreta will fall into the underneath pit, and the remaining 80 % will stay on the slats. Given the fact that excreta left on the slats will eventually fall to the pit (i.e., through cleaning), but excreta in the pit cannot go back to the slatted floor above, a unidirectional transfer is applied daily in AMCLIM–Housing. It is assumed that all pools from the slat reservoir go into pit reservoir by the end of each day, and the slat reservoir is reset to zero subsequently. Excreta that go into the pit are stored for longer time, e.g., weeks to months.

The process of $NH_3$ volatilization differs between the two reservoirs because of the different amount of water held in the two reservoirs. For the slats, excreta are typically a thin wet layer, so the surface concentrations can be expressed by the concentrations of the entire layer. The gaseous $NH_3$ concentration at the surface is directly derived from the aqueous TAN concentration of this layer. By contrast, the pit reservoir holds more water (and faeces) because urine in the excreta accumulate in the pit. There is an additional aqueous transfer process of TAN from the bulk water to the air-water interface. As described in Sect.S1 in the Supplementary Material, AMCLIM–Housing incorporates a two-film model that describes the gas exchange across the air-liquid interface (Liss, 1973; Liss and Slater, 1974).

## 2.3 Manure Management Module of AMCLIM: AMCLIM–MMS

### 2.3.1 Manure management systems

Properly dealing with animal excreta is crucial because poorly managed animal excreta can cause large unintentional N losses due to $NH_3$ emissions. Under adequate management, livestock excreta are a valuable N source as fertilizers. Manure is a mixture of animal excreta (including urine and faeces), bedding, feeds, drinking water and water used for cleaning from the housing. Collected manure is usually stored for a period until it is applied to fields at an appropriate time, while manure can also be used as fuel.

The Manure Management Module (AMCLIM–MMS) was developed to simulate the $NH_3$ emission from the stage after manure is removed from the housing systems and before it is spread on land. The Global Livestock Environmental Assessment Model (GLEAM https://www.fao.org/gleam/en/) considers over 20 manure management systems (MMS) (Uwizeye et al., 2020), with manure in either liquid or solid phase depending on the water content. The main divisions identified for AMCLIM and regrouped from the MMS defined in GLEAM are based on the similarities existing in the general practices, as follows:

A) Indoor storage: manure is stored and managed in stables/barns/enclosed or partially enclosed facilities.

B) Outdoor storage: manure is stored in open environments, i.e., an earthen basin or pond.

C) Covered storage: manure is stored in tanks or containers with a cover/crust on top.

D) Left on land: manure is left on pastures soon after it is removed from housing, or there is daily spreading of collected manure on fields.

The above divisions of MMS were implemented in AMCLIM–MMS for simulating $NH_3$ emissions, and each division may include one or two phases. It is worth emphasizing that the types of manure storage included in the model are a simplification. The current level of complexity is justified as adequate for large-scale/global modelling because it is unrealistic to simulate every specific practice in manure management given the computational costs and the additional uncertainty entailed from more assumptions on data and processes. AMCLIM represents divisions A, B and C of manure storage in different manners. Subsequent land spreading of manure N from these three divisions is simulated by the AMCLIM Land Module. By comparison, manure N from division D which has already been spread or left on land is not subsequently passed to the Land Module of AMCLIM, and the $NH_3$ emission is counted as manure management emission. As described in Table A1, manure can be used as fuel (burned) or converted to fuel (digester), which may cause significant $NH_3$ emissions, but this is not included in the AMCLIM model due to the uncertainty (limited studies) and scope of this study. Meanwhile, these types of management are only a small fraction across the globe. The amount of manure N used as fuel is not simulated further. In addition, there is unmanaged manure N from housing. Although it is not an MMS according to the definitions used in GLEAM, it is still critical because this fraction reflects a direct N loss from the agricultural system to the environment. Unmanaged N is quite common in a few regions and nations for some particular livestock production systems according to FAO (2018). These systems include "discharge", "dumping", "fishpond" and "public sewage", which have adverse impacts on local aquatic systems and ecosystems. Manure illegally discharged to water bodies is expected to contribute much less $NH_3$ emissions (because of dilution), but has other environmental implications (e.g. eutrophication). Manure N not managed at this stage is reported, but not simulated further in AMCLIM. This is treated as a loss or an untraceable term in AMCLIM–MMS.

### 2.3.2 Simulated processes in manure management systems

The AMCLIM model simulates manure management for livestock as a subsequent stage after housing, except in-situ storage of livestock excreta in pits or litter management for poultry, which are counted as part of the housing emissions. In AMCLIM–MMS, there are two types of manure under management: slurry and solid manure, corresponding to liquid manure and manure with a mixture of solid and liquid phases, respectively.

Liquid manure or slurry can have a Dry Matter (DM) content that ranges from 2% to 20%, depending on the amount of water added to the manure. As such, slurry refers to manure with a relatively low DM content, which consists mainly of urine, faeces and added water (Sommer et al., 2006; Vira et al., 2019). In this study, "liquid manure" used in the manure management section is the same as "slurry" in the land application section. Figure 1 illustrates the three types of storage for liquid manure: indoor, outdoor and covered storage.

The indoor storage of liquid manure and pit storage in animal houses are similar, as both reservoirs have high water content (however, as mentioned, it should be noted that $NH_3$ emissions from pits in animal houses are counted as included with housing emissions). The volatilization of $NH_3$ from indoor storage of liquid manure is calculated using the same method as for pit emissions (two-film model; see Sect.S1 in the Supplementary Material). The TAN pool of the storage unit is determined from the TAN pool from housing, conversion from other N species, loss through $NH_3$ volatilization, and removal when manure is used for land application, which can be expressed by the following equation:

$$\frac{dM_{TAN}}{dt} = \psi_{housing}(t, TAN) + F_{TAN} - F_{NH_3} - \psi_{to\ land}(t, TAN), \tag{8}$$

where $\psi_{housing}(t)$ is the function that represents the housing excreta that are transferred to the storage unit. The relationship between $\psi_{housing}(t)$ and the cleaning function $\psi_{cleaning}(t)$ can be expressed as:

$$\psi_{housing}(t) = \frac{\psi_{cleaning}(t)}{f_{store-housing}}, \tag{9}$$

$f_{store-housing}$ is the ratio of storage area to housing area. If the area for manure storage is smaller than the housing area, the pools of manure storage (per unit area) will be larger than housing, as manure concentrates in smaller areas (note that concentrations remain unchanged). The function $\psi_{to\ land}(t)$ represents stored manure used for land application within 24 h ($t_{to\ land}$), and is expressed as follows:

$$\psi_{to\ land}(t, excreta/N/H_2O) = \begin{cases} 0, & \text{if "Not an application day"}, \\ \frac{MM_{excreta/N/H_2O}}{t_{to\ land}}, & \text{if "an application day"}. \end{cases} \tag{10}$$

Similarly, the other N pools during storage can be expressed as follow:

$$\frac{dM_{N_i}}{dt} = \psi_{housing}(t, N_i) - K_{N_i}M_{N_i} - \psi_{to\ land}(t, N_i). \tag{4.13}$$

The water pool of the storage unit is determined by the initial water amount of animal excreta from housing, evaporation, and additional water that may be added ($F_{added\ water}$):

$$\frac{dM_{H_2O}}{dt} = \psi_{housing}(t, H_2O) + F_{added\ water} - F_{evap} - \psi_{to\ land}(t, H_2O). \tag{11}$$

By default, the DM content of liquid manure in AMCLIM–MMS is set to 5 %, but it is allowed to vary by a factor of 2, between 2.5 % to 10 %, due to fluctuations in the water pool. Additional water may be added to maintain the DM content within 10 % ($f_{DM.max}$), as expressed by the following equation:

$$M_{added\ water} = \max\left(0, M_{DM}\left(\frac{1}{f_{DM,max}} - 1\right) - M_{H_2O}\right). \tag{12}$$

Covered liquid manure storage is considered a variation of indoor storage in AMCLIM–MMS. A reduction factor of 0.95 is applied to the $NH_3$ emission from this management system, representing an effective mitigation by covering the manure with lids or coverings (Bittman et al., 2014).

Simulations of outdoor storage of liquid manure are similar to those of indoor storage, but the physical and chemical processes are affected by different environmental conditions. The primary difference is the level of turbulence, which is largely related to wind speed and significantly impacts $NH_3$ volatilization. While indoor storage provides a less "windy" environment,

external storage exposes liquid manure to the outside environment. Temperature differences between indoor and outdoor storage may be less pronounced. In addition, the water pool of outdoor storage is influenced by rainfall ($F_{rainfall}$, mm s$^{-1}$), as expressed by the following equation:

$$\frac{dM_{H_2O}}{dt} = \psi_{housing}(t, H_2O) + F_{added\ water} + F_{rainfall} - F_{evap} - \psi_{to\ land}(t, H_2O). \tag{13}$$

A specific management classified as outdoor storage in AMCLIM–MMS is lagoon systems. Lagoon systems are artificial or natural earthen storage structures that usually provide a largely anaerobic environment for liquid manure treatment. In this study, a simplified representation of lagoon systems is used, where a constant TAN concentration of 600 µg mL$^{-1}$ is set for lagoons (Aneja et al., 2001). This simplification is justified as reasonable due to the large amount of water present in lagoon systems, resulting in low TAN concentrations. Therefore, the instantaneous NH$_3$ emission from a lagoon system is expected

to be smaller than that directly from livestock excreta, which only disturbs the TAN pool to a limited extent. The process of NH$_3$ volatilization is simulated by the same two-film model as other liquid storage management systems (Sect.S1).

     Solid manure has higher DM contents than liquid manure, typically ranging from 30 to 40 % for pigs and ruminants, and up to 50 to 70 % for poultry manure (Sommer and Hutchings, 2001). With lower water content, solid manure storage can facilitate nitrification, providing an additional chemical pathway that depletes the TAN pool, as expressed by the following equation:

$$\frac{dM_{TAN}}{dt} = \psi_{housing}(t, TAN) + F_{TAN} - F_{NH_3} - F_{nitrif} - \psi_{to\ land}(t, TAN). \tag{14}$$

     The nitrification process in solid manure is similar to that in soils as presented in Jiang et al. (2024), but with some variations in parameters. The details of these calculations can be found in Sect.S4 in the Supplementary Material. In solid manure, ammonium can be adsorbed on solid particles, and the manure itself presents an additional barrier to N transport. In AMCLIM– MMS, the partition of TAN into different phases in the bulk manure is determined, and the concentrations at the surface are

used to calculate NH$_3$ emission. Further information is provided in Sect.S5 in the Supplementary Material.

## 2.4 Land application of livestock manure and ruminant grazing

### 2.4.1 Application of manure to land

Livestock manure can be used as fertilizer to land either after being stored for a period of time or directly after being removed from animal houses. The land application of manure is simulated by AMCLIM–Land, which employs the four prescribed soil

layers (as described in Jiang et al., 2024). The N processes involved in the simulations for manure application are the same as those for synthetic fertilizer applications (Jiang et al., 2024). Specifically, the volatilization processes of NH$_3$ have been described in Sect.2.2.1 in Jiang et al. (2024). Manure is assumed to be applied only to the soil surface. Modification to allow soil incorporation and deep injection of manure and slurry is possible, but is not included in the current version of AMCLIM reported here. Stored manure is assumed to be spread on land, and its application is scheduled according to the local crop

planting seasons. Alternatively, manure can be applied daily if it is spread soon after being removed from animal houses.

     Manure application to land provides sources of N to the soil pools. The soil TAN pool in the top layer can be expressed as:

$$\frac{dM_{TAN}}{dt} = \psi_{to\ land}(t, TAN) + \psi_{to\ land}(t, urea/org\ N) + F_{TAN} - F_{NH_3} - F_{TAN\ runoff} - F_{diffusion} - F_{leaching} - F_{nitrif}, \quad (15)$$

where the application rate $\psi_{to\ land}(t)$ hase been shown in Eq.(14). The production of TAN ($F_{TAN}$) is mainly through the decomposition of organic N. The remaining fluxes are removal processes ($F_{TAN\ runoff}$ – flux of surface TAN runoff; $F_{diffusion}$ – diffusive fluxes; $F_{leaching}$ – flux of leaching; $F_{nitrif}$ – nitrification).

Urea in manure is assumed to be fully hydrolysed to TAN during storage upon land application as a simplification, which keeps the soil pH constant. This is true for stored manure and is a reasonable assumption for daily spread manure. Uric acid in poultry manure and organic N are assumed to be retained in the top soil layer as these species typically bond with manure and soil particles, and are assumed in AMCLIM not to move to the underlying layers through diffusion or drainage. These N pools in soils are depleted by hydrolysis or decomposition and surface runoff, which can be expressed as:

$$\frac{dM_{N_i}}{dt} = \psi_{to\ land}(t, N_i) - K_{N_i}M_{N_i} - F_{N_i\ runoff}. \quad (16)$$

The runoff of N species ($F_{N_i\ runoff}$) such as uric acid and organic N is determined by the following equation:

$$F_{N_i\ runoff} = q_r r_N M_{N_i}, \quad (17)$$

where $r_N$ (mm$^{-1}$) represents the wash-off factor for N species that is set at 1 % per millimetre (Riddick et al., 2017).

The application of manure, particularly slurry, can affect soil water content. Misselbrook et al. (2006) reported that 6 mm of pig and cattle slurry infiltrate the soils within an hour after application and causes an increase in the soil moisture content. In AMCLIM–Land, the immediate change in soil water content after manure application is calculated (Jiang et al., 2024). However, the model does not account for the impact of manure application on soil properties, such as porosity or organic matter content. Additionally, AMCLIM–Land allocates N species in solid manure to the topsoil layer instead of a separate manure layer above the soils.

### 2.4.2 Ruminant grazing

Grazing practice is an important component of ruminant farming systems. Animals can spend the whole year or part of the year outside (i.e., on pastures or rangelands), corresponding to year-round and seasonal grazing, respectively. Based on the GLEAM livestock data, ruminants are categorised here into grassland and mixed production systems (Seré and Steinfeld, 1996). In AMCLIM–Land, ruminants in the grassland production system are assumed to graze year-round, whereas those in the mixed production system graze seasonally. This assumption has been made in the FANv2 model (Vira et al., 2020) and was used here. The NH$_3$ emissions during seasonal grazing are considered as a counterpart to the housing emissions. The N pools for seasonal grazing can be expressed as follows:

$$\frac{dM_{N_i}}{dt} = f_{grazing}F_{excretN}f_{N_i} - K_{N_i}M_{N_i}. \quad (18)$$

The amount of ruminant excreta deposited on pastures depends on grazing time, and is determined from the MMS information provided by the GLEAM model and a temperature condition. Specifically, the fraction of excreta deposited on pastures $f_{grazing}$, is calculated as follows:

$$f_{\text{grazing}} = \begin{cases} \begin{cases} \dfrac{f_{\text{MMS(pasture)}}}{N_{T_{10}^{\min}>10°C}/365}, if\ T_{10}^{\min} \geq 10°C \\ 0, if\ T_{10}^{\min} < 10°C \end{cases}, & \text{if } f_{\text{MMS(pasture)}} \leq \dfrac{N_{T_{10}^{\min}>10°C}}{365} \\[3em] f_{\text{MMS(pasture)}}, & \text{if } f_{\text{MMS(pasture)}} > \dfrac{N_{T_{10}^{\min}>10°C}}{365} \end{cases}, \quad (19)$$

where $f_{\text{MMS(pasture)}}$ is the fraction of annual total manure deposited on pastures. $T_{10}^{\min}$ (ºC) is the 10-day running average of daily

minimum temperature (calculated for each day of the year), and $N_{T_{10}^{\min}>10°C}$ is the number of days with $T_{10}^{\min}$ higher than 10 ºC in a year (Pinder et al., 2004). The temperature condition justifies the number of days suitable for grazing in a year, while the MMS statistical data constrains the annual total value of excreted N deposited on pastures. If the values of MMS data ($f_{\text{MMS(pasture)}}$) are smaller than the fraction of suitable days ($N_{T_{10}^{\min}>10°C}$) in the year, ruminants only graze on suitable days (i.e., when $T_{10}^{\min}$ is higher than 10 ºC). If the MMS value is larger, the AMCLIM model assumes that animals graze throughout the

year, but spend only a fraction of time outside on pastures daily. This situation counts as seasonal grazing in AMCLIM, even though animals graze year-round, as the grazing system is determined by the production system. Emissions during seasonal grazing can be crucial, particularly if animals are kept outside for a considerable time.

The AMCLIM-Land module includes two schemes for simulating these emissions: the urine patch scheme and the dung pat scheme, as shown in Fig. 3. The urine patch scheme is focused on $NH_3$ emission from urine deposition, while the dung pat

scheme considers $NH_3$ from both dung-only and dung/urine mixtures situations. These two schemes are analogous to land application of slurry and solid manure, respectively, with the same simulated processes as for the manure application to land.

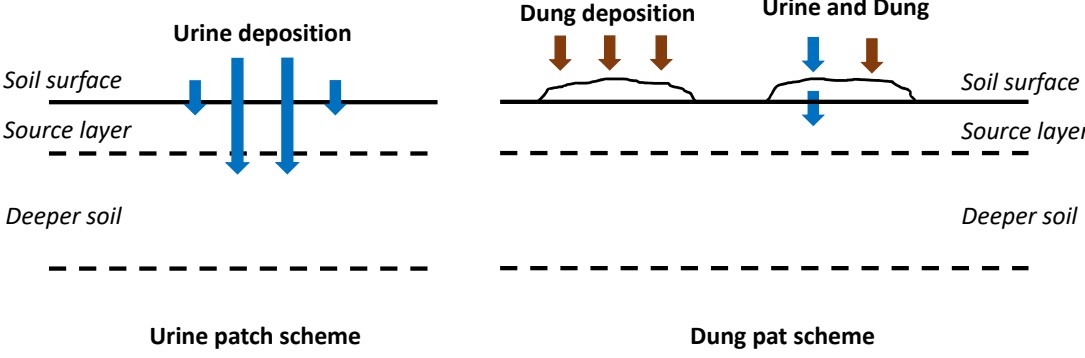

**Figure 3. Sketch of the urine and the dung patch schemes used in the AMCLIM-Land module for grazing simulations.**

Urine can infiltrate into soils relatively quickly and change the water content of the soil surface. Meanwhile, urinary N mainly exists as urea. Hydrolysis of urea in fresh urine results in soil pH change, which is different from slurry application where urea is assumed to be completely converted to TAN and not to affect soil pH. Another difference is the vertical soil layering. In the urine patch scheme, only the surface soil layer is modelled rather than all four soil layers as in simulations for fertilizer applications, for saving computational costs. Also, considering the smaller water volume of ruminant urine compared

with slurry application or irrigation, AMCLIM-Land defines a 4 mm source layer in which all simulated processes take place. The thickness of this source layer is based on Móring et al. (2016).

In the dung pat scheme, $NH_3$ is mainly emitted from the excreta rather than the underlying soils, as the excreta act as a substrate to hold the excreted N. An excreta layer is set up above the soil surface in the dung pat scheme, and the underlying soils are not further simulated. All simulated processes in both schemes are the same as those for the top soil-layer of manure

applications, and the transport distances for diffusive transport are modified accordingly.

Simulating $NH_3$ emissions from grazing is challenging due to the heterogeneity of grazing fields. It is crucial to determine the area of emitting surfaces with the matched N pools. Since animals roam freely and do not urinate and defecate in the same area during every excretion event, fresh excreta do not accumulate on old excreta. In AMCLIM, excreted N from each day are simulated independently and does not accumulate in the common pools. Each day's excreta go into new pools instead of being

added to the previous day's pools. The total $NH_3$ emission from a grazing field can be calculated by the following equation:

$$F_{NH_3} = \sum_{n=1}^{60} F_{NH_3(n)}, \tag{20}$$

where $F_{NH_3(n)}$ represents the $NH_3$ emission from the area where excreta are deposited on day n. Pools from each day are simulated for 60 days, after which, all N pools are assumed to be naturally incorporated into soils and are not simulated further. During the simulation period, input is only from the first day of this 60-day window, and the source area for emissions of each

day is a constant value under the assumption that daily excretion rates (urine and dung) remain the same.

## 2.5 Site-based and global simulations of $NH_3$ emissions from livestock farming

### 2.5.1 Site simulations of housing $NH_3$ emissions

AMCLIM–Housing was applied at site scale and used monitored data from experimental farms of Animal Feeding Operations

(AFOs) to simulate site-specific $NH_3$ emissions from pig, chicken and dairy cattle houses. The AFO monitored data were gathered by the US Environmental Protection Agency (EPA) as part of a study of emissions from several types of livestock from 2007 to 2010 (Lim et al., 2010a; Wang et al., 2010). Four houses with slatted floor and pit storage from a pig farm in Indiana (site IN3B) were selected for the simulations, together with two layer-houses from a chicken farm in North Carolina (site NC2B) and two free stall-barns in a dairy farm in Indiana (site IN5B), as listed in Table A3. The AFO datasets provided

animal data and daily mean environmental data for the two sites. Animal data included animal numbers, body weight and biomaterial data. Environmental data included indoor and outdoor temperature and relative humidity and the interior ventilation given as an airflow rate in cubic meter per second ($m^3 \ s^{-1}$). To keep simulations continuous, missing values in the environmental data due to the unavailable measurements were filled by linear interpolation. AMCLIM–Housing used excreted N that was determined from the livestock excreta data (as shown in Table A1) as an input, together with the indoor

environmental data. Further information on the measured farms can be found in the USEPA AFOs reports (Lim et al., 2010a; Wang et al., 2010), and a summary of model inputs are presented in Table A4. It is worth noting that the evaluations focused

on $NH_3$ emissions from housing, as the processes involved in manure storage are similar to those in housing, and there were limited available measurements for $NH_3$ emissions from manure storage. Additionally, the land simulations for TAN application were evaluated against other datasets as discussed in Jiang et al. (2024).

## 2.5.2 Global simulations of livestock farming $NH_3$ emissions: input and model setup

Once the AMCLIM model has been applied at site scale and evaluated against measurement data, the focus is then on applying the model to global scale. In the present paper, the combined AMCLIM model was applied for 2010 and 2018 to demonstrate full simulations for two different years, with activity data and meteorological variables varied between years, so that the inter-annual variability in both emissions and volatilization rates can be analysed. Global simulations of the AMCLIM model were driven by hourly meteorological inputs from the European Centre for Medium-Range Weather Forecasts Reanalysis v5 (ERA5) reanalysis collection (Hersbach et al., 2020), which has been detailed in Jiang et al. (2024). In addition to the input data that used by the Land module of AMCLIM (AMCLIM–Land), activity data including livestock and MMS information are required for simulating livestock farming. The global livestock and MMS data used in AMCLIM are obtained from FAO GLEAM. The global livestock data include information on the geographical distribution of livestock heads, average liveweight, and total N excretion rates, which are categorized by production system. The global livestock populations were based on FAOSTAT data for 2010 (FAOSTAT). The geographic distributions were based on the Gridded Livestock of the World (GLW) model, which produced density maps for the main livestock species based on observed densities and explanatory variables such as climatic data, land cover and demographic parameters (Robinson et al., 2014). The reference year of these data is 2010. For simulations for the year 2018, livestock population and N excretion rates were extended by a linear interpolation based on the inter-annual variations between 2005 and 2015 suggested by Lu and Tian (2017). The MMS data that determines the fraction of a manure management system are assumed to be constant through the year. Excretion rates of each livestock are derived from the difference between nitrogen intake and retention based on the GLEAM approach. A summary of model input data for global simulations is given in Table A4. More information on the properties and characteristics of livestock excreta, including urinary N concentrations, faecal N content, dry matter content and pH, are presented in Table A1 in Appendix.

For pig farming, three production systems are used in the global simulations: industrial, intermediate and backyard. For poultry, only chicken are included, which account for over 95 % of poultry by numbers based on the FAO (FAOSTAT) data for 2010. Chicken have three production systems: broilers, layers and backyard chicken. Ruminants have two production systems: grassland and mixed production systems, except for feedlot cattle, which is treated as a specialized production system: feedlot. In feedlots, cattle are fed with a specialized diet to stimulate weight gain. They are normally kept in concentrated areas to facilitate the fattening processes with high stocking densities according to FAO (2018). The characteristics and housing features of livestock production systems can be found in GLEAM. The MMS data provide geographical distributions of the fraction that a system is used for manure management, which differs between livestock sectors and production systems for pigs and poultry. As described in previous sections, these MMS are regrouped into the four divisions used in AMCLIM–MMS. More details are available in Table A2 in the Appendix. Land application of manure is assumed to take place throughout the

spring and winter planting seasons, respectively. All stored manure is applied on fields without explicitly simulating vegetation cover. Sect.S6 in the Supplementary Material gives detailed global setups for each practice.

To estimate the environmental conditions in livestock houses, empirical relationships between outdoor temperature and indoor environments including temperature and ventilation were developed based on data from the Animal Feeding Operations (EPA, 2012) and theoretical parameterizations of indoor conditions by Gyldenkærne et al. (2005). Equations that present the

relationships between indoor temperature, ventilation and outdoor temperature for different housing systems are given in Sect.S6.1. The RH of indoor environments is assumed to be equivalent to outdoor RH.

Global simulations were performed to estimate $NH_3$ emissions from livestock farming for the year 2010 and 2018, and had the consistent setup as simulations for synthetic fertilizer use, as described by Jiang et al. (2024). AMCLIM was applied using a longitude–latitude grid at a resolution of $0.5° \times 0.5°$. All model inputs were regridded to the model resolution if necessary.

The simulations were performed at an hourly time step, and the prognostic variables at each time step were solved by the Euler method in the model.

## 2.6 Update of AMCLIM-Poultry model

Jiang et al. (2021) previously described the development of the AMCLIM-Poultry model ("Poultry Model" for short in the following text), which provided a starting point and a pilot study that uses a process-based model to simulate $NH_3$ emissions

from global chicken farming. The Poultry Model has been incorporated into the full AMCLIM model as a component unit, and several processes have been improved. Major advances in the current AMCLIM model (for simulating poultry farming) compared with the Poultry Model include the following:

- The adsorption of TAN on manure particles is included in the current AMCLIM by using a linear equation (Sect.S3.2) that describes the equilibrium between aqueous TAN and solid exchangeable TAN.

- Initial water content of the excreta is considered rather than assuming an immediate equilibrium moisture content of the excreta.

- Other organic forms of N in the excreta are included in addition to uric acid.

- A separate manure management stage is included by operating the AMCLIM–MMS. Litter management is distinguished from other management.

- Housing of backyard chicken and subsequent manure management replace the original "manure left on land" scenario, according to the characteristics of the production system and the corresponding MMS information (Table A1).

- The simulations for housing were operated in the updated AMCLIM model at an hourly time-step instead of daily time-step of the original Poultry Model.

-   Land application of manure is simulated by the Land Module of AMCLIM, which includes more soil processes and N pathways and employs a four-layer soil profile, as compared with the simpler land application scheme in the Poultry Model.

-   Nitrogen application rates are derived from recommended or reference manure application rates (Sect.S6.2).

Other smaller changes in the AMCLIM model include:

-   The new resistance scheme in the poultry houses consists of a resistance for gas transfer and a litter resistance rather than using a single constant housing resistance in the Poultry Model. The newly parametrized gas transfer resistance is dependent on temperature and ventilation inside the house, while the litter resistance is a constant value used the same inversion method as in the previous Poultry Model (see Sect.S3.2).

-   Manure is no longer only applied to the six prescribed crops based on expert judgement. Instead, manure is assumed to be applied to land depending on a generalised crop calendar which is derived from major crops (see Sect.S6.2).

## 3 Results

### 3.1 NH$_3$ emissions from individual animal houses

The focus of results presented here for the site simulations is the pig houses and dairy barns. Results of layer house simulations are provided in Sect.S7 in the Supplementary Material to avoid repeated content that has previously been presented by Jiang et al. (2021), together with a summary update compared with the prior publication. These site simulations were conducted to evaluate the model performance prior to conducting the global simulations.

### 3.1.1 Pig houses with slat and pit

Figure 4 shows the results of simulated NH$_3$ emissions from a pig house in Indiana, US, with slatted floor and pit storage, alongside comparisons with measurements, stocking data, and the indoor environments (simulations for other similarly managed houses are shown in Fig.A1 to A3 in Appendix). The simulated period is two years from 1 July 2007 to 31 July 2009. Gaps shown in the figure represent unavailable measurements, while the model was kept running to produce a continuous output. The indoor temperature of the pig house ranged between 20 to 30 °C, showing moderate daily and seasonal variations, with higher temperature in summer than winter. There were two obvious temperature drops in March 2008 and March 2009 due to the emptying of pigs from the house, as illustrated in Fig.4b. This also led to low values of TAN concentration on slats during the simulation periods. By contrast, the airflow rate inside the house shows significant seasonal variabilities, with higher ventilation occurring in summer (to keep animals cool) and lower ventilation in winter (to keep animals warm). The relative humidity exhibits strong daily variations, ranging from 40 % to 80 %.

There were several growth cycles of pigs in this farm during the simulated period (Fig.4b). Over 2000 weaner pigs started in the house, and half of the pigs were moved to other houses after three to four weeks once the pigs gained sufficient weight. As a result, the house had twice as many pigs at the beginning of each growth cycle. Approximately 1000 to 1200 pigs were kept in the house during the subsequent fattening stage.

Measured daily $NH_3$ emissions from the pig house generally increased as ventilation increased. High emissions occurred mostly in summer, with the highest daily values of over 25 kg $NH_3$ $d^{-1}$ in July 2007. AMCLIM–Housing is able to reproduce the overall trend of measured $NH_3$ emissions in the first year from July 2007 to July 2008. However, it underestimates the winter emissions (January 2009) by 30 %, which might be caused by overestimated resistances because the simulated TAN concentrations were comparable to the measurements (Fig. 4e). AMCLIM overestimates the summer emissions (June 2009 and July 2009) by a factor of two (Fig. 4c), corresponding to higher simulated TAN and total N concentrations of the slatted floor and the pit than those of measurements (Fig. 4e). This is possibly due to underestimations of indoor evaporation in AMCLIM. The average modelled daily $NH_3$ emission is 10.4 kg $d^{-1}$ (when measurements available; 9.9 kg $d^{-1}$ for the entire simulation period), compared with 8.8 kg $d^{-1}$ recorded by the measurements. According to AMCLIM–Housing, 42 % of this animal house's total excreted N volatilizes as $NH_3$. The slats and the pit are estimated to contribute to 57 % and 43 % of the total emissions, with the average daily emissions being 5.7 kg $d^{-1}$ and 4.2 kg $d^{-1}$, respectively. As shown in Fig.4d, simulated $NH_3$ emission originating from the slats is typically larger than from the pits, especially in summer when the ventilation is high. Modelled slat $NH_3$ emissions increase periodically throughout the simulated period, which is closely associated with the animal mass in the house. The blue dashed lines in Fig.4d and 4e show when the pit was cleaned. Modelled TAN concentrations on the slats are compared with the measurements, as well as the N concentrations in the pit, with reasonably close agreement being found between the modelled and measured values (Fig.4e). For the other animal houses, assessed 41% to 42% of the excreted N was estimated to be emitted as $NH_3$ (Fig. A1-A3).

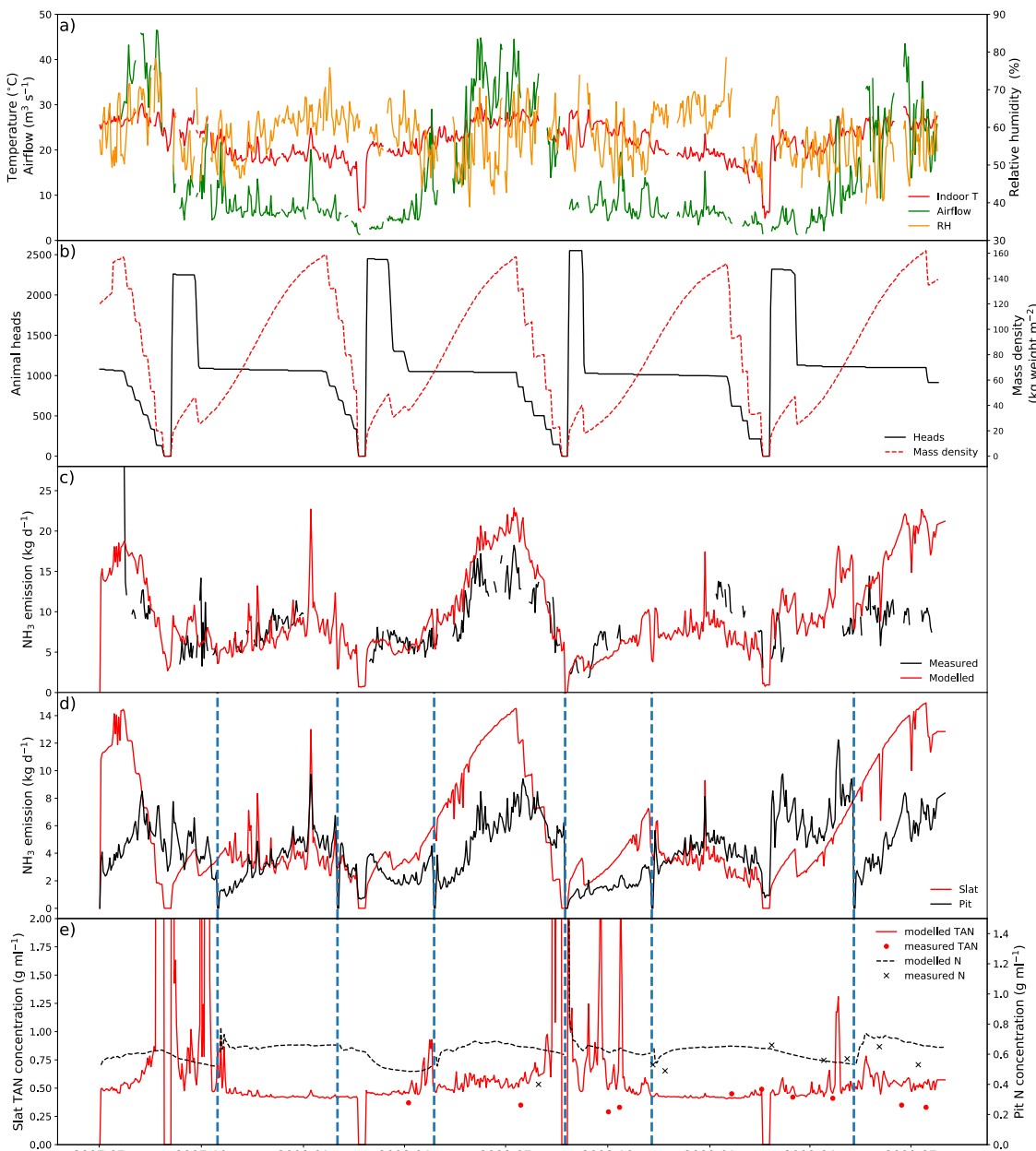

**Figure 4.** Site simulations of House 1 in a pig farm at site IN3B, Carroll, Indiana, from 01 July 2007 to 31 July 2009. (a) Measured daily mean indoor temperature, airflow rate and relative humidity of the house. (b) Animal heads and mass density of the house. (c) Comparison between modelled NH₃ emissions and calculated NH₃ emissions from measured indoor concentrations. (d) Modelled NH₃ emissions from the slats and the pit. (e) Comparison between measured and modelled TAN concentration of the slats and between measured and modelled N concentration of the pit. Vertical blue dashed lines refer to excreta removal from the pit. See Fig. A1 to A3 in the Appendix for results from other pig houses.

### 3.1.2 Dairy barns

AMCLIM–Housing was applied to simulate $NH_3$ emissions from two free-stall dairy barns in Indiana, US. Measurements at these farms were available from two years of monitoring. Each barn contained around 1600 Holstein cows. The barns had exhaust fans to facilitate ventilation, and scrapers were used to clean the barn floors and remove manure. More information about the farm can be found in Lim et al. (2010). In the simulations, the cleaning events were assumed to take place every day to represent manure removal by scrapers (Eq. 6-8). As a result, the N pools and manure pool were reset on a daily basis. Manure removed from barns was not simulated further.

The simulated period is from 01 July 2007 to 31 July 2009, as shown in Fig. 5 (simulations for other similarly managed barns are shown in Fig. A4 in the Appendix). The daily average temperature inside the barn is very close to the outdoor temperature, ranging from -10 to 25 °C (Fig. 5a). Strong seasonal variations are found in ventilation, with higher ventilation in summer and lower ventilation in winter, while inside temperature exhibits the same trend. The relative humidity also shows strong daily variations (higher at night), with the highest RH being over 85 % and lowest values being below 55 %.

Overall, AMCLIM–Housing reproduces the $NH_3$ emissions well and captures the daily and seasonal variations. The average modelled $NH_3$ emission from the dairy barn is 32.4 kg $d^{-1}$ (when measurements are available; 35.2 kg $d^{-1}$ for the entire simulation), compared with 32.5 kg $d^{-1}$ reported by the measurements. As shown in Fig. 5, high $NH_3$ emissions occur not only in summer but also in spring, especially in 2009, resulting from high temperature and high ventilation. Meanwhile, emissions decrease in winter when both temperature and ventilation are low. The highest emission is over 100 kg $d^{-1}$ in April 2009, while the lowest emission is less than 10 kg $d^{-1}$ in winter days. According to the model for this farm, 15 % of excreted N from dairy is lost due to $NH_3$ emissions for both simulated barns. Overall, the lower volatilization rate for the Indiana cattle houses compared with the North Carolina pig houses (42%) can be attributed in the model to a combination of a) cooler temperatures and b) scraped floor (removing manure to a separate store).

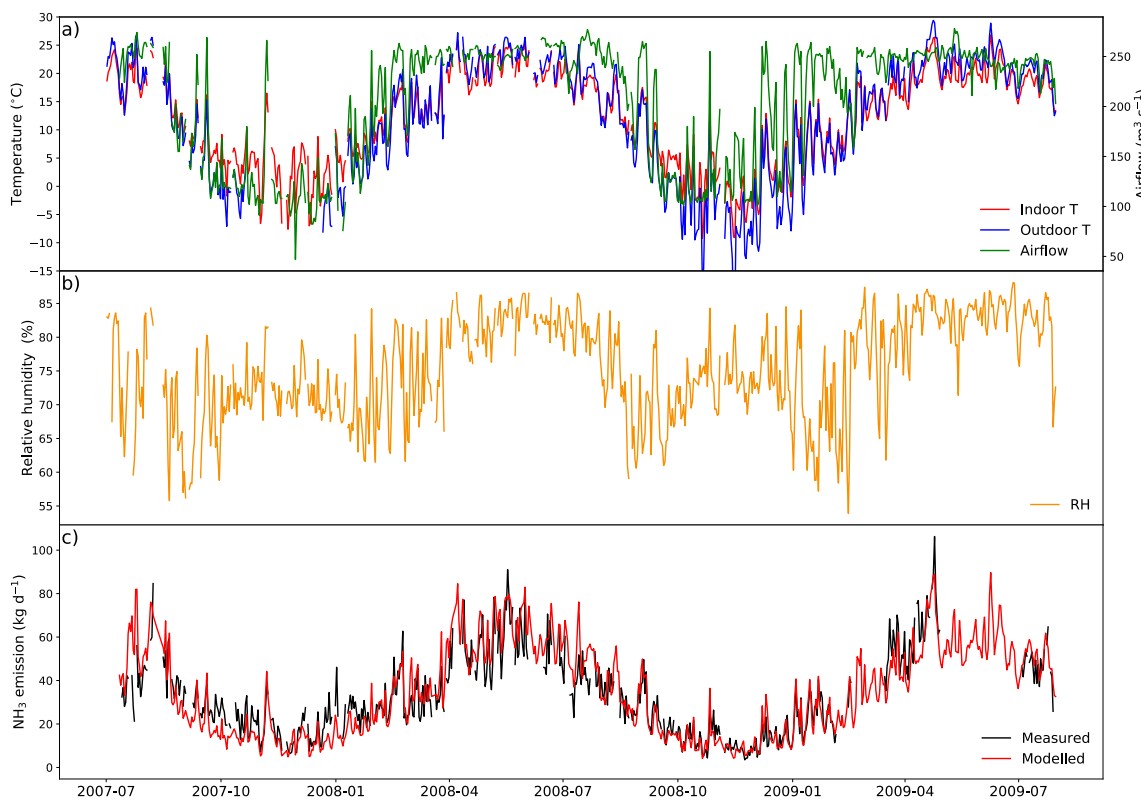

**Figure 5. Site simulations of Barn 1 in a dairy farm at site IN5B, Jasper, Indiana, from 01 July 2007 to 31 July 2009. (a) Measured daily mean indoor temperature, indoor airflow rate of the barn and outdoor temperature. (b) Measured daily mean relative humidity of the barn. (c) Comparison between modelled NH₃ emissions and calculated NH₃ emissions from measured indoor concentrations. See Fig. A4 in the Appendix for results from the other dairy house.**

### 3.1.3 Sensitivity tests for model parameters of AMCLIM–Housing

Sensitivity tests were conducted to examine the effects of changes in model parameters on the simulated $NH_3$ emission from animal housing, including results for the pig and dairy simulations described above, and for the chicken house simulations described in Sect.S7. Nine model parameters with varying ranges were selected for the sensitivity analysis, based on expert judgement, and the corresponding percentage changes in the $NH_3$ emissions are highlighted in Table 1. The estimated pH of excreta used in AMCLIM is identified to be the most important parameter that has significant impacts on the $NH_3$ emissions, especially for the dairy simulations. Varying the evaporation of water in animal houses ($F_{evap}$) by a factor of 2 only results in very small changes in emissions compared with other parameters. Moreover, changes in $NH_3$ emissions from layer chicken housing are almost negligible when varying the indoor $NH_3$ concentration by a factor of 2 or set to a constant zero, which demonstrates the feasibility of neglecting the indoor $NH_3$ concentration in global simulations for chicken housing. The same assumption was also applied to simulations for other livestock.

The NH$_3$ emissions from housing of all livestock change by the same extent as the changes in N excretion rates ($F_{excretN}$). The impact of the housing resistance ($R_{g, house}$) on NH$_3$ volatilization in the animal houses is different: NH$_3$ from layer chicken housing is much less influenced by the housing resistance than pig and dairy housing. Litter resistance ($R_{litter}$) plays a more dominant role in affecting the emission than housing resistance for layer chickens, as well as for cattle and pigs. The partitioning coefficient for TAN adsorption on excreta solids ($K_d$) is also important for NH$_3$ emission from layer chicken

housing. Excluding the adsorption leads to a 60 % of increase of NH$_3$ emissions, while doubling the adsorption results in nearly 30 % of less NH$_3$. Although ammonia emissions from chicken excreta are known to be dependent on the uric acid hydrolysis, rate, doubling the uric acid hydrolysis rate ($K_{UA}$) only resulted in NH$_3$ emissions increasing by 9 %, while the emissions decrease by 15 % if the hydrolysis rate is halved. This indicates that much of the uric acid was ultimately hydrolyzed, so that the hydrolysis rate mainly affected the time-course of emissions rather than the total magnitude of annual emissions.

For pig housing, varying the excreta water ($F_{urine}$ and $F_{fecal\ water}$) by 20 % results in around 5 % changes in NH$_3$ emissions. Doubling or halving the urea hydrolysis constant ($k_h$; details given in Sect.S2) almost has no impacts on the NH$_3$ emission. Rapid urea hydrolysis in pig slurry indicates that even with the sensitivity tests, almost all excreted urea is hydrolyzed to ammonia in these AMCLIM simulations. Increasing the gap space of the slatted floor ($f_{gap}$) from 0.2 to 0.3 of the house leads NH$_3$ emissions to decline by 8 %, while the NH$_3$ emission increases by 9 % when decreasing the gap space to 0.1. Aarnink et

al. (1997) found that more open space on the slatted floor significantly reduces NH$_3$ emissions from the slats, which could explain the decline in total NH$_3$ emissions from the simulated pig houses.

**Table 1. Percentage changes in NH$_3$ emissions from pig/dairy and layer housing in the sensitivity tests for the parameters in AMCLIM.**

| Model parameters | Value tested | ΔNH$_3$ emission % (pig/dairy) | ΔNH$_3$ emission % (layer) |
|---|---|---|---|
| $F_{excretN}$ | +10 % | +10.0/+10.0 | +10.0 |
| | −10 % | −10.0/−10.0 | −10.0 |
| pH of excreta | +0.5 | +22.1/+133.0 | +46.1 |
| | −0.5 | −37.4/−68.0 | −50.4 |
| $F_{evap}$ | 0.5× | −3.4/−2.3 | −1.3 |
| | 2.0× | +2.0/+5.1 | +0.6 |
| $R_{G, house}$ | 0.5× | +15.4/+73.6 | +1.1 |
| | 2.0× | −21.1/−46.7 | −2.0 |
| $R_{litter}$ | 0.5× | − | +32.3 |
| | 2.0× | − | −33.1 |
| $K_d$ | +100 % | − | −27.5 |
| | −100 % | − | +59.7 |
| $K_{UA}$ | 0.5× | − | −14.5 |

| | | | |
|---|---|---|---|
| | 2.0× | – | +9.0 |
| $F_{urine}$ and $F_{fecal\ water}$ | +20 % | −5.6/−17.3 | – |
| | −20 % | +4.1/+26.5 | – |
| $k_h$ | 0.5× | −0.5/−1.5 | – |
| | 2.0× | +0.03/+8.9 | – |
| $f_{gap}$ | +0.1 | −7.9/none | – |
| | −0.1 | +8.9/none | – |
| $\chi_{in}$ | 0 | – | +0.4 |
| | 0.5× | – | +0.2 |
| | 2.0× | – | −0.4 |

$F_{excretN}$ – total N excretion rate from the livestock; $F_{evap}$ – evaporation flux of water; $R_{G,\ house}$ – resistance for NH$_3$ volatilization in the house; $R_{litter}$ – poultry litter resistance; $K_d$ – adsorption coefficient of TAN on excreta solids; $K_{UA}$ – uric acid hydrolysis rate; $F_{urine}$ – water from urination; $F_{fecal\ water}$ – water in feces; $k_h$ – urea hydrolysis constant; $f_{gap}$ – gap space of the slats; $\chi_{in}$ – indoor concentration of NH$_3$

## 3.2 Global simulations for livestock housing, manure management and land application of manure

In the following sections, emissions are presented in the order of livestock housing, manure management and application of manure, while emissions from grazing are considered in section 3.3. As with the previous sections, pigs are presented first, followed by poultry, as representing systems dominated by all-year animal housing. Then ruminants including cattle, sheep and goats are presented, as systems which are complicated by the widespread practice of partial-year housing.

### 3.2.1 Pig NH$_3$ emissions and volatilization rates

Figures 6 and A5 show the geographical distributions of NH$_3$ emissions from pig agriculture and the volatilization rates for 2010 and 2018. For housing, the volatilization rates ($P_V$) are expressed as percentage of total N by excreted by livestock in the animal houses. For manure management and application to land, the volatilization rates are expressed as percentage of the total remaining N from the previous stage that is volatilized as NH$_3$. For pig housing and manure management, the spatial distributions of both emissions and volatilization rates are similar for both years. The highest volatilization rates of pig housing (over 35 %) are found in Australia, US, Thailand, Malaysia, the northern Africa and the northern South America. European countries and Brazil also show relatively high volatilizations rates between 20 and 30 %, while the rest of the world show low to moderate volatilization, ranging from 5 to 20 %. China, with the highest pig housing emissions, generally have low simulated $P_V$ rates of around 10 %. In contrast, Australia shows high $P_V$ rates but low emissions. For manure management, manure N volatilizes as NH$_3$ can be over 30 % in India, northwestern Australia, Southeast Asia, Africa and several countries in South America, while other regions typically have volatilization rates less than 20 % (Fig. 6d).

As shown in Fig.6e, in 2010, high total simulated emissions resulted from pig manure application to land mostly occur in China and Europe. Conversely, high volatilization rates are found in several places across the globe (Fig.n6f): The highest volatilization rates that exceeded 50 % can be seen in northern and southern Africa, India and western Australia. China, Southeast Asia, Europe, US and South America showed slightly lower volatilization rates, but also higher than 30 % (Fig. 6f). Only certain countries in Africa, Canada, Scandinavia and northern and eastern Russia exhibit lower $P_V$ rates less than 30 %. The volatilization rates $P_V$ in 2018 are sometimes lower than 2010 in several regions, such as southern Russia and western US (Fig. A5). The fact that several national boundaries can be seen in the $P_V$ maps (Figs. 6b, d, f) shows how these are not just affected by changes in environmental conditions (which do not generally change suddenly at national boundaries), but also by input datasets that are linked to national conditions in other model inputs. This especially concerns differences in assumed animal liveweights, housing practice and manure management as available from the GLEAM model database, which incorporates national estimates.

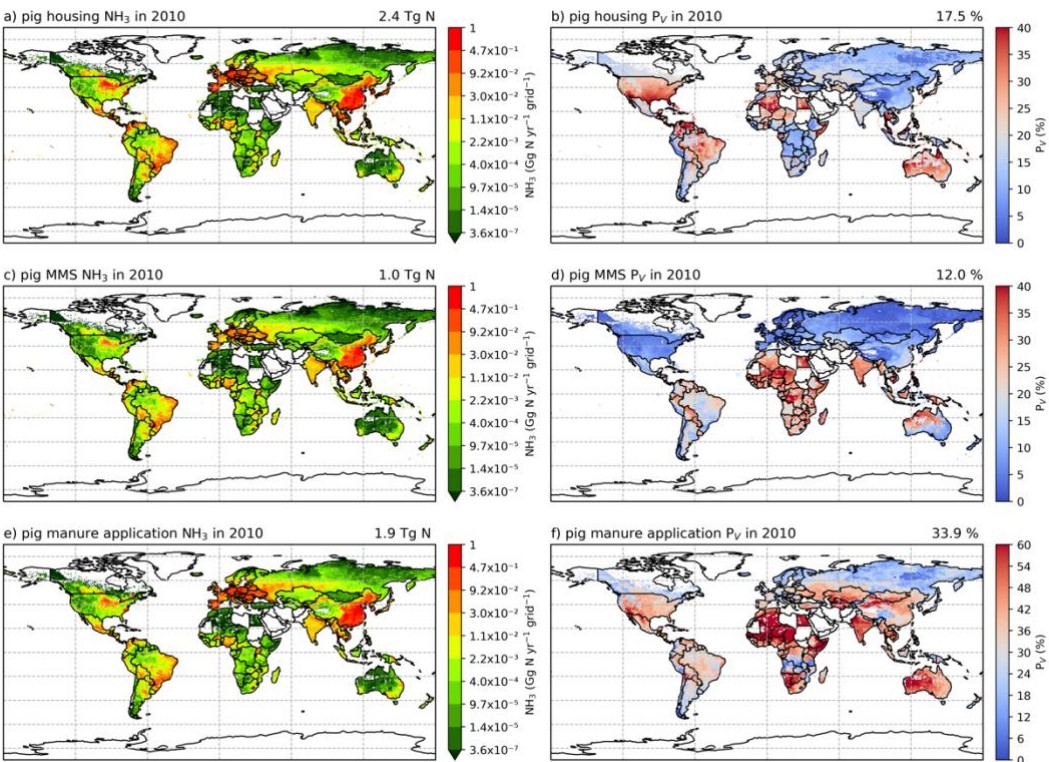

**Figure 6. AMCLIM pig simulations for the year 2010: NH₃ emissions (a, c, e) and percentage volatilization rates ($P_V$) (b, d, f), for housing, MMS and manure application, respectively. Global total NH₃ emissions and global average $P_V$ values for each activity are shown at the top right of the maps. The resolution is 0.5° × 0.5 °. White areas indicate zero activity data.**

### 3.2.2 Poultry (chicken) NH₃ emissions and volatilization rates

As shown in Fig.7 and A6, housing and manure application show much higher simulated volatilization rates than manure management. For housing, highest $NH_3$ volatilization rates of around 40 % are found in tropical regions along the equator, such as northern South America, central Africa and Southeast Asia. Meanwhile, India and southeast China also show high $P_V$ rates of over 30 %. Northern Africa, the Middle East and western Russia have moderate $P_V$, ranging between 20 to 30 %, while the other parts of the world have $P_V$ rates of less than 20 %. In contrast, volatilization rates for poultry manure management

are generally lower than 20 % across the globe, with larger values occasionally occurring in southeast Asia, Africa and the Middle East. For manure application to land, the highest volatilization rates can exceed 50 %, which can be seen in India, Australia, Mexico, part of US, northern and southern Africa and the Middle East in both 2010 and 2018 (Figs. 7f and A6f). China, Europe and South America also show high volatilization rates of over 30 %. Volatilization rates are generally lower for the year 2018 than 2010, with a clear difference seen in Southeast Asia. Since the current model version has updated processes

for simulating $NH_3$ emissions from poultry agriculture, a comparison of results between current model version as described in this section and the previous model version by Jiang et al. (2021) is discussed in Section 4.5.

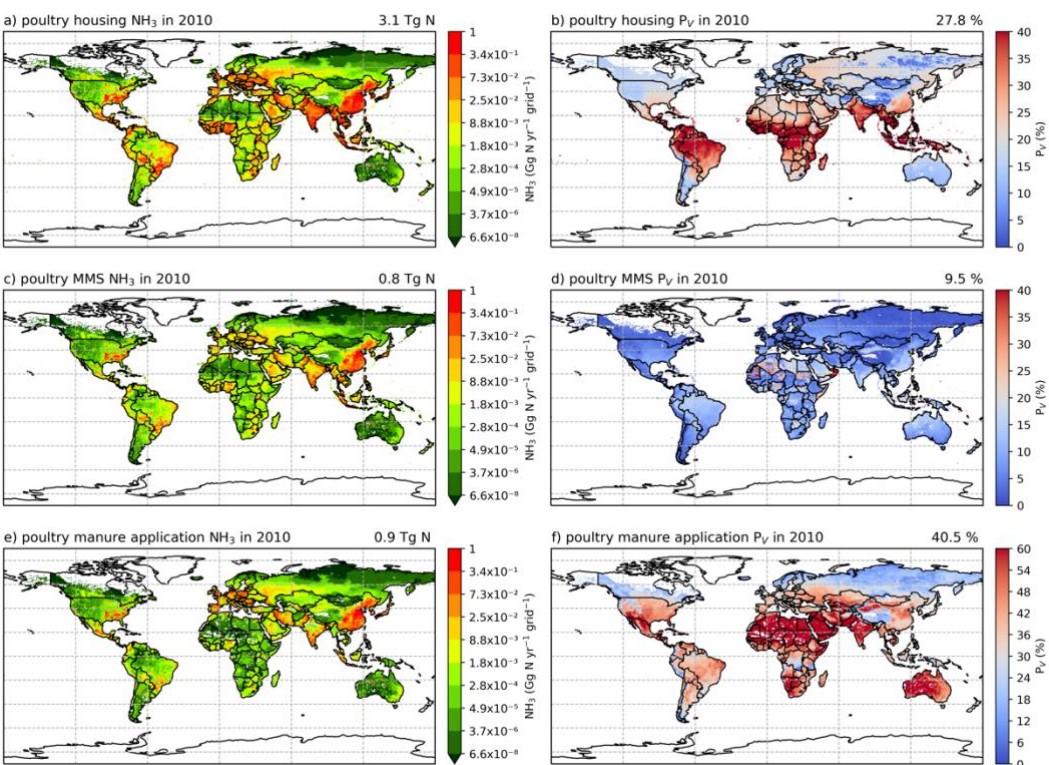

**Figure 7. The same as Figure 6, but for poultry (chicken).**


### 3.2.3 Cattle NH₃ emissions and volatilization rates

The geographical distributions of total simulated cattle $NH_3$ emissions and the volatilization rates are shown in Figs. 8 and A7, with no clear difference found for housing and manure management between the two years. The volatilization rates for each activity show different patterns. For housing (Figs. 8b and A7b), countries in South America, such as Bolivia, Brazil, Paraguay and Venezuela show the highest volatilization rates of over 25 %, as well as New Zealand. Part of India and several Sahel countries show moderate volatilization rates of 15 to 20 %, while the other regions in the world generally have volatilization rates of less than 10 %. For the volatilization rates of manure management (Figs. 8d and A7), highest rates of more than 35 % are found in countries in northern South America (Bolivia, Brazil, Paraguay and Venezuela), the Sahel and Southeast Asia. India, Pakistan, central and southern Africa also show high volatilization rates of over 20 %. China, US and Europe have lower volatilization than the regions mentioned above, with typically less than 10 % of N lost through $NH_3$ emissions. By comparison, the percentage volatilization rates are high across most of the regions on the globe (>36 %), with India having particular high rates of nearly 60 % (Fig. 8f). Only Brazil, northern Europe and Southeast Asia show lower rates of less than 30 %. Compared with 2010, Russia, eastern US and Europe show lower volatilization rates, while volatilization rates remain high in Australia, India, NCP and western US. Emissions of $NH_3$ related to manure application across Africa are estimated to be small (see Fig. 8e). This is mainly due to very little manure applied to crop fields in these regions. In AMCLIM, only stored manure is subsequently applied to land as fertilizer (as explained in Sect.2.3.1), while the majority of cattle excreta in Africa was estimated to be left outside without management according to GLEAM data. Manure left on land without further management is assumed to decompose and be naturally incorporated into soils.

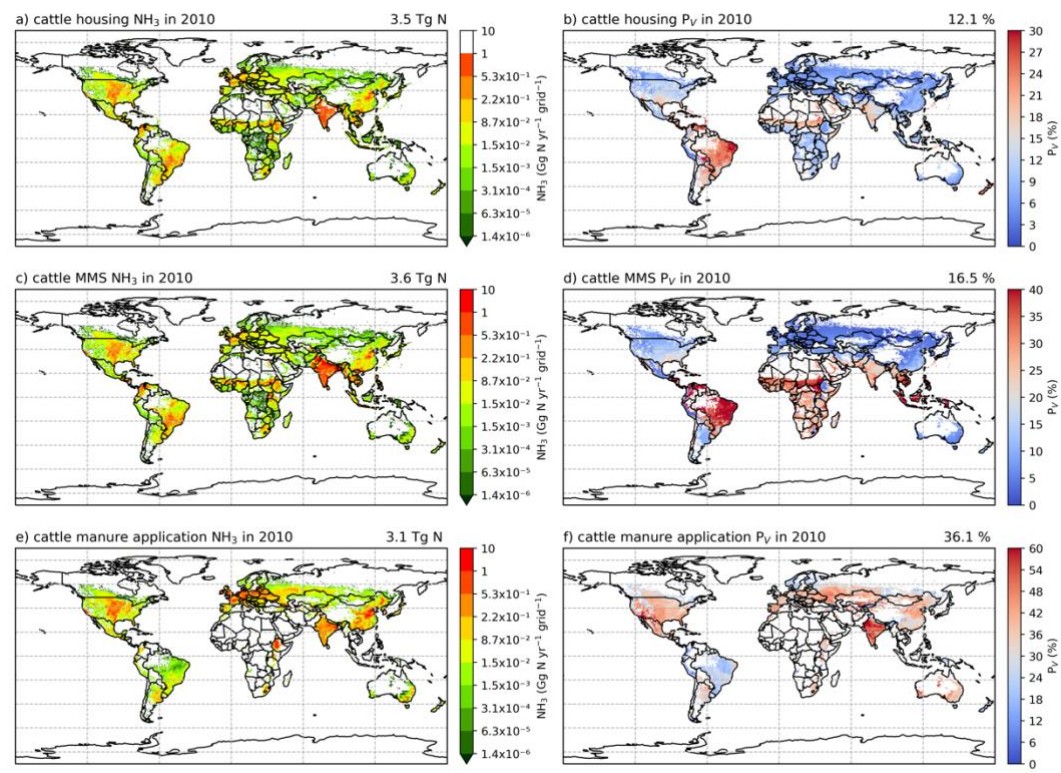


**Figure 8. The same as Figure 6, but for cattle. For many parts of Africa, cattle manure is not recorded as being applied to cropland or farmland, hence the extended white areas (no activity data for land application of manure).**

### 3.2.4 Sheep and goat NH$_3$ emissions and volatilization rates

As shown in Fig.9 and A8, India, NCP and Europe typically show higher simulated total NH$_3$ emissions resulting from sheep and goat farming than other regions in the world. However, regions with high emissions are not always consistent with regions with high percentage volatilization rates. For example, highest simulated housing volatilization rates are found in Africa, South Africa and southern North America, while Asia and Europe generally have lower volatilization rates of less than 20 % (Fig.9b). Overall, the $P_V$ rates for manure management are higher than housing, with large $P_V$ found in Africa, Central and South Asia,

Europe, South America and Southeast Asia (Figs. 9b and 9d). For manure application, the corresponding emissions are lower than emissions from housing and manure management, but the volatilization rates are higher. High emissions mostly can be seen in India, NCP, Pakistan and Europe. High volatilization rates for MMS are found across all the regions with emissions, while high volatilization rates occur only in NCP, Spain, western US and South Asia.

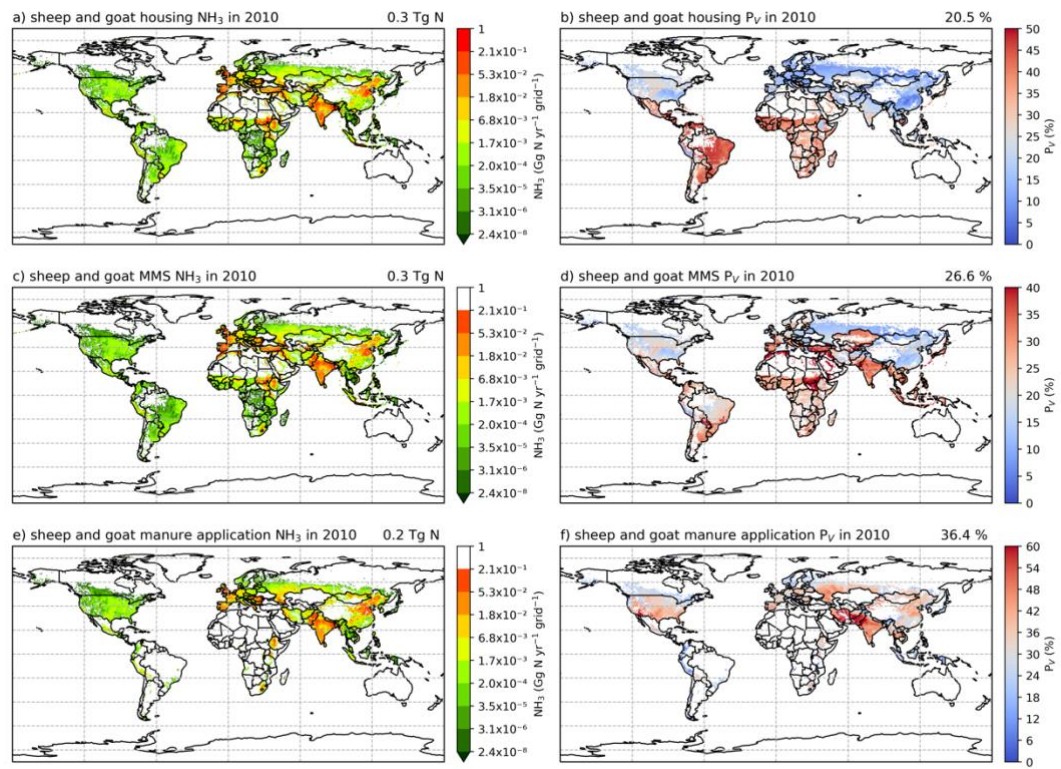


**Figure 9. The same as Figure 6, but for sheep and goat. For many parts of Africa, sheep and goat manure is not recorded as being applied to cropland or farmland, hence the extended white areas (no activity data for land application of manure).**

### 3.3 NH₃ emissions from ruminant grazing and volatilization rates

#### 3.3.1 Seasonal grazing and year-round grazing

The simulated NH₃ emissions from ruminant grazing comprise two parts: emissions from seasonal grazing and year-round grazing. For the seasonal grazing, N excreted by the mixed production system cattle is estimated at 15.3 Tg N yr⁻¹ in 2010 and 15.8 Tg N yr⁻¹ in 2018, respectively, accounting for around 35 % of total excreted N, with the remaining 65 % of N being excreted in animal houses. Overall, 21 % and 19 % of the N excreted during grazing is volatilized as NH₃ in 2010 and 2018, 700 respectively. As shown in Fig. 10 and Fig. A9, high total emissions are found in India, Pakistan and South America, and high percentage volatilization rates are found in France, Mexico, Spain, southern US, Africa, South Asia and the Middle East. Beef cattle is the largest emitter contributing over 50 % of the estimated emissions, whereas dairy cattle and buffalo are responsible for 30 % and 20 % of emissions, respectively. Buffaloes have the highest percentage volatilization rates, of more than 25 %, reflecting their predominant location in tropical climates, while beef and dairy generally have lower volatilization rates of less 705 than 20 %.

For year-round grazing, total excreted N from the grassland production system cattle is estimated at 23.8 Tg N yr$^{-1}$ in 2010 and 24.4 Tg N yr$^{-1}$ in 2018, with 18 % and 15 % of excreted N being lost through NH$_3$ emissions in each simulated year, respectively. The overall estimated percentage volatilization rate for year-round grazing of cattle is lower than that for seasonal grazing. Countries and regions with high seasonal grazing emissions also have high emissions from year-round grazing (e.g., Argentina, Brazil, India and Pakistan). Moreover, high emissions also occur in Mexico and US. Compared with seasonal grazing, the percentage volatilization rates of year-round grazing are generally smaller. High volatilization rates are found across the Africa, South Asia, the Middle East and part of Australia. Again, beef contributes over 50 % of total year-round grazing emissions, which is the largest emitter. Dairy contributes 40 % of emissions, while buffaloes result in less than 10 %. All types of cattle exhibit similar volatilization rates, with rates in 2018 being lower than 2010.

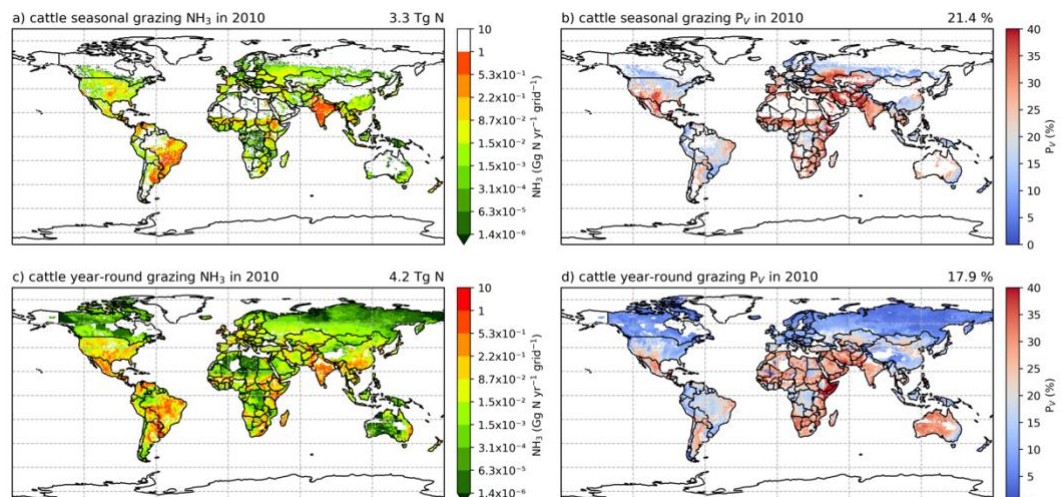

**Figure 10. Simulated (a) annual global NH$_3$ emissions (Gg N yr$^{-1}$) from cattle seasonal grazing in 2010. (b) Percentage of excreted N from cattle while grazing seasonally that volatilizes ($P_V$) as NH$_3$ in 2010. (c) Annual global NH$_3$ emissions (Gg N yr$^{-1}$) from cattle year-round grazing in 2010. (d) Percentage of excreted N from cattle while grazing year-round that volatilizes ($P_V$) as NH$_3$ in 2010. The resolution is 0.5º × 0.5 º.**

Sheep and goat grazing together resulted in an estimated 1.4 Tg N yr$^{-1}$ of NH$_3$ emissions in 2010 and 2018, according to simulations using AMCLIM. The mixed production systems contribute 0.6 Tg N yr$^{-1}$ of emissions, while the grassland production systems contribute 0.8 Tg N yr$^{-1}$. Contrary to cattle, around 65 % of N in excreta from mixed production system sheep and goat is deposited on pastures rather than in houses, with an estimated 25 % and 20 % of excreted N lost through NH$_3$ volatilization in 2010 and 2018, respectively. High emissions occur in India, Iran, NCP, Pakistan, Spain, Turkey, and several Sahel countries (Figs. 11a and 11c). The highest volatilization rates are found in southwestern Russia. India, Africa, and Europe also have high rates (Figs. 11b and 11d). Sheep contribute to over 60 % of total emissions, while goats contribute 40 %. The estimated volatilization rates for both livestock are similar.

For the grassland production system of sheep and goats, it is estimated that around 25 % and 20 % of excreted N volatilizes as $NH_3$ during year-round grazing in 2010 and 2018, respectively. As shown in Figs. 11 and Fig. A10, high emissions are found in southeastern Australia, northern China, eastern Africa, the Middle East, and South Asia. High volatilization rates occur in Australia, Mexico, part of the US, Africa, the Middle East, South Asia, and South America (Figs. 11b and 11d). Sheep are responsible for two-thirds of the estimated emissions, and the volatilization rates for sheep and goats are estimated at

around 20 %. It is notable that year-round grazing of sheep and goats generally results in similar volatilization rates to seasonal grazing, which is different from cattle grazing.

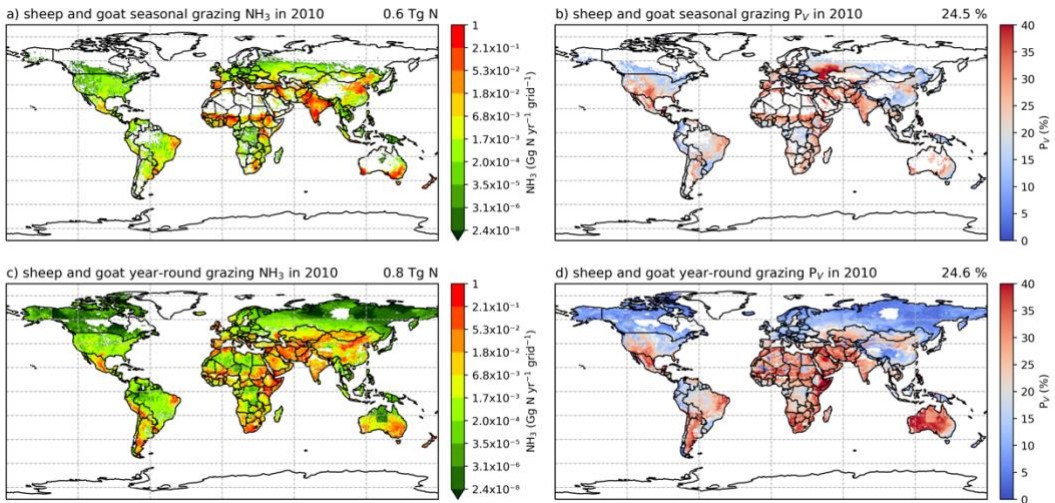

**Figure 11. The same as Figure 10, but for sheep and goats.**


     Emissions of $NH_3$ from the different grazing schemes estimated by AMCLIM–Land are summarised in Table 2. In both years, urine patches contribute the highest estimated $NH_3$ emissions and the highest volatilization rates. About 70 to 75 % of $NH_3$ emissions from grazing result from urine patches according to AMCLIM, while the remaining 25 to 30 % is from dung pat (a combination of dung only and mixed dung and urine). Within the dung pat scheme, around 3 % of excreted N volatilizes

as $NH_3$ from dung itself. By comparison, about 17 % N is lost as $NH_3$ from the mixture of dung and urine.

**Table 2. Total excreted N (Tg N yr$^{-1}$), $NH_3$ emissions (Tg N yr$^{-1}$) and volatilization rates (%) from each grazing scheme for ruminants.**

| Year | Scheme | Total excreted N while grazing (Tg N yr$^{-1}$) | $NH_3$ from grazing (Tg N yr$^{-1}$) | Average $P_V$ (%) |
|------|--------|--------------------------------------------------|---------------------------------------|-------------------|
| 2010 | urine patch | 24.66 | 6.78 | 27.5 |

| | | | | |
|---|---|---|---|---|
| | dung pat (dung only) | 8.61 | 0.25 | 2.9 |
| | dung pat (mixed) | 11.51 | 1.92 | 16.7 |
| | Total | 44.78 | 8.95 | 19.9 |
| 2018 | urine patch | 25.42 | 5.64 | 22.2 |
| | dung pat (dung only) | 8.87 | 0.29 | 3.3 |
| | dung pat (mixed) | 11.88 | 1.96 | 16.5 |
| | Total | 46.17 | 7.89 | 17.1 |

### 3.3.2 Comparison of grazing NH$_3$ emissions estimated using AMCLIM with observations

The simulated NH$_3$ volatilization rates from grazing by AMCLIM were compared with measurements, which mainly focuses on evaluation against experimental studies that measured NH$_3$ emissions from urine deposition, as NH$_3$ emissions mainly result from urine patches during grazing. Two types of observations were selected for the comparisons: real livestock grazing and urine application. One of the constraints of such studies is that they tend to report insufficient input data in the reported measurements to allow AMCLIM simulations on a detailed site basis (cf. Figs. 4 and 5). Therefore, simulated volatilization

rates were extracted from a large number of global experimental studies and compared with the measured percentage volatilization rates derived from these experimental studies, depending on geographical locations and time of the year.

Figure 12a shows the comparisons between modelled and measured $P_V$ for actual livestock grazing. Simulated $P_V$ for cattle grazing is comparable to the measurements at sites in the UK (Jarvis et al., 1989a; Ryden et al., 1987), Switzerland (Voglmeier et al., 2018), France (Bell et al., 2017) and New Zealand (Laubach et al., 2013). The annual mean volatilization rate (%

NH$_3$/total excreted N) of grazing in northern Europe estimated by AMCLIM (9.5 %) also agrees with Hutchings et al. (1996) (<10 %). However, large differences exist between the modelled and measured $P_V$ (% NH$_3$/urinary N) for cattle and sheep grazing in the UK (Jarvis et al., 1989b, 1991), as well as the volatilization rate (% NH$_3$/total excreted N) of cattle grazing in the Netherlands (Bussink, 1992), where AMCLIM largely overestimates the measured volatilization rates. These overestimations might be due to local management practices, and the fact that AMCLIM estimates gross emissions rates,

excluding possible canopy recapture, which is expected to be more significant in cool-wet climates, such as United Kingdom and the Netherlands. Bussink (1992) and Jarvis et al. (1989; 1991) measured NH$_3$ loss from grazed land with different levels of synthetic fertilizer inputs that varied between 210 kg N ha$^{-1}$ to 550 kg N ha$^{-1}$. The observed volatilization rates are normally very low (< 5%), while simulated volatilization rates are much higher (8 to 22 %). It remains unclear exactly how additional fertilizer affects the NH$_3$ volatilization from livestock excreta on grasslands (e.g., by increasing the N content of urine versus

by direct emission from vegetation).

By comparison, there is closer agreement between the volatilization rates estimated by AMCLIM and those measured for urine application than for real animal grazing (Fig.12a vs. Fig.12b). Figure 12b shows that the majority of the modelled $P_V$ is within a factor of 2 relative to the measured $P_V$ (FAC2 = 0.86); the correlation between the model and measurements was 0.47 (r=0.47). Specifically, $P_V$ estimated by AMCLIM is generally consistent with real livestock urine application experiments conducted at sites in Australia (Vallis et al., 1982, 1985), Ireland (Fischer et al., 2016) and New Zealand (Ball et al., 1979; Laubach et al., 2012; Sherlock and Goh, 1984), as well as two studies using artificial urine in Finland (Saarijärvi et al., 2006) and US (Frank and Zhang, 1997). In particular, AMCLIM captures a very high $P_V$ measured for cattle urine application in a tropical place in Australia (symbol "A" in Fig.12b). However, only very limited measurements have been taken under tropical climates, indicating a need for more experiments in hot regions.

It is worth mentioning when comparing with experiments carried out under dry soil conditions, the volatilization rates of urine application estimated by AMCLIM are either overestimated for three experiments in UK summer by Lockyer and Whitehead (1990), or underestimated for one experiment in New Zealand summer by Carran et al. (1982). Low $P_V$ measured by Lockyer and Whitehead (1990) in June and July at a UK site shows clear differences compared to other measurements of the same study (symbol "D"s in Fig.12b), which remained unclear to the original authors, and no clear explanations were provided (Lockyer and Whitehead, 1990). However, since AMCLIM was not applied at each site and was not driven by the same environmental and meteorological variables, the simulated $P_V$ is not distinguished between dry or wet soil conditions. Higher $P_V$ in dry soils (soil moisture close to wilting point) than wet soils (soil moisture close to field capacity) reported by Carran et al. (1982) might be related to the retention of urine in soils and slower drainage, as well as due to reduced foliar interactions that are expected in wetter situations.

There is less literature investigating $NH_3$ volatilization from dung than urine. In general, the $P_V$ of dung was found to vary between 1 to 5 % in Europe (Fischer et al., 2016; Whitehead, 1990), while Laubach et al. (2013) reported that 11 % of N in dung was lost through $NH_3$ emissions in an experiment in New Zealand. Meanwhile, it is broadly agreed that $NH_3$ emissions from grazing mainly come from urine deposition, which ranges from 87 to 96 % based on existing studies (Laubach et al., 2013; Saarijärvi et al., 2006). Simulations using AMCLIM suggest a lower contribution from the urine patch because the mixture of urine and dung in the dung pat scheme was also included in the model. The results from AMCLIM can be considered broadly consistent with experimental studies.

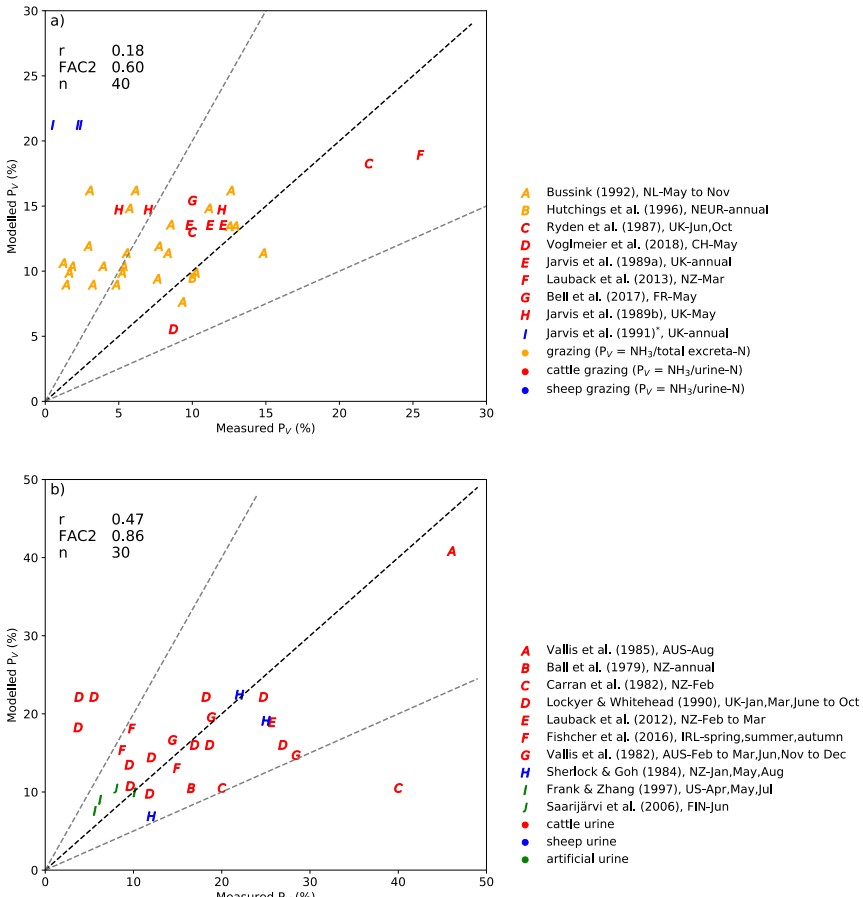

**Figure 12. Modelled percentage volatilization rates ($P_V$, %) compared with field measurements. Measurement data were from literature that studied real ruminant grazing (a) and ruminant urine application (b). Pearson's correlation coefficient (r), fraction of values within a factor of 2 (FAC2) and number of model-measurements comparisons (n) are presented at the top left corner. [*]In Jarvis et al. (1991), $P_V$ of the grazed land with 0 and 420 kg N ha[-1] fertilizer input and mixed grass/clover were 0.5 %, 2.2 % and 2.4 %, respectively.**

### 3.4 Nitrogen flows and NH₃ emissions of global livestock farming

Figure 13 summarizes the simulated N flows of global livestock farming for the reference year 2010 by AMCLIM, which are allocated to housing, manure management, and application to land, with a focus on $NH_3$ emissions. Other simulated nitrogen pathways include surface runoff, nitrification, leaching and diffusion to deeper soils, uptake by plants and amount left in soils. As specified in the description of soil processes in AMCLIM (Jiang et al., 2024), denitrification and emission of NO, $N_2O$ and $N_2$ are not explicitly included in this study. The flux of "nitrification" simulated in this study can be seen as a sum of both nitrified and denitrified N, with the amount of all relevant species (NO, $N_2O$ and $N_2$) being included, since the focus here on nitrification as a loss pathway of ammonium.

For pig farming, global total excreted N is estimated at 13.51 Tg N yr$^{-1}$ in 2010. All excreted N is allocated to housing, which resulted in NH$_3$ emissions of 2.37 Tg N yr$^{-1}$. A further 2.78 Tg N yr$^{-1}$ is estimated to be lost because of manure burning (0.50 Tg N yr$^{-1}$) and unmanaged manure (2.27 Tg N yr$^{-1}$). The remaining 8.36 Tg N yr$^{-1}$ undergoes management and leads to 1.01 Tg N yr$^{-1}$ of NH$_3$ emission. A small part (0.33 Tg N yr$^{-1}$) is either washed off, nitrified, or left in lagoon systems, while 1.32 Tg N yr$^{-1}$ is left on land without being stored. Subsequently, 5.71 Tg N yr$^{-1}$ from storage are applied to land, which results in an estimated 1.94 Tg N yr$^{-1}$ of NH$_3$ emission, 2.87 Tg N yr$^{-1}$ entering soils and plants and 0.90 Tg N yr$^{-1}$ being depleted by other processes (e.g., runoff, nitrification, leaching and diffusion to deep soils). Manure left on land is assumed to be completely incorporated into soils or used by plants and is not further simulated by AMCLIM. Nitrified N was also not further simulated by the model.

Global total excreted N from chicken is estimated at 11.22 Tg N yr$^{-1}$ in 2010, which resulted in NH$_3$ emissions of 3.10, 0.77 and 0.87 Tg N yr$^{-1}$ from housing, manure management and application to land, respectively. Only 0.04 Tg N yr$^{-1}$ is estimated to be burned as fuel, and most manure N (8.07 Tg N yr$^{-1}$) is managed. During chicken manure management, Nitrification and N loss associated with runoff are tiny (0.11 Tg N yr$^{-1}$). A large fraction of manure N (5.04 Tg N yr$^{-1}$), which is mainly from deep litter system broiler chicken, is left on land rather than being stored. By comparison, 2.15 Tg N yr$^{-1}$ of stored manure N that is mainly from layer chicken is applied to land. In addition to NH$_3$ emission, estimated at 0.99 Tg N yr$^{-1}$ entering soils and plants, the remaining 0.29 Tg N yr$^{-1}$ is nitrified or lost via runoff, leaching and diffusion to deep soils.

For ruminants, the global total excreted N from cattle, sheep, and goats is estimated to be 74.84 Tg N yr$^{-1}$. About 40 % of N (30.06 Tg N yr$^{-1}$) is excreted to the housing systems, while 60 % (44.78 Tg N yr$^{-1}$) is excreted to grazing land. Ruminant housing results in an estimated 3.75 Tg N yr$^{-1}$ of NH$_3$ emission, while 3.53 Tg N yr$^{-1}$ of manure N is used as fuel. Manure management results in an estimated 3.89 Tg N yr$^{-1}$ of NH$_3$ emission, accounting for 17 % of total managed manure N (22.79 Tg N yr$^{-1}$). During manure management, nitrogen left on land without being stored is estimated at 8.28 Tg N yr$^{-1}$, while runoff and nitrification together account for 1.62 Tg N yr$^{-1}$. Nitrogen that is introduced to land consists of two parts: 9.01 Tg N yr$^{-1}$ from manure storage by land application (17%) and 44.78 from grazing (83%), which together result in an estimated 12.20 Tg N yr$^{-1}$ of NH$_3$ emission (3.25 Tg yr$^{-1}$ from manure application to land and 8.95 Tg yr$^{-1}$ from grazing, amounting to $P_V$ of 36 % and 20 %, respectively). Meanwhile, 26.07 Tg N yr$^{-1}$ enters soils used by plants, and 15.55 Tg N yr$^{-1}$ of N undergoes other processes (e.g., runoff, nitrification, leaching, and diffusion to deep soils).

Overall, NH$_3$ emissions from global livestock farming are estimated to be 29.9 Tg N yr$^{-1}$, with high NH$_3$ emissions found in China, India, the US and European countries, accounting for 30 % of total N from livestock excreta (Fig. 14). Cattle (including buffaloes) is the largest emitter group among livestock, contributing over 60 % of livestock NH$_3$ emissions). Both pigs and poultry result in more than 15 % of livestock NH$_3$ emissions, while sheep and goats are responsible for the remaining 7 %.

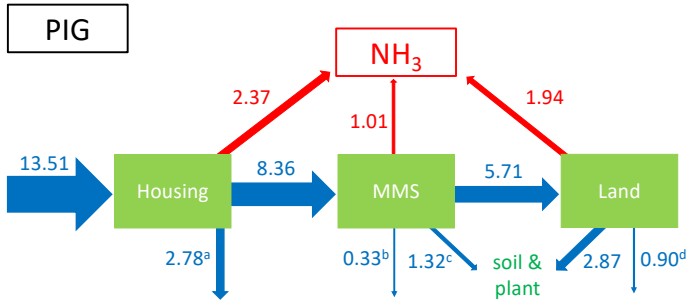

PIG

<sup>a</sup>manure used as fuel (0.50) unmanaged manure (2.27)
<sup>b</sup>runoff (0.02) nitrification (0.27) lagoon (0.04)
<sup>c</sup>this 1.32 represents the amount of manure N left on land (not stored) and is not further simulated; NH₃ emission counted as MMS emissions
<sup>d</sup>runoff (0.19) nitrification (0.48) leaching & diffusion to deep soil (0.23)

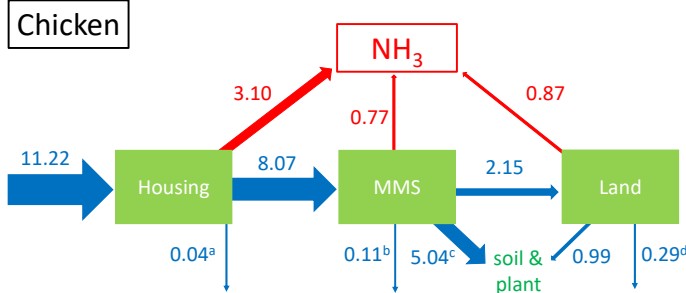

Chicken

<sup>a</sup>manure used as fuel (0.04)
<sup>b</sup>runoff (0.01) nitrification (0.10)
<sup>c</sup>this 5.04 represents the amount of manure N left on land (not stored) and is not further simulated; NH₃ emission counted as MMS emissions
<sup>d</sup>runoff (0.08) nitrification (0.13) leaching & diffusion to deep soil (0.08)

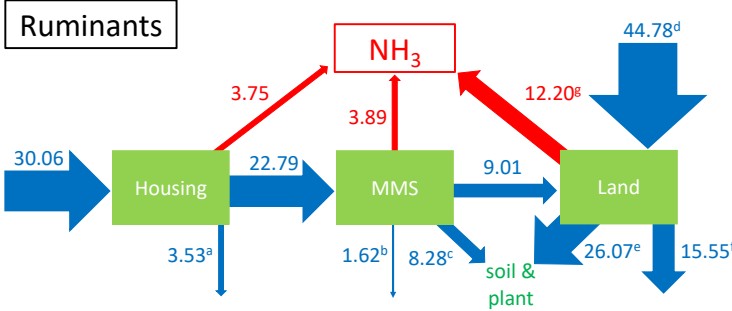

Ruminants

<sup>a</sup>manure used as fuel (3.53)
<sup>b</sup>runoff (0.22) nitrification (1.44) lagoon (<0.01)
<sup>c</sup>this 8.28 represents the amount of manure N left on land (not stored) and is not further simulated; NH₃ emission counted as MMS emissions
<sup>d</sup>excreted N deposited on pastures while grazing
<sup>e</sup>manure application (4.43) grazing (21.64)
<sup>f</sup>runoff (1.13) nitrification (1.47) leaching & diffusion to deep soil (12.95)
<sup>g</sup>manure application (3.25) grazing (8.95)

**Figure 13. Nitrogen budget of global livestock farming estimated by AMCLIM for the year 2010. Activities include housing, manure management systems (MMS), application to land and grazing (for ruminants only). Dark blue arrows are liquid and solid N flows. Red arrows represent gaseous NH₃ emissions. All numbers have the unit of Tg N yr⁻¹. Size of the arrows is proportional to the flux.**

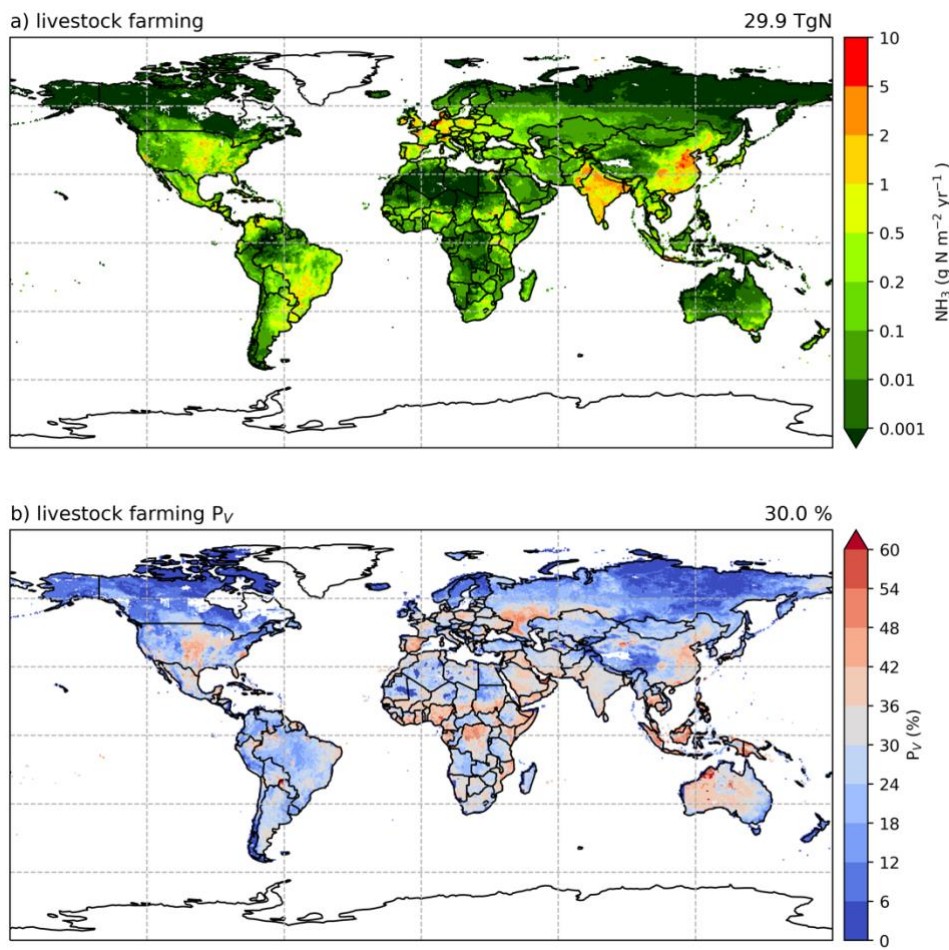

**Figure 14. Simulated (a) annual global NH₃ emissions (gN m⁻² yr⁻¹) from livestock farming (including housing, manure management, land application of manure and grazing) in 2010. (b) Percentage of total livestock excreted N that volatilizes ($P_V$) as NH₃ in 2010. The resolution is 0.5° × 0.5°.**

As shown in Fig. 15, NH₃ emissions and volatilization rates vary across different geographical regions and between the two simulated years, i.e., 2010 and 2018. The highest NH₃ emissions from livestock agriculture are estimated to occur in East and South Asia. In general, the volatilization rates for livestock are lower in 2018 than 2010, except for poultry. This is because a large fraction of poultry which are broiler and layer production systems are assumed in the model to be kept in houses with controlled temperature and ventilation, so the $P_V$ rates were less impacted by the differences in environmental conditions between years. By comparison, the $P_V$ rates from sheep and goats show the largest inter-annual variability among all livestock groups because sheep and goats typically graze outside and are housed in naturally ventilated barns. As a result, NH₃ volatilization from sheep and goats is more dependent on the environmental conditions than poultry. The differences in housing

and grazing management also explain why $P_V$ rates from pigs showed the second smallest difference and cattle showed the second largest difference between the two simulated years.

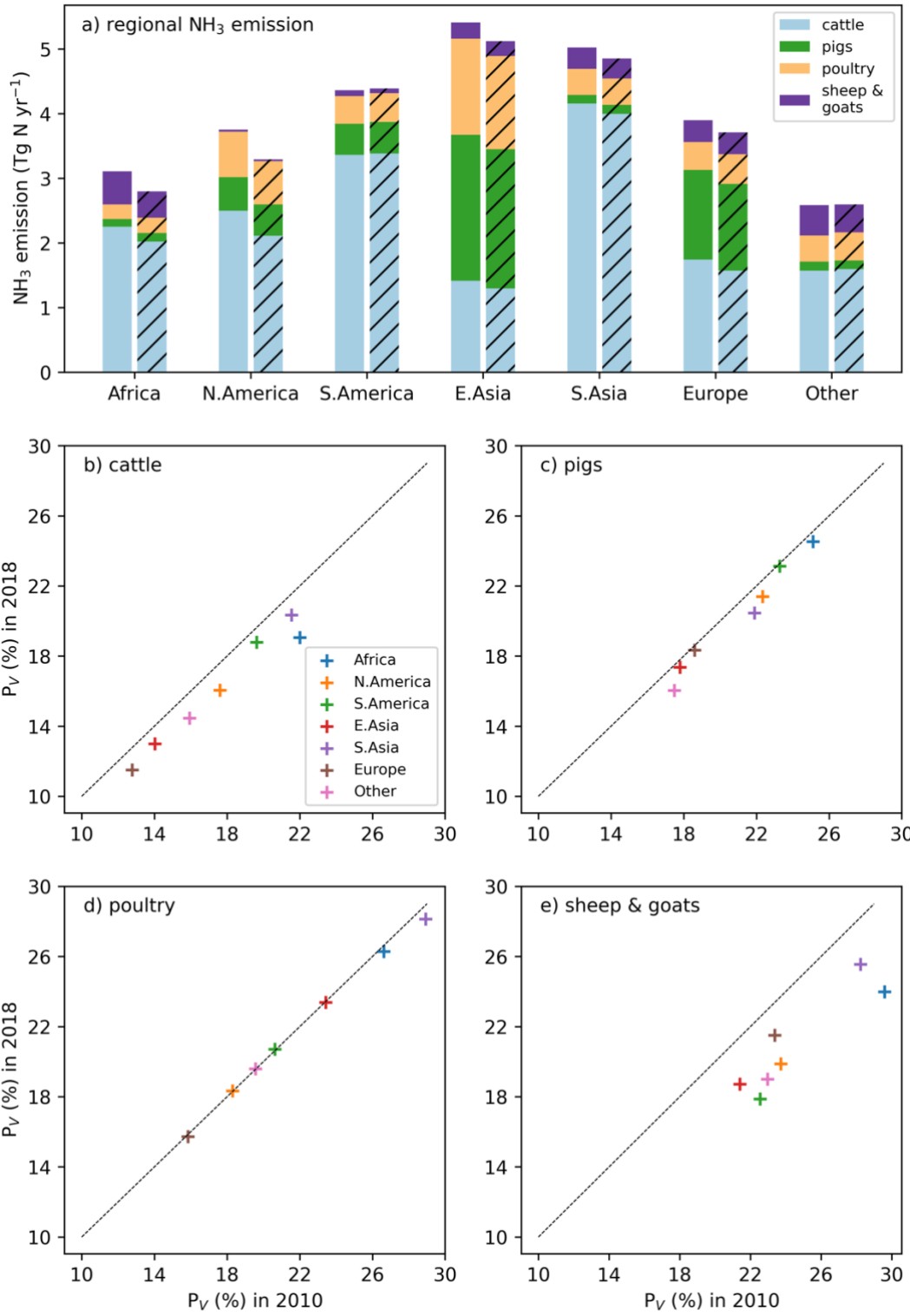

**Figure 15. Estimated (a) NH$_3$ emissions from livestock farming in seven geographical regions for the two studied years, 2010 (left) and 2018 (right, hashed), and comparisons of volatilization rates $P_V$ between the two years for (b) cattle, (c) pigs, (d) poultry and (e) sheep and goats.**

## 4 Discussion

### 4.1 Contributions to global emissions and evaluation in relation to measurements

Ammonia emissions can occur from various activities in livestock agricultural systems, from housing to subsequent manure storage and the ultimate spreading of manure. The three modules in the AMCLIM model all contribute to simulating a substantial share of total NH$_3$ emissions from livestock farming. Summing for all livestock types, the amounts are 9.22 Tg N yr$^{-1}$ from AMCLIM-Housing, 5.67 Tg yr$^{-1}$ ACLIM-MMS and 15.01 Tg yr$^{-1}$ from AMCLIM-Land, contributing to a global total of 29.9 Tg N yr$^{-1}$ of NH$_3$ emission from global livestock. The relative shares (31%, 19% and 50%, respectively), indicate the need for all three AMCLIM modules.

AMCLIM has been developed based on prior testing for chicken houses (Jiang et al., 2021), with the model principles building on the earlier simulation approach of the GUANO model (Generalisation of Uric Acid Nitrogen emissions), a process-based model designed for simulating and predicting NH$_3$ emissions from a source of seabird-derived uric acid, which has been tested in relation to measurements from seabird colonies (Riddick et al., 2017). Extension and further development of AMCLIM–Housing has allowed it to be applied at the site scale to simulate housing emissions and provided reasonable estimates for pigs, layers and dairy, which are all in close agreement with the measurements. This was only possible because of the valuable, detailed and well-documented measurement datasets reported by Lim et al. (2010a) and Wang et al. (2010).

The NH$_3$ emissions from livestock housing are found to be strongly related to environment of the animal houses, where emissions increase as inside temperature and ventilation increase. Meanwhile, management practices also play an important role in affecting emissions. For example, emissions vary with the growing cycles in the pig farm at site IN3B, and the removal of the excreta causes a short "stoppage" of NH$_3$ emission (see Fig. 4). Unlike the pigs and layer chicken, the indoor conditions of the dairy barn are closer to the ambient environment, which is generally cooler throughout the year. As a result, the annual average volatilization rates are lower for dairy barns compared with pig and layer houses.

In contrast to the simulations of NH$_3$ emissions from housing, there are many emission datasets for grazing livestock and for urine application to land, but these lack the details needed to allow specific site application of the model, as done for housing. Accordingly, the comparison with measurements in Fig. 12, focuses on a multi-site comparison for a wide range of published datasets globally. While this comparison is useful, it also highlights the great value of publishing carefully documented detailed time-resolved datasets as reported by Lim et al. (2010a) and Wang et al. (2010). Future work should look to acquire further quality assured measurement datasets for additional testing.

## 4.2 Comparison of AMCLIM with previous studies and emission factor approaches

Estimated $NH_3$ emissions in this study are generally in line with the analysis by Uwizeye et al. (2020). Global total emissions from livestock agriculture were estimated to be 29.9 Tg N $yr^{-1}$ according to AMCLIM's simulations, while Uwizeye et al. (2020) reported that global livestock supply chain contributed 26.4 Tg N $yr^{-1}$. Emissions from animal production related to housing and manure management are higher from AMCLIM (14.9 Tg N $yr^{-1}$) than those from Uwizeye et al. (2020) (11.6 Tg N $yr^{-1}$), while estimated $NH_3$ emissions from land application of manure to cropland and grasslands (including grazing) by AMCLIM (15.0 Tg N $yr^{-1}$) are comparable to the emissions from feed production estimated by Uwizeye et al. (2020) (14.8 Tg N $yr^{-1}$). Sutton et al. (2013) also reported a similar value of 15 Tg N $yr^{-1}$ $NH_3$ emissions from crops and grasses used in livestock.

Although AMCLIM is a dynamical process-based model that is not dependent on emission factors (EFs), results of its simulations can be averaged as a basis to compare with existing EFs from other studies and emission inventories. Livestock-specific EFs $NH_3$ derived from ACMLIM simulations and comparisons with EFs from the reviewed range of literature reported by Yang et al. (2023) are summarised in Table 3. On the global scale, the EFs derived from AMCLIM are generally comparable to values Yang et al. (2023). However, AMCLIM predicted higher EFs for buffaloes and lower EFs for dairy cattle than the literature reviewed by Yang et al. (2023). Although this broad comparability may be considered encouraging, one of the key reasons for developing AMCLIM is to illustrate how climatic variation affects ammonia emission. Such effects are not well treated by average EFs, therefore, it is not appropriate to search for exact agreement, which cannot be achieved by the EF approach.

**Table 3. Simulated animal $NH_3$ emission factors (EFs) (kg N head$^{-1}$ yr$^{-1}$) for livestock derived from the global simulations of AMCLIM compared with Yang et al. (2023), who summarize the range of EFs from literature. The AMCLIM values in the first row are the global mean EF, and values in the brackets represent the 10th and 90th percentile of the 0.5° x 0.5° resolution values, respectively.**

| Study | Ruminants EFs (kg N head$^{-1}$ yr$^{-1}$) | | | | | | |
|---|---|---|---|---|---|---|---|
| | Pigs | Chicken | Beef | Buffaloes | Dairy | Goat | Sheep |
| AMCLIM | 5.5 (2.8 – 9.4) | 0.24 (0.11 – 0.29) | 9.7 (2.5 – 15.9) | 11.8 (3.7 – 15.5) | 11.6 (2.1 – 19.1) | 0.8 (0.2 – 1.7) | 1.2 (0.2 – 2.0) |
| Yang et al. (2023) | 1.2 – 7.2 | 0.08 – 0.37 | 3.0 – 14.3 | 2.8 – 8.7 | 14.5 – 21.8 | 0.6 – 5.0 | 0.6 – 2.5 |

Table 4 compares the simulated EF (expressed as % of TAN) by AMCLIM with EFs reported by EMEP/EEA (2019) and Sommer et al. (2019). The table summarizes both global mean and EU mean EFs, with range of EFs between the 10th and 90th percentile of both spatial scales. As shown in Table 4, estimated EFs for livestock housing and manure application to land by

this study are generally comparable to the values from literature and reports. The EFs of manure storage derived from AMCLIM simulation are often lower than EMEP/EEA (2019) and Sommer et al. (2019), while the grazing EFs are higher than previous studies. As noted in Sect.3.3.2, the simulated volatilization rates of AMCLIM may be higher than other studies, since AMCLIM estimates gross emission, without accounting for canopy recapture, that can be significant for grassland contexts, especially in wet climates (Massad et al., 2010; Sutton et al., 2013). For housing, AMCLIM did not differentiate slurry from
solid manure for ruminants. The largest difference is from broiler housing, the EF of which is as two times as reported. Although estimated EFs for manure storage in this study are low, the AMCLIM model showed that slurry storage typically had lower EFs than solid manure, which is consistent with EMEP/EEA. Both this study and EMEP/EEA (2019) agreed that the highest EFs are from manure application among the four practices. However, AMCLIM also takes other organic N other than urea into account. The organic N from dung can be a slow but significant source to the TAN pool as a result of
mineralization. In all AMCLIM simulations, $NH_3$ emissions volatilized from the TAN pool were not differentiated from urea N or other organic N. In order to include this effect for the comparison, a set of correction factors were applied to the total N to obtain the amount of TAN. It is assumed that the TAN amount is the sum of urea fraction and half of the organic N fraction (the "available organic nitrogen compound" assumed in the model, which accounts for 50 % of organic nitrogen other than urea; see Supplementary Materials Sect.S2) of livestock excreta. As a consequence, estimated EFs are more robust for housing
and grazing in terms of $NH_3$ lost as percentage of TAN compared with manure storage and application to field because the quantities of different N forms in latter stages are more uncertain and difficult to estimate. On the one hand, it has been argued that the EF method is not ideal for calculating $NH_3$ emissions due to its limitations in excluding the climate-dependence of $NH_3$. On the other hand, the EFs given as the percentage of TAN do not explicitly include the organic N input from manure, which might result in either overestimation or underestimation in $NH_3$ emissions, depending on the activity data used.


**Table 4. Averaged simulated $NH_3$ emission factors expressed as percentage of TAN for livestock derived from the global simulations of AMCLIM compared with EMEP/EEA and Sommer et al. (2019). *Global mean EF. **Mean EF for Europe. Values in the brackets represent the 10[th] and 90[th] percentile of the 0.5°×0.5° resolution values, respectively.**

|  | Livestock | AMCLIM | EMEP/EEA 2019 & Sommer et al. (2019) |
|---|---|---|---|
| Housing | Dairy | 19[*,b](8 − 36); 13[**,b](8 − 20) | 24[a], 8[b], 9[a,c], 19[b,c] |
|  | Non-dairy | 15[*,b](5 − 29); 8[**,b](5 − 13) | 24[a], 8[b] |
|  | Buffalo | 24[*,b](7 − 39); 21[**,b](8 − 25) | 20[b] |
|  | Sheep & Goat | 28[*,b](12 − 58); 25[**,b](12 − 36) | 22[b] |

| | | | |
|---|---|---|---|
| | Pigs | $40^{*,a}(21-68)$; $34^{**,a}(25-48)$ <br> $15^{*,b}(5-31)$; $13^{**,b}(7-22)$ | $27^{a,d}$, $23^{b,d}$, $24^{a,e}$, $35^{b,e}$ |
| | Chicken | $45^{*,b,f}(32-58)$; $41^{**,b,f}(34-44)$ <br> $36^{*,b,g}(19-55)$; $24^{**,b,g}(13-39)$ | $41^{a,f}$, $20^{b,f}$, $21^{b,g}$ |
| Storage | Dairy | $12^{*,a}(4-26)$, $8^{**,a}(5-17)$ <br> $25^{*,b}(5-55)$, $15^{**,b}(6-26)$ | $25^{a}$, $32^{b}$ |
| | Non-dairy | $6^{*,a}(3-47)$, $5^{**,a}(3-10)$ <br> $24^{*,b}(3-52)$, $13^{**,b}(4-20)$ | $25^{a}$, $32^{b}$ |
| | Buffalo | $8^{*,a}(4-16)$, $8^{**,a}(6-13)$ <br> $39^{*,b}(3-58)$, $31^{**,b}(4-25)$ | $17^{b}$ |
| | Sheep & Goat | $23^{*,a}(18-20)$, $23^{**,a}(28-28)$ <br> $36^{*,b}(15-47)$, $39^{**,b}(16-53)$ | $32^{b}$, $28^{b}$ |
| | Pigs | $14^{*,a}(3-40)$, $9^{**,a}(4-21)$ <br> $27^{*,b}(4-66)$, $15^{**,b}(8-36)$ | $11^{a}$, $29^{b}$ |
| | Chicken | $27^{*,b,f}(4-64)$; $19^{**,b,f}(6-44)$ <br> $8^{*,b,g}(<1-17)$; $5^{**,b,g}(2-10)$ | $14^{a,f}$, $8^{b,f}$, $30^{b,g}$ |
| Manure application | Dairy | $51^{*,a}(31-75)$, $47^{**,a}(25-63)$ <br> $56^{*,b}(29-69)$, $51^{**,b}(27-64)$ | $55^{a}$, $68^{b}$ |

|  |  |  |  |
|---|---|---|---|
|  | Non-dairy | $48^{*,a}(25-70)$, $46^{**,a}(24-60)$ $50^{*,b}(25-65)$, $47^{**,b}(25-61)$ | $55^a$, $68^b$ |
|  | Buffalo | $43^{*,a}(28-66)$, $43^{**,a}(36-66)$ $48^{*,b}(31-67)$, $47^{**,b}(39-68)$ | $55^b$ |
|  | Sheep & Goat | $43^{*,a}(32-55)$, $43^{**,a}(34-55)$ $49^{*,b}(21-62)$, $41^{**,b}(25-55)$ | $90^b$ |
|  | Pigs | $51^{*,a}(17-84)$, $43^{**,a}(20-67)$ $47^{*,b}(16-77)$, $45^{**,b}(21-63)$ | $40^{a,d}$, $45^b$, $29^{a,e}$ |
|  | Chicken | $61^{*,b,f}(19-95)$; $50^{**,b,f}(23-78)$ $50^{*,b,g}(17-93)$; $49^{**,b,g}(18-63)$ | $69^{a,f}$, $45^{b,f}$, $38^{b,g}$ |
| Grazing | Dairy | $25^{*}(2-44)$; $16^{**}(<1-45)$ | $14^{EMEP/EEA}$, $9^{Sommer}$ |
|  | Non-dairy | $26^{*}(2-44)$; $17^{**}(<1-43)$ | $14^{EMEP/EEA}$, $9^{Sommer}$ |
|  | Buffalo | $37^{*}(2-42)$; $36^{**}(7-46)$ | $14^{EMEP/EEA}$ |
|  | Sheep & Goat | $33^{*}(<1-46)$; $31^{**}(<1-45)$ | $9^{EMEP/EEA}$ |

[a]slurry

[b]solid manure

[c]tied housing

[d]finishing pigs (8-110 kg)

[e]sows and piglets (up to 8 kg)

[f]laying hens

[g]broilers

## 4.3 Spatial distribution of simulated ammonia emissions in relation to climate and management

The spatial distributions of both emissions and the percentage volatilization rates simulated by AMCLIM, as shown in Figs. 6 to 11 and Fig. 14 demonstrate substantial variations. High values of total emissions (expressed as kg N per grid yr$^{-1}$) primarily

coincide with high animal populations in countries or regions with intensive livestock farming, such as China, India, US and Europe.

The volatilization rates (expressed as the percentage volatilized, $P_V$) differ across the globe due to a combined effect of environmental conditions and management practices. High simulated volatilization rates of chicken housing are found in the tropical regions along the equator, indicating that hot and humid conditions tend to cause larger emissions. Among cattle and

buffaloes, the overall simulated volatilization rates for buffaloes are higher than other types of cattle. This is because buffaloes are predominantly reared in hot regions such as southern China, South Asia and southeast Asia compared with other cattle, which are widely distributed across the globe, resulting in higher $P_V$ for buffaloes due to generally hotter conditions. Also, the estimated volatilization rates for sheep and goat farming are higher than those of cattle farming, which is partly due to a higher N concentration in sheep and goat's urine compared with cattle. Another reason is that sheep and goat are more "concentrated"

in the Middle East and South Asia where they tend to have higher volatilization rates due to warmer climates. In addition to temperature, soil pH plays an important role in NH$_3$ volatilization. As pointed out by Jiang et al. (2024), simulated high $P_V$ values have been found in regions with high soil pH, such as the western US, Namibia, Mongolia and part of northern China.

Various management practices can lead to very different volatilization rates. For housing, industrial pigs show higher volatilization compared to intermediate and backyard pigs because the industrial pigs are kept in buildings with heating systems

and excreta are kept longer in the houses as in-situ storage is available. Moreover, the pits for manure storage provides an additional emitting surface of NH$_3$. The housing density assumed in AMCLIM is another factor that affects the volatilization rates. The volatilization rates of feedlot cattle housing are the second lowest among ruminants. This is partly because the feedlot cattle had the highest stocking density in the model. Increasing the stocking density results in a smaller source area for NH$_3$ emission, which leads to lower emissions.

For manure management, especially in warm climates, manure left on land without much management is identified to result in much higher NH$_3$ emissions than manure that is stored either as liquid or solid manure, leading to larger $P_V$ values. Such practice is common in Africa and some countries in South Asia like India and Myanmar, and these regions have hot climate (as reflected in high $P_V$ values). Conversely, manure storage under cover greatly reduces NH$_3$ emissions (Bittman et al., 2014). Although the effect of covering stored manure has not been the focus of the present study, the process-based nature of

AMCLIM would lend itself to a future examination of such effects.

## 4.4 Comparison of ammonia emissions and volatilization rates in 2010 and 2018

The AMCLIM simulations were calculated for two recent years (for which meteorological data were available) in order to illustrate how annual weather differences could influence $NH_3$ emissions globally. Of the two years studied, temperatures and water availability conditions changed across the globe. In general, regions with higher temperature in 2018 than 2010 had less precipitation and drier soil conditions (Fig. A11). As summarized in Table 5, the overall estimated volatilization rates of livestock farming for the year 2010 are found to be very similar to the values for 2018. Specifically, $NH_3$ emissions from housing and manure management for pig and chicken agriculture show small differences between 2010 and 2018, with slightly higher $P_V$ occurring in 2018 compared to 2010 (reflecting warmer conditions in 2018).

Such relatively "stable" $P_V$ rates for housing and manure storage are possibly due to largely controlled indoor environments of animal houses compared with natural conditions (as shown in Figs. A12). Enclosed animal houses have their own regulated temperature inside, and naturally ventilated barns are not as windy as outside and the floor temperature of these barns are less varied than air temperature. For manure management, although the global average $P_V$ values between the two simulated years are similar, the changes in $P_V$ are site-specific. As shown in Figs. A12 and A13, the differences in $P_V$ of manure management of chicken and pigs between 2010 and 2018 show stronger spatial variations than those of housing (as shown in Figs. A12 and A13), so the two very similar annual volatilization rates can be a result of global averaging of various manure management systems.

Compared with housing and manure management, emissions from land application of manure vary by a larger amount between the two years for all livestock groups (Table 5; Figs. A6 to A10; Figs. A12 to A17). In this case, 2010 shows both higher total emissions and percentage volatilization rates than 2018 (reflecting that 2018 was wetter than 2010). Such inter-annual differences in $NH_3$ emissions from manure application to land are found to be consistent with AMCLIM simulations for synthetic fertilizer application, as discussed in Jiang et al. (2024). The relevant processes that govern the $NH_3$ emissions from land application are dependent on naturally varying environmental conditions, while there are more N pathways, such as runoff, drainage and diffusion. As more processes are involved under natural conditions, $NH_3$ emissions may show larger variations. The simulations by Jiang et al. (2024) for synthetic fertilizer application indicate that the lower volatilization rates in 2018 than 2010 can be attributed to larger leaching and diffusive fluxes in 2018 (reflecting the wetter conditions in 2018 than 2010), which depletes the soil N and leading to smaller $NH_3$ emissions. The different $P_V$ of manure application to land between the two years shown in Table 5 and Figs. A12 to A17 may result from the same reason. These differences are evident from the comparison for 2010 vs 2018 for different livestock types, reflecting these differences in source category contributions to the emissions from each livestock type (Fig. 15). Finally, Fig. 15 shows that the largest relative difference between emissions for 2010 and 2018 applies to sheep and goats, with lower $P_V$ for in 2018. This is because $NH_3$ emissions from sheep and goats are dominated by grazing, where the wetter conditions in 2018 had a larger effect than the warmer temperatures in 2018 (Table 5).

**Table 5. Annual mean volatilization rates of livestock housing, manure management, land application of manure and grazing in years 2010 and 2018. Annual mean temperature, soil water content and subsurface percolation flux for locations where NH₃ emissions occur and where these three activities took place in 2010 and 2018.**

| | Activity | Year | |
|---|---|---|---|
| | | 2010 | 2018 |
| $P_V$ (%) | Housing | 16.8 | 16.9 |
| | Manure management | 14.4 | 14.6 |
| | Manure app | 35.9 | 32.2 |
| | Grazing | 19.9 | 17.0 |
| Meteorological and environmental variable | | | |
| T (°C) | Manure app | 12.2 | 12.4 |
| | Grazing | 10.9 | 11.0 |
| Soil water (m³ m⁻³) | Manure app | 0.25 | 0.26 |
| | Grazing | 0.23 | 0.26 |
| Subsurface percolation flux (×10⁻² mm d⁻¹) | Manure app | 2.0 | 2.8 |
| | Grazing | 1.8 | 2.7 |

## 4.5 Global chicken farming: comparison with the previous version of AMCLIM

As described in the Methods (Sect.2.6), several further developments are included in the AMCLIM simulations for ammonia emission than in the earlier 'Poultry Model' previously reported by Jiang et al. (2021). With the improvements and modifications, the current AMCLIM model provides very similar estimates of the housing simulations at the site scale (see Sect.S7). This can be explained by the various process parametrizations having that have opposite effects. The inclusion of other organic forms of N gradually expands the TAN pool, which leads to more N that is available for NH₃ emissions. Conversely, the adsorption of TAN on manure solids and more moisture in the excreta decrease the aqueous TAN concentration so compensating this effect.

For the global simulations, NH₃ emissions from chicken farming are estimated at 4.8 Tg N yr⁻¹ in 2010 estimated by the current AMCLIM simulations, which is about 13 % less than those from the Poultry Model's estimation of 5.5 Tg N yr⁻¹. The

relative contribution to the total emissions shifts from land application of manure to housing, which is largely due to the fact that emissions from backyard chicken are counted as housing emissions in the current AMCLIM version, rather than as part of land application emissions as in the earlier Poultry Model.

Housing emissions from broilers are comparable between AMCLIM and the Poultry Model, while AMCLIM estimates higher housing emissions from layers compared with the earlier model. Lower land application emissions were estimated by the current AMCLIM model, which is partly because: 1) less N is applied to land, 2) more N pathways that are included act as competing fluxes to volatilization could decrease the emission, 3) the adsorption of TAN on soil solids leads to lower emission potential.

## 4.6 Ruminant grazing

Grazing is an additional component of the simulations for ruminant farming compared with pigs and poultry. The estimated $NH_3$ emissions from grazing are 9.0 and 7.9 Tg N $yr^{-1}$, for 2010 and 2018, respectively, accounting for around 19 and 16 % of excreted N from ruminants while grazing. Emissions related to grazing exhibit the largest annual difference between the two simulated years compared with other activities. The total grazing emissions estimated by AMCLIM are lower than the 12 Tg N $yr^{-1}$ suggested by the FANv2 model (Vira et al, 2020a), but the volatilization rates are comparable to the 18 % reported by FANv2. The differences in emissions between AMCLIM and FANv2 are partly due to the different estimates of excreted N on pastures. In general, excreted N on pastures during grazing results in lower simulated volatilization loss of $NH_3$ in AMCLIM compared with manure application.

For the "mixed" production system, about 65 % of N in cattle excreta is excreted in animal houses, compared with less than 40 % for sheep and goat, based on the GLEAM MMS data. The volatilization loss of excreted N during seasonal grazing of cattle is around 20 %, which is similar to the value for sheep and goats. It should be noted that the regional variations in the volatilization rates of year-round sheep and goat grazing are larger than the seasonal grazing (Fig. 11 b, d). By comparison, the regional variations in year-round grazing of cattle are similar to the seasonal grazing (Fig.10 b, d). As a result, with the "grassland" production system (year-round grazing) being more widely spread across the globe especially in temperate and cold regions than the "mixed" production system, year-round grazing of cattle shows lower volatilization rates compared with seasonal grazing, while there is not much difference between year-round and seasonal grazing for sheep and goat.

It is evident that the urine patch scheme in the grazing simulations results in much higher $NH_3$ emission and higher volatilization rate compared with the dung pat scheme (Fig. 3). Urea in urine deposited on pastures is readily able to hydrolyse to TAN, which can lead to higher emission than dung itself due to the slower decomposition of organic forms of N in dung. Existing experimental studies have reported that $NH_3$ loss accounts for 0.5 % to up to 46 % of urinary N, while AMCLIM predicted 5.5 to 41 % (Fig. 12). The differences can be caused by different environmental and meteorological conditions such as temperature, soil moisture, precipitation and soil texture, between the year when experiments were conducted and the modelled year 2010. It also remains unclear why several experiments showed very small volatilization rates, which were not clearly explained by experimentalists, and needs further investigations (see Lockyer and Whitehead, 1990; Jarvis et al., 1991).

Overall, estimated volatilization rates by AMCLIM are broadly consistent with measurements (Fig. 12), especially in warm regions such as Australia and New Zealand, but overestimates compared with some measurements from wetter climates, such as in the UK and Netherlands. The differences indicate that the infiltration and drainage (also diffusion) in AMCLIM might not be sufficiently representative. In addition, it should be noted that the present version of AMCLIM does not include a

1070 vegetation layer, and therefore represents gross emissions from the land surface. In cool wet climates bi-directional $NH_3$ exchange can occur from grazing land, and the low percentage emission of ammonia reported by Jarvis et al. (1989) can be considered as net fluxes including this effect. Considering such bi-directional interactions is to be treated as part of future development of AMCLIM.

By comparison to urine, dung contributes less $NH_3$ emissions resulting from the lower volatilization rates from dung.

However, the mixture of dung and urine scheme implies that urine deposited on dung can also result in considerable $NH_3$ emissions, which is due to the slow infiltration of urine to the soil underneath as dung partly retains the liquid. A similar example is surface application of manure to land which can cause large $NH_3$ emissions as a percentage of the N applied, highlighting the need for immediate incorporation of manure to reduce emissions.

### 4.7 Uncertainty and limitations

Uncertainty in $NH_3$ emissions from livestock farming simulations arises from multiple sources, as illustrated by the sensitivity analysis (Table 1). Here it was shown that the largest overarching uncertainty in the model (relevant for all modules) is the pH at an emitting surface. In practice, surface pH changes dynamically, also in response to urea and uric acid hydrolysis, since production of $NH_3$ increases pH (Bittman et al., 2014). Although it is possible to simulate surface pH dynamically, as shown by Móring et al. (2016) for grazing, this is computationally expensive and brings its own uncertainties, make it less appealing

for application in global simulations. In the present study, empirical pH values were used based on Chantigny et al (2004) and Móring et al (2016). But in practice, estimating a bulk surface pH of the emitting solution can also be considered as a potential model tuning parameter for simulations in relation to measured $NH_3$ emissions.

In addition to the effect of pH (Table 1), the adsorption of TAN on manure particles can be considered as a further uncertainty. This is represented by a linear relationship with a constant coefficient that describes the equilibrium, which may

influence the calculation for TAN concentrations (see Sect.S3.2 in the Supplementary Material). Uncertainty associated with the adsorption scheme mainly exists in solid manure simulations, such as poultry simulations and solid manure storage.

For the housing simulations, the relationships used to parameterize the indoor conditions may not be representative as they are largely derived from US farms. The ventilation in the naturally ventilated barns can be uncertain, which can influence the rates of the simulated processes. In addition, for the pit housing system, the gap area between floor slats and the width of the

1095 slats in animal houses may vary in practice. AMCLIM used a fixed value that assumes a 20 % gap space of the slatted floor. Meanwhile, the surface area of the pit for manure storage may not be the same as the floor area above, even though it was assumed to be equivalent to the floor area in AMCLIM.

For manure management simulations, the largest uncertainty comes from the source area. In AMCLIM, the area for $NH_3$ emissions in this stage was assumed to be proportional to the housing area based on the logic that more area might be required for manure storage for more animals. This is a reasonable assumption, but the ratio is unclear. The determination of the emitting surface of $NH_3$ emission during manure storage can be considered as a major limitation of AMCLIM because it has a significant effect on emissions, while detailed global data on these differences are apparently not available.

Only surface broadcasting was simulated for land spreading of manure, which may not reflect the reality in countries with policies that require manure to be incorporated in soils, such as the Netherlands and Demark. Emissions from manure application are expected to be overestimated in these places. In order to incorporate such effects in the global simulations of AMCLIM an international database would need to be established that provides statistics on the extent to which solid and liquid manure are immediately incorporated, injected or applied using band-spreading (see Bittman et al., 2014; Sutton et al., 2022). In principle, AMCLIM is well suited to treating such effects. Other uncertainties in the land application have been discussed in the companion paper (Jiang et al., 2024), which also influence the grazing simulations, including: input data for soil characteristics (soil texture, pH and organic matter content), the representation of soil pH dynamic after urea deposition during grazing, and linear relationships used for calculating diffusive and drainage fluxes of N species.

## 5 Conclusions

This paper presents the development and application of the AMCLIM model to simulate $NH_3$ emissions from livestock farming, including pigs, poultry (chicken), cattle, sheep and goats. AMCLIM follows the N flow from animal housing, manure management to the ultimate the land application of manure, with impacts of environmental factors being included in the model. AMCLIM–Housing includes two housing systems and three housing types, and AMCLIM–MMS includes four major manure management types, which allows the impacts of management practices to be reflected, i.e., simulations for livestock sectors and production systems can be differentiated. AMCLIM has also incorporated substantial updates for simulating poultry farming emissions, with more processes being included compared with the previous version of Jiang et al. (2021).

A major effort has been given to the evaluation of AMCLIM–Housing against measurements by USEPA for Animal Feeding Operations (AFO). The simulated $NH_3$ emissions from pig, layer and dairy housing showed close agreement with measurements. AMCLIM–Housing was able to broadly reproduce the $NH_3$ emissions from two types of animal houses with different processes and settings, and roughly replicate the daily variations in $NH_3$ emissions. The other two modules were not specifically tested against measurements because of the similarities in the processes for housing and manure management and lack of available datasets. By comparison AMCLIM–Land has been tested elsewhere and details are given in Jiang et al., (2024). In the case of grazing, lack of comparably detailed measurement data to the USEPA studies, meant that this study focused on a multi-side global comparison with average emissions from studies according to climatic and management differences. While the detailed and multi-site comparisons with measurements are encouraging, there is an ongoing need for

detailed reporting of quality assured $NH_3$ flux measurements with known uncertainties as a basis for further model-measurement comparisons.

Based on simulations using AMCLIM, global livestock farming results in an estimated 29.9 Tg N of $NH_3$ emissions for the reference year 2010. Specifically, cattle are found to the be the largest emitting group, resulting in 17.7 Tg N yr$^{-1}$ emissions. Pig farming is estimated to contribute 5.3 Tg N yr$^{-1}$ and chicken farming results in 4.8 Tg N yr$^{-1}$, while sheep and goat together lead to the remaining estimated 2.4 Tg N yr$^{-1}$. This indicates that overall, around 30 % of total excreted N is lost due to $NH_3$ emissions. High emissions from livestock farming are typically found in Brazil, China, India, US and Europe, which coincides with regions that have high livestock population numbers. The volatilization rates show strong spatial variations across the globe, with the highest volatilization rates being up to 60 % or 70 % of excreted N. In particular, hot regions generally exhibit larger volatilization rates than cold places for livestock farming. By comparison, water conditions can have different effects regarding the specific livestock management, i.e., wet conditions can facilitate uric acid hydrolysis and cause larger $NH_3$ emissions for poultry (chicken) housing. Conversely, drier conditions increase the concentration of urea and TAN for other livestock so that emission potential can be higher. These findings once again demonstrate that simple EFs maybe not sufficient to reflect real world conditions, and the need to refine current EFs to incorporate climate dependence of $NH_3$ emissions.

As compared with the reference year 2010, results of simulations for 2018 show little overall differences in annual volatilization rates for livestock housing (16.8 % for 2010 and 16.9 % for 2018) and manure management (14.4 % for 2010 and 14.6 % for 2018). By contrast, land application of livestock manure shows more inter-annual variability than housing and manure management, with obviously lower volatilization rates in 2018 (32.2 % for 2018 vs 35.9 % for 2010). This is consistent with differences observed between 2010 and 2018 for simulations of $NH_3$ emissions from synthetic fertilizer (Jiang et al., 2024), reflecting year 2018 being a hotter but wetter year than 2010. Such a phenomenon is largely due to the fact that environment of housing and some manure management practices are controlled and less varying than emissions from land. The estimates by the AMCLIM model emphasize the importance of both environmental factors and local management practices. Compared with the traditional approach of estimating emission factors, AMCLIM provides an appropriate tool to allow process-based estimation of the ways in which climatic and management factors may affect $NH_3$ emissions at site, regional and global scales.

**Appendix**

**Table A1. Biomaterial and characteristic information of livestock excreta as used in AMCLIM.**

| Livestock | Urinary N : Faecal N ratio | Urinary N concentration (g N L$^{-1}$ urine) | Faecal N content (g N kg$^{-1}$ faeces) | Fraction of urinary N as urea | Urination (L head$^{-1}$ d$^{-1}$) and defecation (kg head$^{-1}$ d$^{-1}$) | DM (g per kg excreta) | pH |
|---|---|---|---|---|---|---|---|
| Beef/Feedlot Cattle | 3:2 | 7.2 | 4.85 | 0.75 | 12.0 (U) 20.9 (D) | 181.5[a,b] | 7.8 |
| Dairy/Other dairy | 8.8:5 | 6.9 | 4.85 | 0.75 | 21.0 (U) 27.0 (D) | 181.5 | 7.8 |
| Sheep | 2:1 | 8.7 | 6.40 | 0.80 | 2.4 (U) 1.2 (D) | 155.0 | 8.0 |
| Goat | 1:1 | 12.0 | 6.40 | 0.80 | 2.4 (U) 1.2 (D) | 155.0 | 8.0 |
| Pigs | 2:1 | 6.4 | 11.90 | 0.75 | 3.8 (U) 1.2 (D) | 222.0 | 7.7 |
| Poultry | -- | -- | 50 (g N kg$^{-1}$ excretion) | 0.6 (excreted N as UA) | 0.0 (U) 0.03 (Excretion) | 574.0 | 8.5 |

UA is uric acid; U is urine; D is dung.

a, Waldrip et al. (2019); b, Smith et al. (1800) etc.

Vu et al. (2009a, b); Andersen et al. (2020); Haynes and Williams (1993); Marsden et al. (2020); Dong et al. (2014); Waldrip et al. (2013); Nahm (2003); Hoogendoorn et al. (2011); Choirunnisa et al. (2019); Zhao et al. (2016); Reed et al. (2015); Sommer and Hutchings (2001); Misselbrook et al. (2016); Selbie et al. (2015).


**Table A2. Divisions of manure management used in AMCLIM.**

| Category | Solid | Liquid |
|---|---|---|
| A | composting, deep litter, litter (poultry)[a], pit storage (intensive layers)[a], solid storage | aerobic processing, pit storage (livestock except for intensive layers) |
| B | -- | aerobic lagoon, liquid[b] |

| | | |
|---|---|---|
| C | -- | lagoon, liquid crust |
| D | daily spread (cattle, small ruminants, chickens)[c], dry lot, outdoor confinement area | daily spread (dairy cattle, pigs)[b] |
| Grazing | pasture, pasture + paddock | |
| Fuel | burned, digester (biogas) | |
| Unmanaged | discharge, dumping, fishpond, public sewage | |
| Other | sold, thermal drying | |

[a] Counting as housing emissions. [b] To differentiate with "liquid crust", "liquid" is assumed to be an uncovered storage. [c] Counting as MMS emissions.

**Table A3 Information of US Environmental Protection Agency Animal Feeding Operations monitoring data.**

| Site name | Location | Livestock/Production system | Number of rooms/houses monitored | Monitored period |
|---|---|---|---|---|
| IN3B | Carroll, Indiana | Pig | 4 | 01 July 2007 to 31 July 2009 |
| NC2B | Nash, North Carolina | Chicken (layer) | 2 | 15 March 2008 to 15 March 2009 |
| IN5B | Jasper, Indiana | Dairy | 2 | 01 July 2007 to 31 July 2009 |

**Table A4 Model inputs for site simulations and global simulations. *The reference year of these data is 2010, and changes in livestock population and N excretion rates over time are based on the variations suggested by Lu and Tian (2017) to derive livestock data in year 2018.**

| | Environmental variables | Activity and management data |
|---|---|---|
| Site simulations | indoor and outdoor temperature, relative humidity and ventilation from US EPA AFO datasets (Lim et al., 2010a; Wang et al., 2010) | animal number, biomaterial information, animal house information and management events from US EPA AFO datasets (Lim et al., 2010a; Wang et al., 2010) |
| Global simulations | ERA5 reanalysis meteorological variables and soil data from HWSD v1.2 as described in Jiang et al. (2024) | *livestock population, distribution, N excretion, production systems and manure management systems from FAO GLEAM model, FAOSTAT and GLW model. |


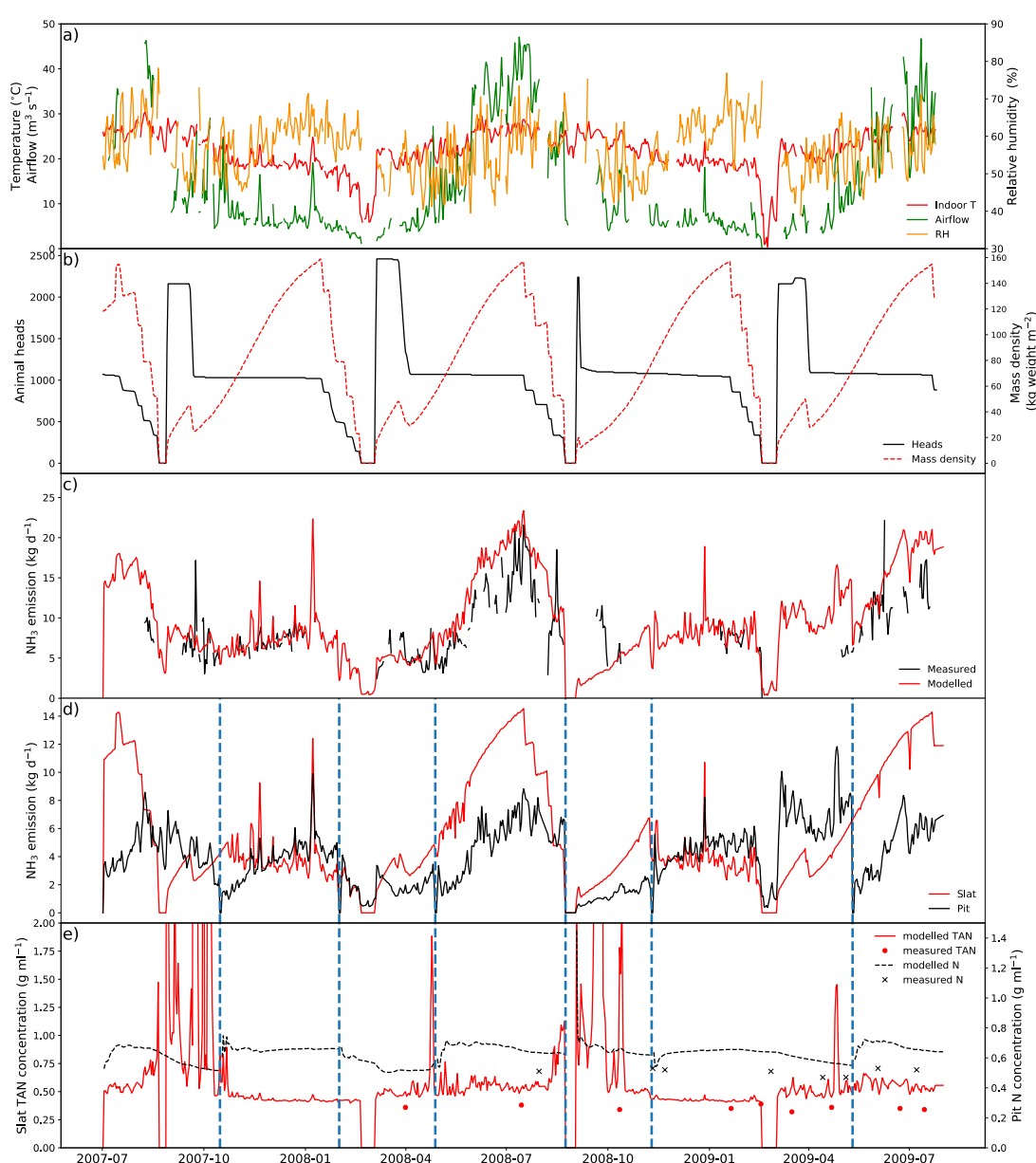

**Figure A1. Same as Figure 4, but for House 2 at site IN3B.**

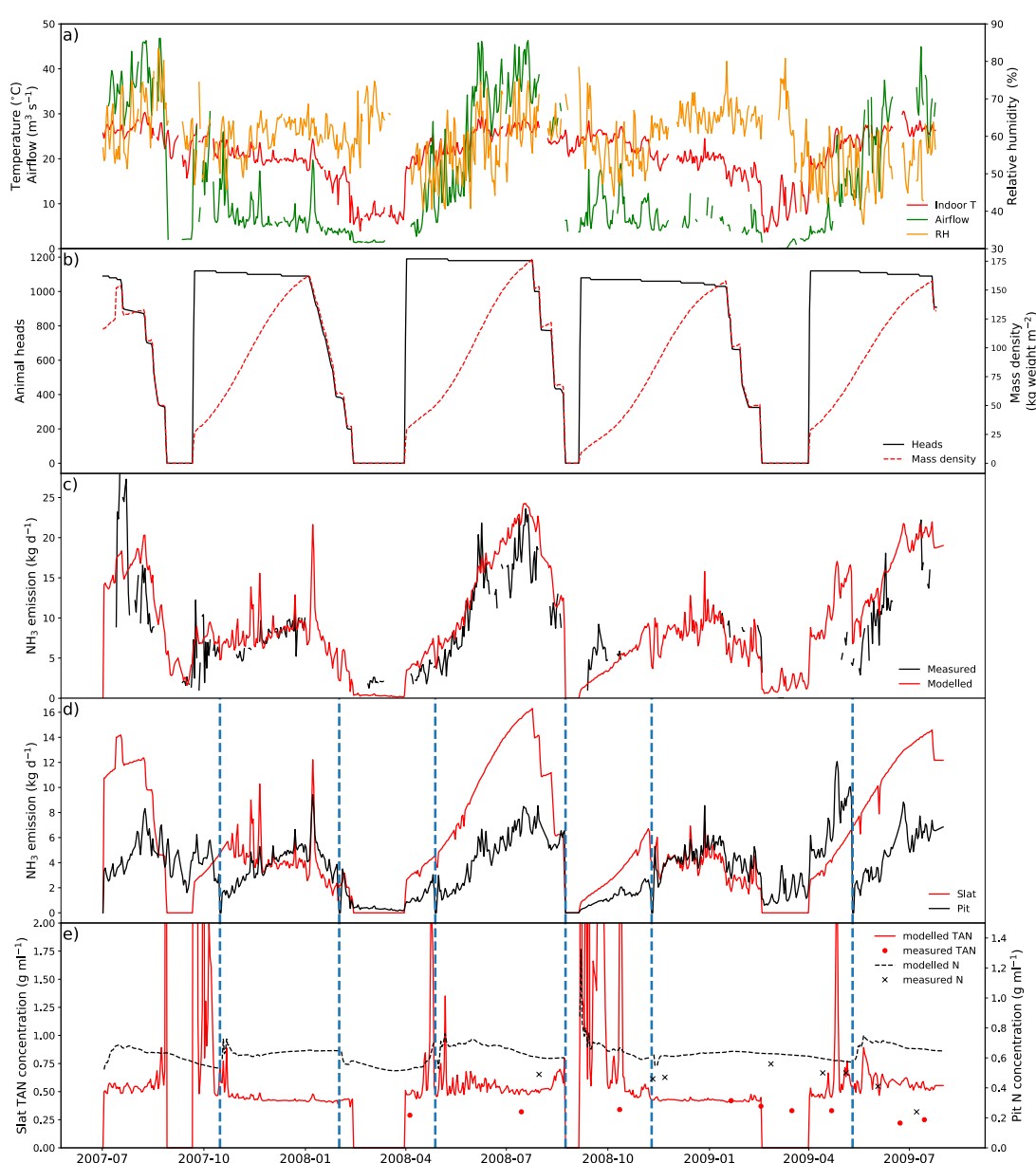

**Figure A2. Same as Figure 4, but for House 3 at site IN3B.**

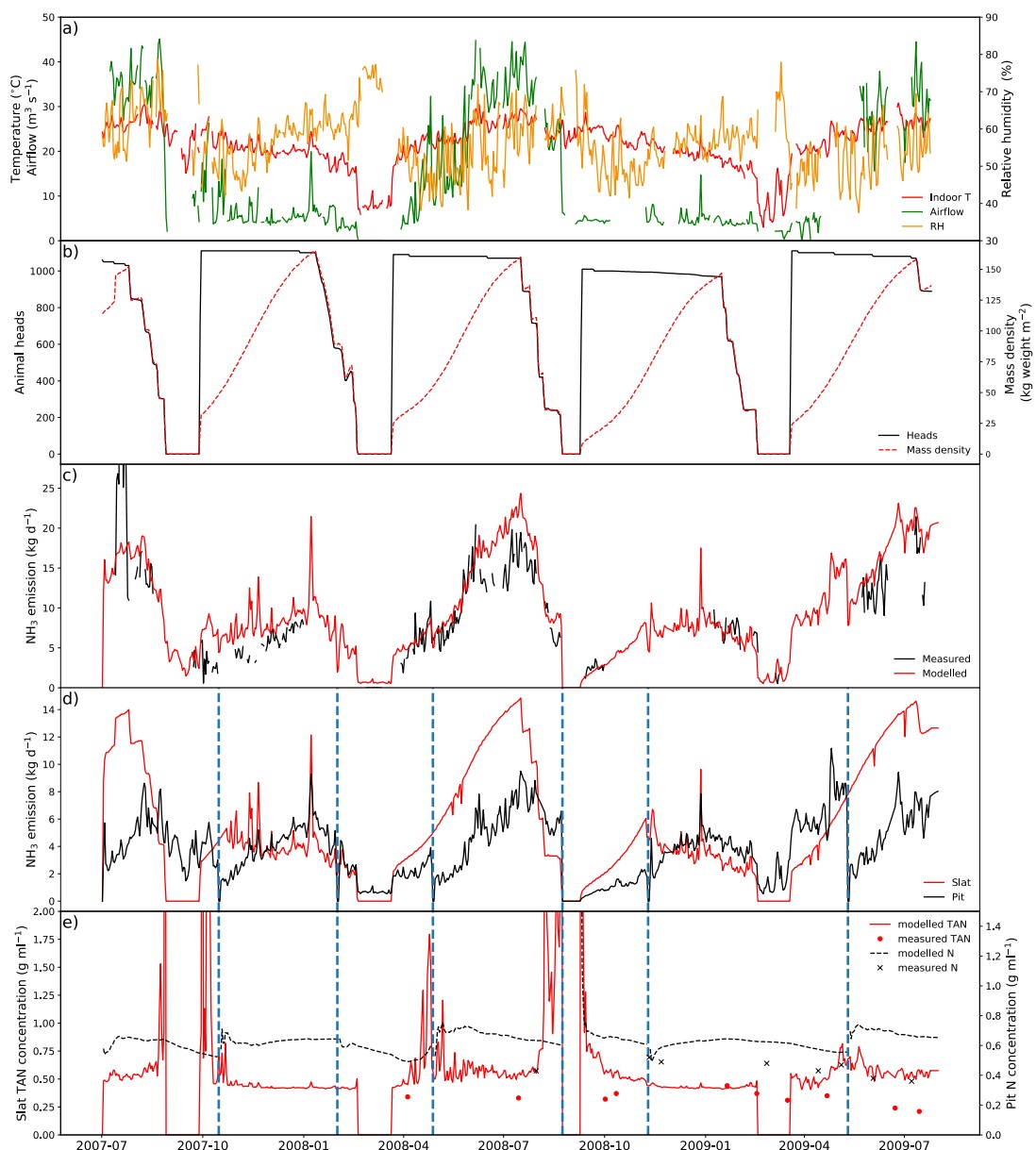


**Figure A3. Same as Figure 4, but for House 4 at site IN3B.**

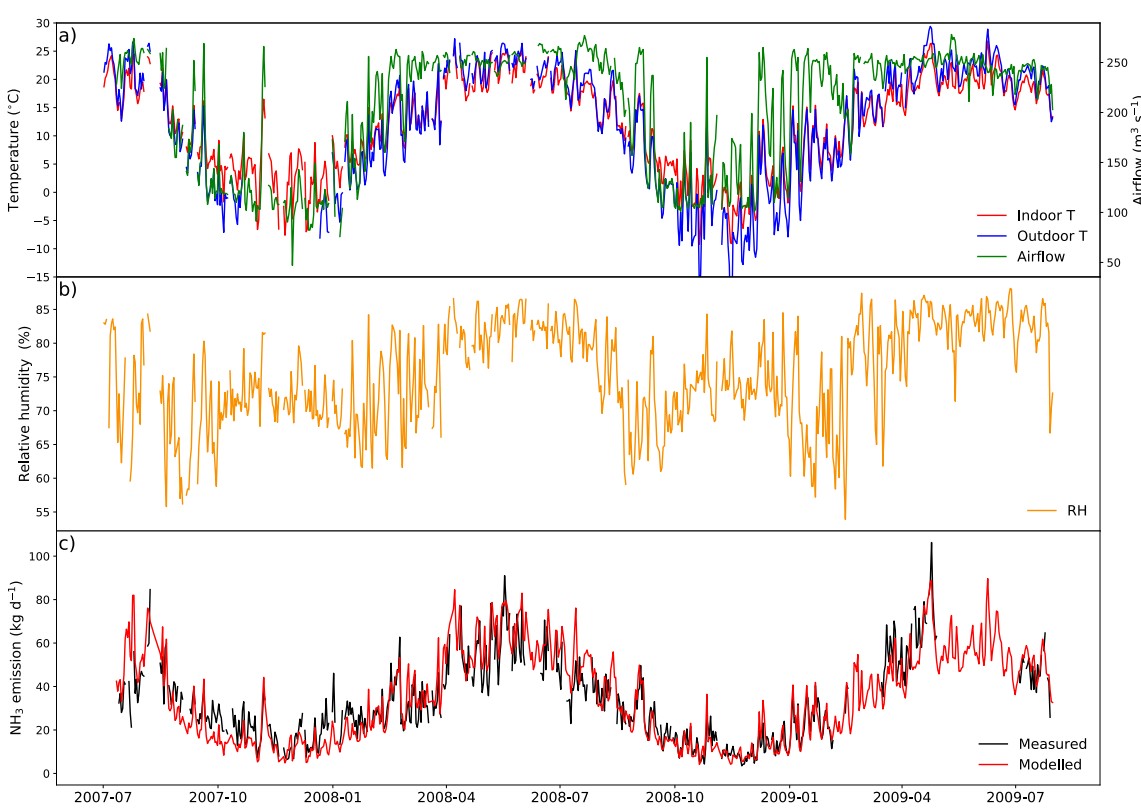

**Figure A4. Same as Figure 5, but for Barn 2 at site IN5B.**


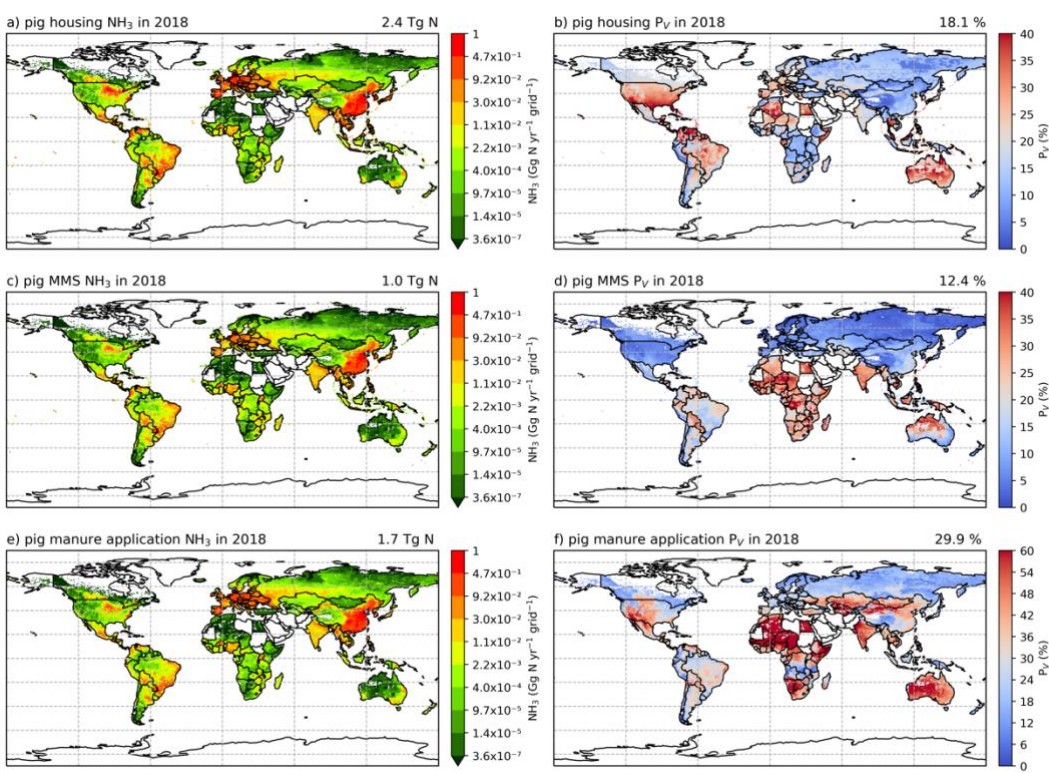

**Figure A5. The same as Figure 6, but for 2018.**

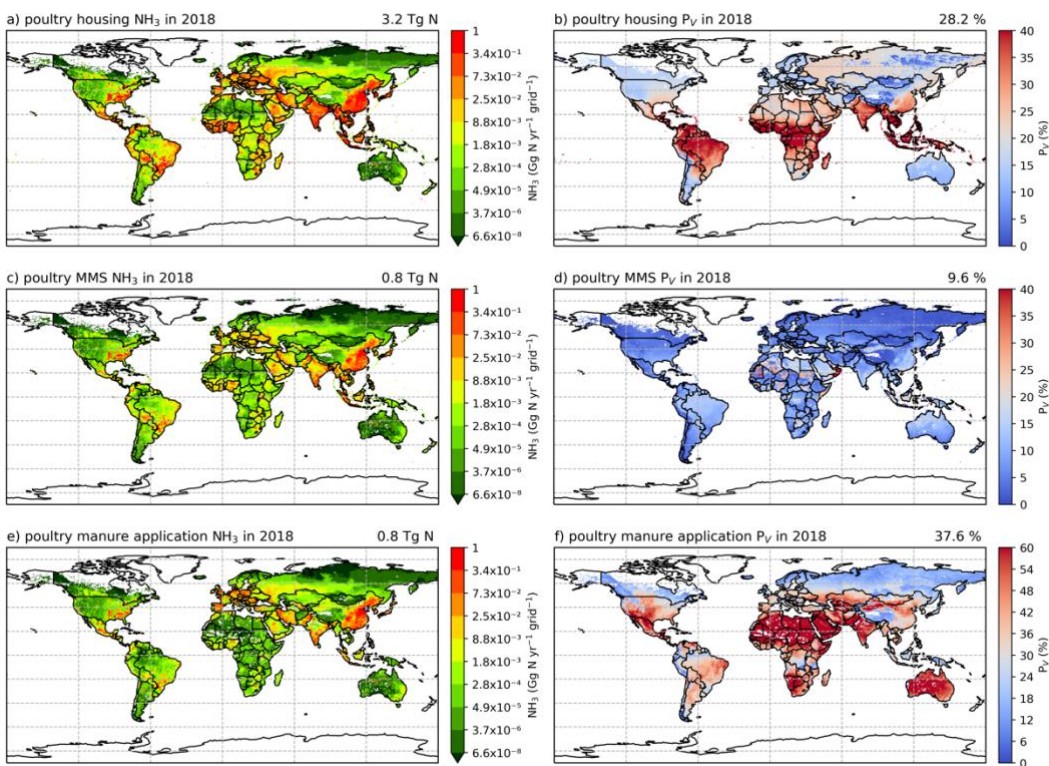

**Figure A6. The same as Figure 7, but for 2018.**


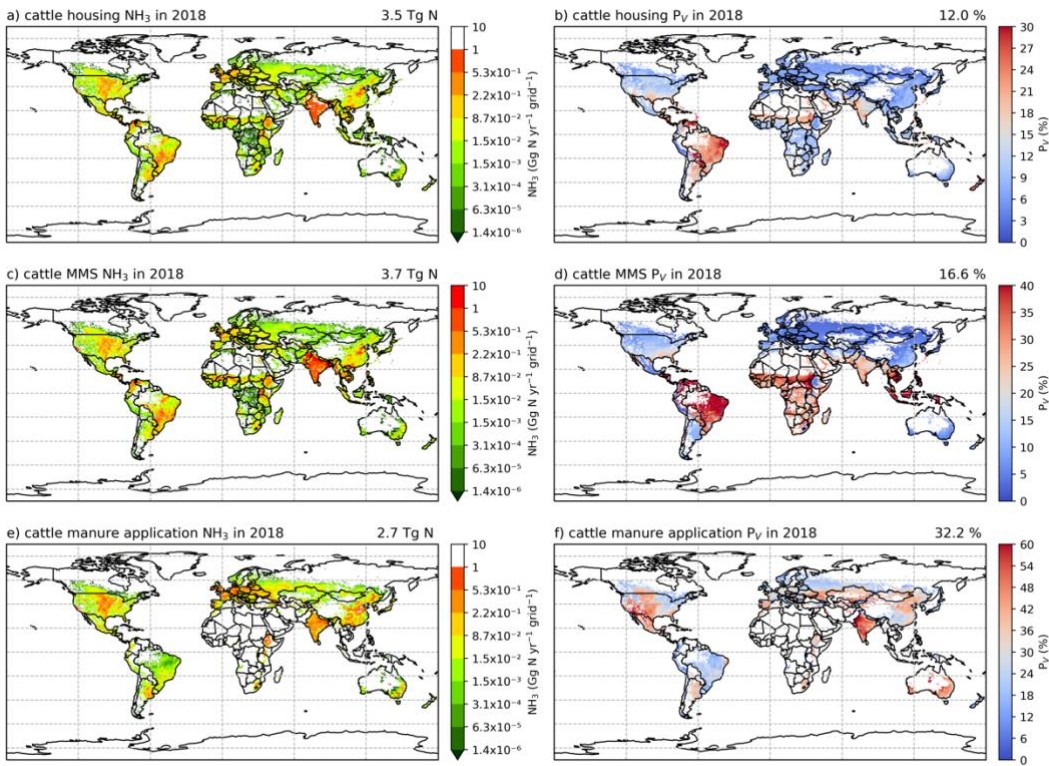

**Figure A7. The same as Figure 8, but for 2018.**

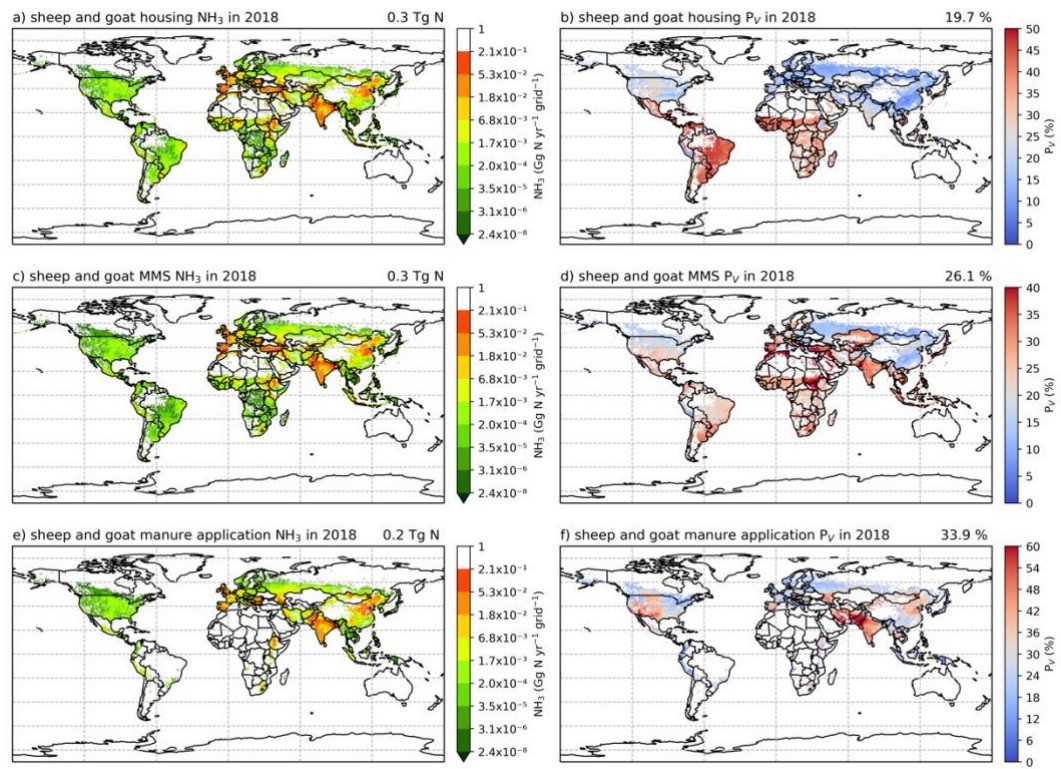

**Figure A8. The same as Figure 9, but for 2018.**


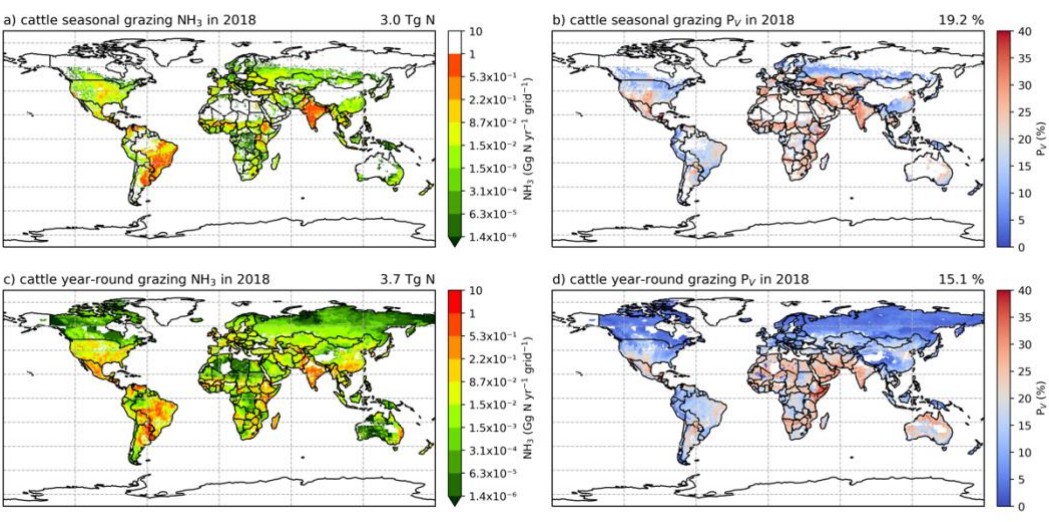

**Figure A9. The same as Figure 10, but for 2018.**

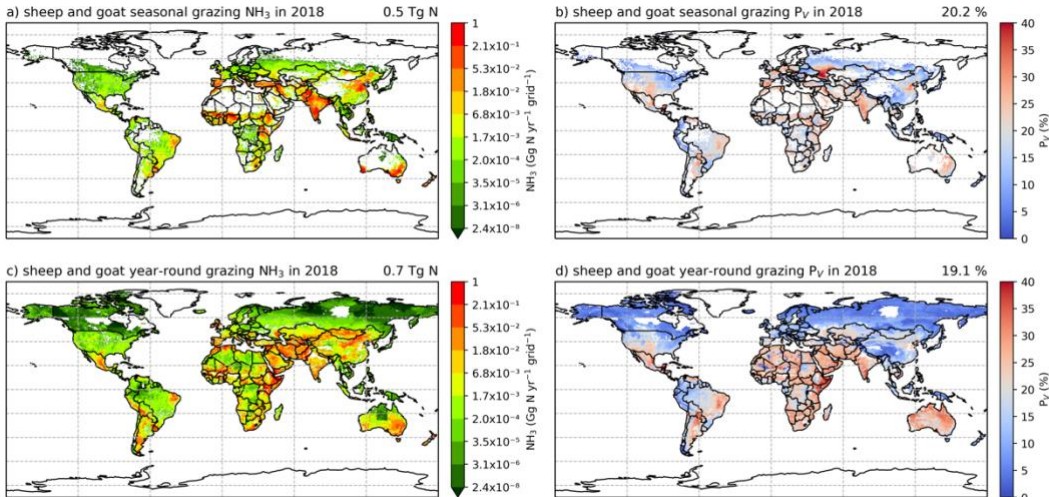

**Figure A10. The same as Figure 11, but for 2018.**

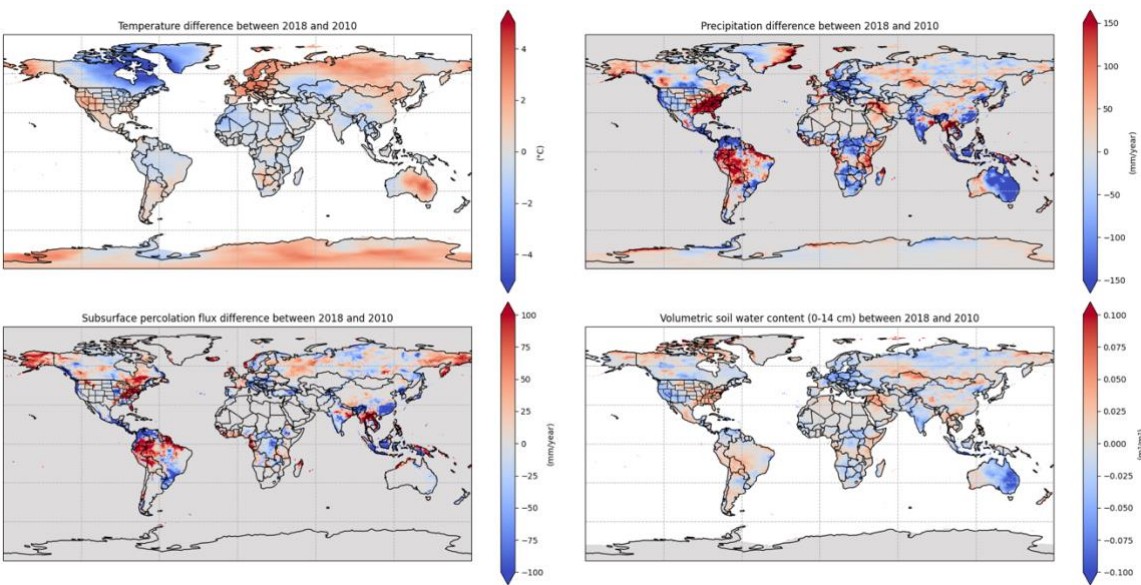

**Figure A11. Differences of meteorological variables, including annual mean temperature (2m), precipitation, subsurface percolation flux and volumetric soil water content (0-14 cm) between 2010 and 2018.**

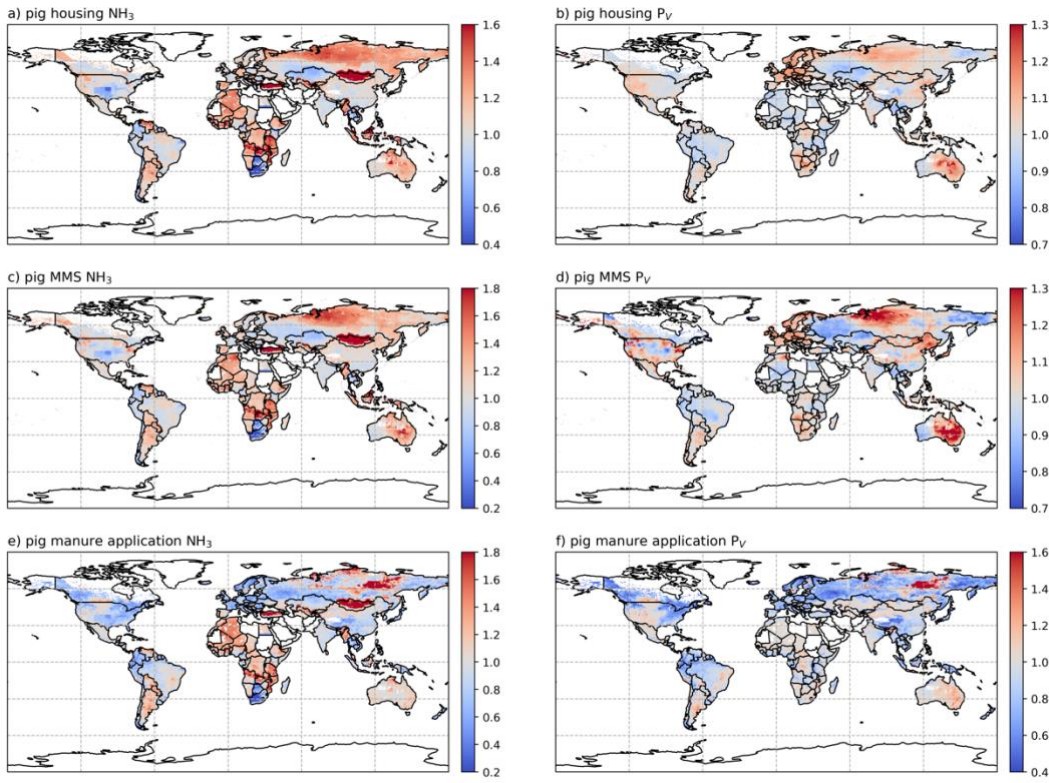


**Figure A12. Ratios of 2018 to 2010 NH₃ emissions (a, c, e) and volatilization rates ($P_V$) (b, d, f) from housing, manure management and manure application for pigs.**

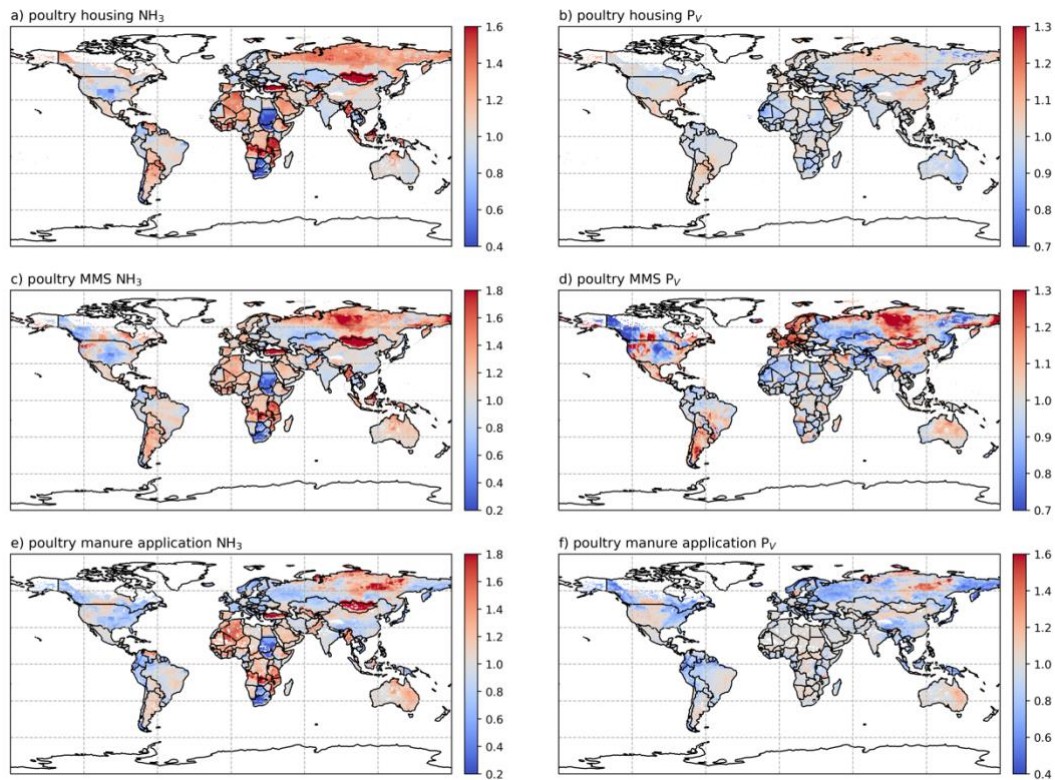

**Figure A13. The same as Figure A12 but for poultry (chicken).**

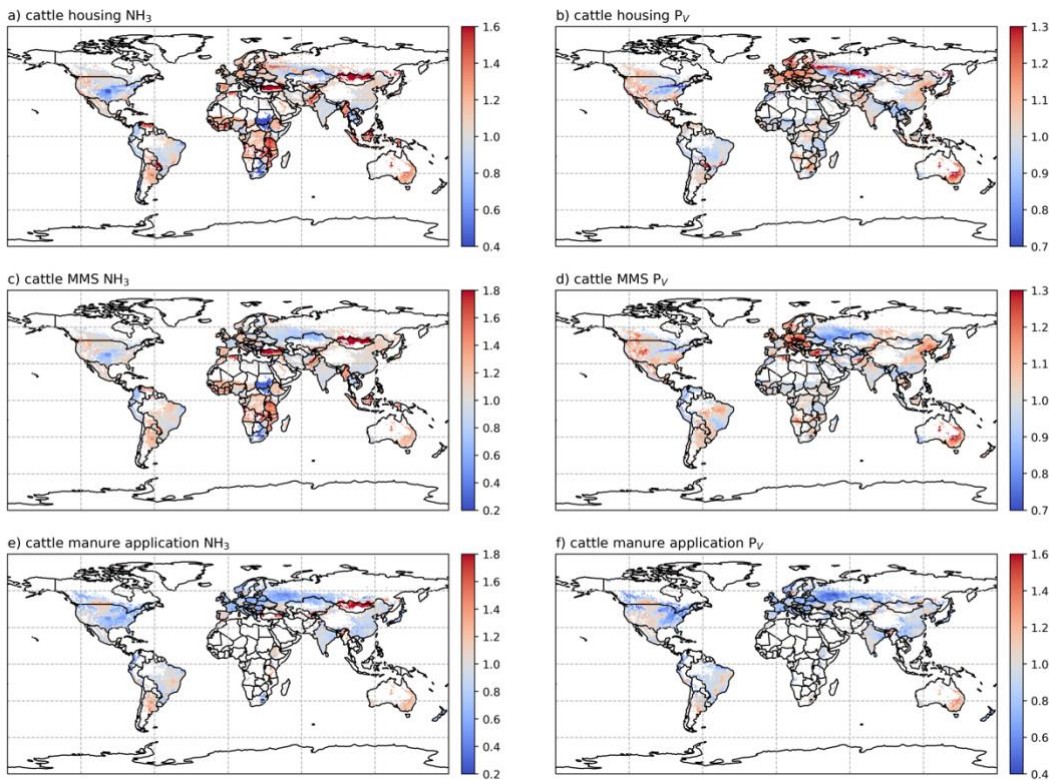


**Figure A14. The same as Figure A12 but for cattle.**

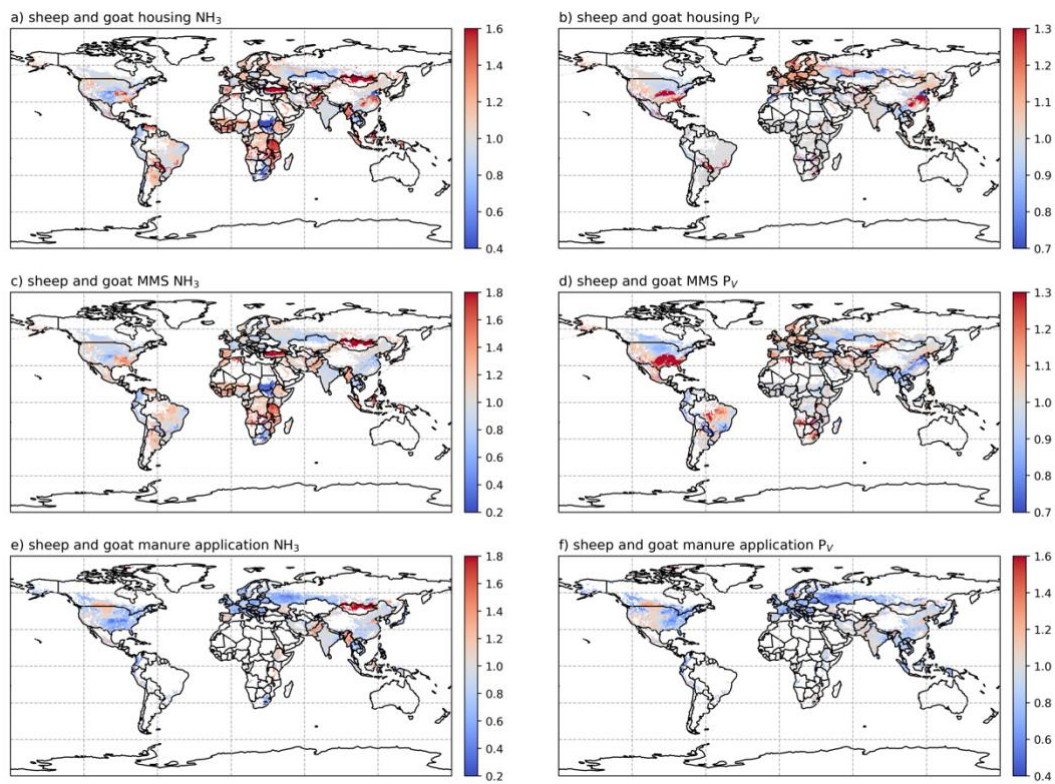

**Figure A15. The same as Figure A12 but for sheep and goats.**

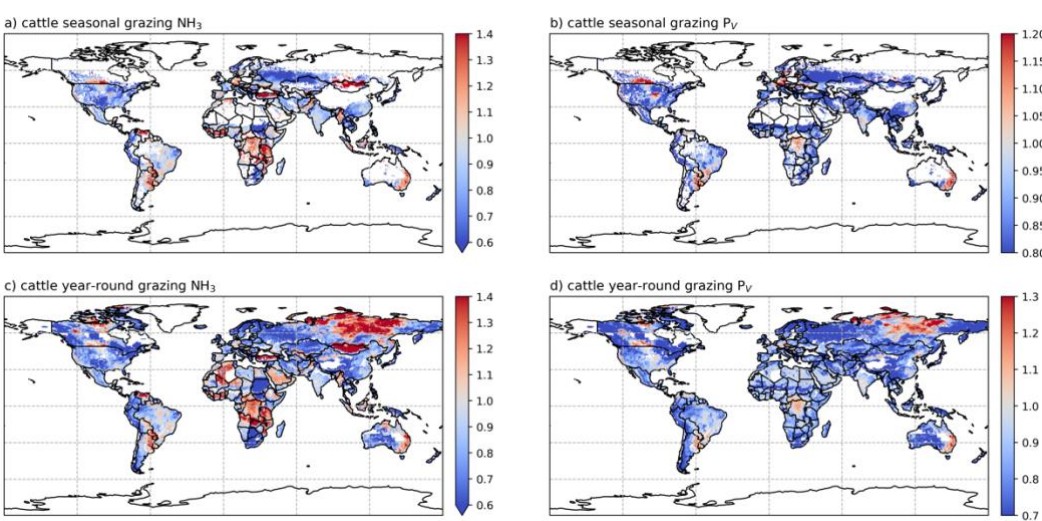

**Figure A16. Ratios of 2018 to 2010 NH₃ emissions (a, c) and volatilization rates ($P_V$) (b, d) from seasonal and year-round grazing for cattle.**

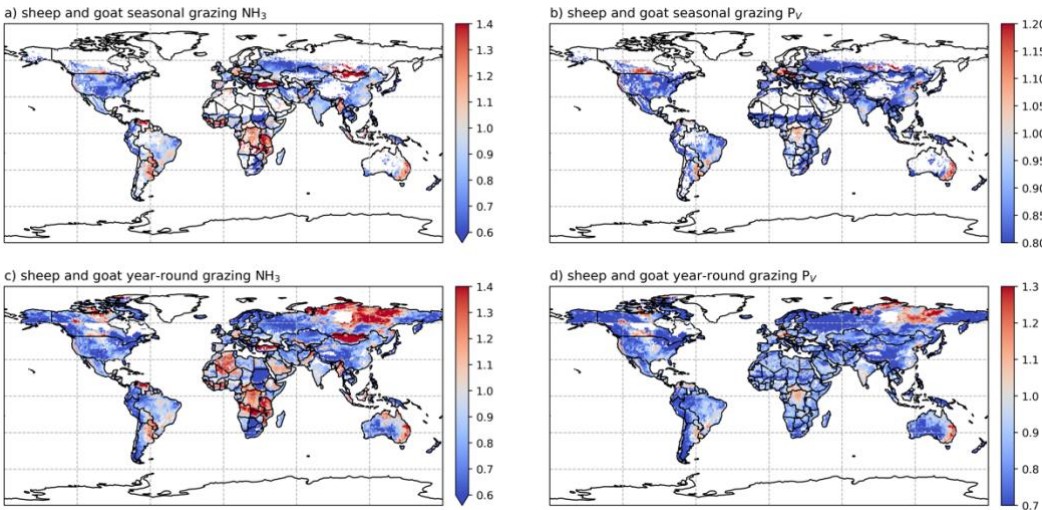

**Figure A17. The same as Figure A16 but for sheep and goats.**


*Code and data availability*. Code of the model can be obtained at github (https://github.com/jjzwilliam/AMCLIM last access: 03 April 2024) and Zenodo (https://zenodo.org/records/10911886 last access: 03 April 2024). Model results presented in this study are in netCDF format and can be freely accessed from the Edinburgh DataShare (https://datashare.ed.ac.uk/handle/10283/8953, Jiang et al., 2025).

*Supplementary Materials.* The Supplementary Materials related to this article has been submitted.

*Author contributions.* JJ, DSS and MAS designed the research. JJ developed the model, write the code, performed the simulations and wrote the paper. AU, GT, AF and FC provided model input data. All authors contributed to analysis and discussion of the model outputs, interpretation of results, writing and critical revision.

*Competing interests.* The authors declare that they have no conflict of interest.


*Acknowledgement.* Jize Jiang gratefully acknowledges support from University of Edinburgh, UK Centre for Ecology and Hydrology (UK CEH), ETH Zurich, and the UK national supercomputing ARCHER2.

*Financial support.* This research has been supported by the UK Natural Environment Research Council (grant no. NE/S009019/2), by UKRI through the GCRF South Asian Nitrogen Hub, funded through the Global Challenges Research Fund, and by the ReCLEAN Joint-Initiative co-financed by the ETH board. The authors are grateful for the support from the Global Environment Facility (GEF) through the UN Environment Programme (UNEP) for the project "Towards the International Nitrogen Management System (INMS)"; from the UKRI under its Global Challenges Research Fund for support of the GCRF South Asian Nitrogen Hub (grant no. NE/S009019/2); from NERC for National Capability support, including through the CEH SUNRISE project; and from the ReCLEAN Joint Initiative at ETH Zurich under the ETH Board Joint Initiatives scheme.

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
