# Peer review of "A dynamical process-based model for quantifying global agricultural ammonia emissions – AMmonia–CLIMate v1.0 (AMCLIM v1.0) – Part 2: livestock farming"

_EGUsphere, 2024_

## Author Comment (AC1)

**Response to Anonymous Referee #1**
**Overall response:** We would like to thank reviewer 1 for these insightful and useful comments. These help improve the manuscript. Here we outline the point-by-point responses below in blue, and the relevant figures are attached.

**Comment:** The paper represents the second part of the description of the AMmonia-CLIMate (AMCLIM) model for quantifying ammonia emissions specifically from livestock production responding to environmental conditions and agricultural practices. The model which incorporates a high level of detail is well described in general and seems to perform well compared to observations and also to previous global-scale estimates. A serious and appropriate sensitivity analysis is also conducted confirming the important role of pH in the emissions.

The strength of the approach lies in the diversity of agricultural practices data and a detailed representation of ventilation scheme in the housing that are considered for ammonia emissions.

The study will likely provide a major advancement for the community especially for designing precise mitigation strategies for $NH_3$ losses but I have some comments before it can be considered for publication.

**Reply:** We appreciate that the reviewer recognizes the value of our study. We thank the reviewer for spending time reviewing the manuscript and the development of the AMCLIM model.

**Comment:** I believe it is crucial in the introduction to put the AMCLIM approach in the state-of-the-art modeling context for ammonia emissions from agriculture. From l.51, this part can be more developed/precise to highlight the strength of AMCLIM. For instance, global processed-based models included in Land Surface Models (LSM) such as FAN v2 in CESM (Vira et al., 2020) and CAMEO in the IPSL ESM (Beaudor et al., 2023) in which detailed soil C/N cycles are implemented with multiple interactions (vegetation, soil BGC, water, and energy budget) also incorporate agricultural practices (by livestock type) and run with dynamic environmental conditions at the global scale. However one of their weaknesses is to rely on Emission Factors (EF) for the manure management part. I think it would be helpful to mention it in the introduction because AMCLIM is an interesting trade-off between process-based models/ Earth System Models and more socio-economic models such as the Integrated Assessment Models (IAMs).

**Reply:** We agree with the reviewer. We improved the text to highlight the state-of-art of AMCLIM for simulating agricultural $NH_3$ emissions. From L61, we added

"A process-based, dynamical emission model, AMmonia–CLIMate (AMCLIM) has been specifically designed that incorporates the effect of both environmental conditions and management practice to simulate agricultural $NH_3$ emissions. Compared with existing process-based models, AMCLIM is thought to be the first model that simulates $NH_3$ emission from both synthetic fertilizer use and livestock farming using a consistent process-based modelling approach, with high levels of detail of the representation of agricultural practices. There are other process-based models, such as the 'Flow of Agricultural Nitrogen' model, version 2' (FANv2; Vira et al. 2020) that simulates agricultural $NH_3$ emissions interactively within the Community Earth System Model (CESM) with detailed soil processes for land application of fertilizers and ruminant grazing. Another is the 'Calculation of AMmonia Emissions' model (CAMEO), which includes several management modules for livestock feed, manure management and agricultural handling practices within the global land surface model ORCHIDEE (Beaudor et al., 2023). While these models still largely rely on emission factors (EFs) for estimating $NH_3$ emissions from livestock sectors, AMCLIM explicitly models the N flows within the systems and includes several major N processes. AMCLIM uses an integrated approach to simulate how various N species are influenced by environmental factors in a sequence of the practices in the livestock sector, from livestock housing to manure management and ultimate application of manure to fields, as well as ruminant grazing. By following this sequence in AMCLIM, changes in emissions at an early stage of livestock agriculture influence the simulated N pools, and can thereby affect emission at a later stage of these activities. The simulations for global synthetic fertilizer use have been presented in the companion paper (Jiang et al, 2024)."

**Comment:** It is hard to get the global picture of the results of the different $NH_3$ volatilizations; the "per grid" unit for the emission flux distributions (Fig 7-11) is not common and it is hard to compare with other literature results (which are usually given by /m2). Since the spatial resolution of the model is also not clearly stated, it is almost impossible to compare it with any other results. I know that the Part 1 paper also presented the ammonia flux in that unit and maybe for consistency reasons, the authors would like to stick to that unit but a clear justification should be included.

**Reply:** We thank the reviewer for pointing this out. The global maps of $NH_3$ emissions have a unit of Gg N $yr^{-1}$ $grid^{-1}$. The resolution for all simulations is the same, that is 0.5° x 0.5°. We improved the manuscript by adding the following text from L492

"Global simulations were performed to estimate $NH_3$ emissions from livestock farming for the year 2010 and 2018, and had the consistent setup as simulations for synthetic fertilizer use, as described in the first part of the model (Jiang et al., 2024). AMCLIM was applied using a longitude–latitude grid at a resolution of 0.5° × 0.5°. All model inputs were regridded to the model resolution if necessary. The simulations were performed

at an hourly time step, and the prognostic variables at each time step were solved by the Euler method in the model."

**Comment:** On the same topic, to ease the comparison with other global results, one suggestion would be to include a global map of the combined synthetic fertilizer and livestock emissions (i.e. total agricultural sources) in the common units: gN/m2/yr. This map could be shown as a final result?

**Reply:** To address the question of units, we have now added a final map of total $NH_3$ emissions from livestock in units of gN/m2/yr as requested (see below; Figure 14 in the revised manuscript). However, the synthesis of emissions from livestock and fertilizers is part of another planned publication, and we think it better to present that map there as part of a discussion of global agricultural ammonia, rather than to risk duplication.

[Figure]

[Figure]

**Figure R1-1. Simulated (a) annual global $NH_3$ emissions (gN m$^{-2}$ yr$^{-1}$) from livestock farming (including housing, manure management, land application of manure and grazing) in 2010. (b) Percentage of total livestock excreted N that volatilizes ($P_V$) as $NH_3$ in 2010. The resolution is 0.5° × 0.5°.**

**Comment:** The 2-year (2010 vs. 2018) comparison needs to be justified earlier in the manuscript and I am missing a description of the input that has been used for 2018. The setup for this experiment is not properly described in the Methods. Is the livestock distribution dataset for 2010 vs. 2018 the same? How the 2010 year inputs have been extrapolated?

If the meteorological impact is the major reason behind the 2-year comparison maybe the title of section 4.4 could be more explicit.

Please consider that 2018 results already appear in the previous sections when describing the maps of the volatilization rates.

**Reply:** Livestock inputs for 2010 and 2018 are different. Livestock and MMS data for the year 2010 were obtained from the FAO GLEAM model. We applied the interannual variability between 2005 to 2015 reported by Lu and Tian (2017) and extended to year 2018 by a linear interpolation. The meteorological inputs were from ERA5 reanalysis datasets for both years. To address the reviewer's comments, we made the following changes for the revised manuscript, from L455

"In the present paper, the combined AMCLIM model was applied for 2010 and 2018 to demonstrate full simulations for two different years, with activity data and meteorological variables varied between years, so that the inter-annual variability in both emissions and volatilization rates can be analysed."

and from L466

"...The reference year of these data is 2010. For simulations for the year 2018, livestock population and N excretion rates were extended by a linear interpolation based on the inter-annual variations between 2005 and 2015 suggested by Lu and Tian (2017). The MMS data that determines the fraction of a manure management system are assumed to be constant through the year."

**Comment:** The maps related to 2018 are in the Appendix, this makes the manuscript somehow very heavy for the reader to do this back and forth (especially for 6 maps x 3 animal types). Is there any way to combine these results maybe in a histoplot by region instead of multiple maps?

**Reply:** Here we provide a histogram as suggested by the reviewer to show the yearly difference of $NH_3$ emissions from the livestock sectors across different geographical regions, together with simulated volatilization rates for all livestock groups. We have added this as a new Figure 15 to the main text of the paper. We added the following texts from L901

"As shown in Fig. 15, $NH_3$ emissions and volatilization rates vary across different geographical regions and between two simulated years. The highest $NH_3$ emissions from livestock agriculture are estimated to occur in East and South Asia. In general, the volatilization rates for livestock are lower in 2018 than 2010, except for poultry. This is because a large fraction of poultry which are broiler and layer production systems are assumed in the model to be kept in houses with controlled temperature and ventilation, so the $P_V$ rates were less impacted by the differences in environmental conditions between years. By comparison, the $P_V$ rates from sheep and goats show the largest inter-annual variability among all livestock groups because sheep and goats typically graze outside and are housed in naturally ventilated barns. As a result, $NH_3$ volatilization from sheep and goats is more dependent on the environmental conditions than poultry. The differences in housing and grazing management also explain why $P_V$ rates from pigs showed the second smallest difference and cattle showed the second largest difference between the two simulated years."

[Figure]

**Figure R1-2. Estimated (a) NH$_3$ emissions from livestock farming in seven geographical regions for the two studied years, 2010 (left) and 2018 (right, hashed), and comparisons of volatilization rates $P_V$ between the two years for (b) cattle, (c) pigs, (d) poultry and (e) sheep and goats.**

Minor comments:

**Comment:** Fig 1: I think representing the NH$_3$ losses on that schematic would be useful

**Reply:** We would like to keep Fig. 1 the same as the companion paper (Jiang et al., 2024).

**Comment:** L. 174 : Is there any difference between $F_{NH3\ volatilization}$ and $F_{NH3}$ throughout the equations?

**Reply:** No, $F_{NH3\ volatilization}$ and $F_{NH3}$ are the same. We removed $F_{NH3\ volatilization}$ in the revised manuscript.

**Comment:** L.190 : I think the pH variation with the different processes would be interesting to describe. Perhaps a little paragraph explaining its drivers would be beneficial since it is a critical parameter admitted by the authors and known by the community.

**Reply:** We added the following paragraph to the revised manuscript from L219

"As discussed in the companion paper (Jiang et al., 2024), substrate pH is a critical factor that impacts the NH$_3$ emission. The dynamic equilibrium between gaseous NH$_3$ and aqueous ammonium is dependent on pH. On the other hand, pH affects the rates of uric acid hydrolysis and nitrification, which together control the TAN pool. In AMCLIM, the pH of the livestock excretion is used for determining the decomposition rates of N species and chemical equilibria in housing simulations."

**Comment:** L.343 : I am not sure this reference is well placed in the context

**Reply:** We updated the revised manuscript from L382

"In AMCLIM−Land, ruminants in the grassland production system are assumed to graze year-round, whereas those in the mixed production system graze seasonally. This assumption has been made in the FANv2 model (Vira et al., 2020) and was used here."

**Comment:** Section 2.52 : I think a table gathering information on the different inputs used for the global scale (for both 2010 and 2018) and site simulations would be useful

**Reply:** Here we added a new table that summarises the input data used for both the site and global simulations.

**Table A4 Model inputs for site simulations and global simulations. *The reference year of these data is 2010, and changes in livestock population and N excretion rates over time are based on the variations suggested by Lu and Tian (2017) to derive livestock data in year 2018.**

|  | Environmental variables | Activity and management data |
|---|---|---|
| Site simulations | indoor and outdoor temperature, relative humidity and ventilation from US EPA AFO datasets (Lim et al., 2010a; Wang et al., 2010) | animal number, biomaterial information, animal house information and management events from US EPA AFO datasets (Lim et al., 2010a; Wang et al., 2010) |
| Global simulations | ERA5 reanalysis meteorological variables and soil data from HWSD v1.2 as described in Jiang et al. (2024) | *livestock population, distribution, N excretion, production systems and manure management systems from FAO GLEAM model, FAOSTAT and GLW model. |

**Comment:** L. 408: What are the spatial and temporal resolutions of the model?

**Reply:** For global simulations, AMCLIM was applied using a longitude–latitude grid at a resolution of 0.5° × 0.5°. All model inputs were regridded to the model resolution if necessary. The simulations were performed at an hourly time step, and the prognostic variables at each time step were solved by the Euler method in the model.

We have clarified this in the manuscript at the end of Section 2.5 from L492, as follows:

"Global simulations were performed to estimate $NH_3$ emissions from livestock farming for the year 2010 and 2018, and had the consistent setup as simulations for synthetic fertilizer use, as described in the first part of the model (Jiang et al., 2024). AMCLIM was applied using a longitude–latitude grid at a resolution of 0.5° × 0.5°. All model inputs were regridded to the model resolution if necessary. The simulations were performed at an hourly time step, and the prognostic variables at each time step were solved by the Euler method in the model."

**Comment:** L. 486: What could be a reasonable explanation for these discrepancies?

**Reply:** There are a few possible explanations for these differences. First, modelled $NH_3$ emissions were higher than the measurements for summer months in both 2008 and

2009, corresponding to higher simulated TAN (and total N) concentrations of the slatted floor and the pit (as shown in Fig. 4e) than the measured values. This is possibly due to underestimated evaporation in the animal house by AMCLIM. The underestimation of $NH_3$ emissions in Jan 2009 might be caused by a different reason because the modelled TAN concentrations were comparable to the measurements (as shown in Fig. 4e). This indicates that the calculated indoor resistances that constrain the volatilization of $NH_3$ in AMCLIM may be overestimated.

We have clarified these points by updating the following in Section 3.1.1 from L559

"…However, it underestimates the winter emissions (January 2009) by 30 %, which might be caused by overestimated resistances because the simulated TAN concentrations were comparable to the measurements (Fig. 4e). AMCLIM overestimates the summer emissions (June 2009 and July 2009) by a factor of two (Fig. 4c), corresponding to higher simulated TAN and total N concentrations of the slatted floor and the pit than those of measurements (Fig. 4e). This is possibly due to underestimations of indoor evaporation in AMCLIM."

**Comment:** Section 3.2: Is there any logic behind the order of the animal types that are presented? I wonder why cattle are not presented first since they are responsible for the highest emissions.

**Reply:** The results for cattle (and sheep and goats) include $NH_3$ emissions from grazing, which has been described in the methods section after housing, manure management and land spread of manure. We would like to keep the same order in the results description. There is no specific logic behind the order of animals. We agree that cattle are responsible for the largest emissions from the livestock sector, but pigs and poultry farming also represent two major components.

We have addressed this in the manuscript by adding to the start of Section 3. From L649

"In the following sections, emissions are presented in the order of livestock housing, manure management and application of manure, while emissions from grazing are considered in section 3.3. As with the previous sections, pigs are presented first, followed by poultry, as representing a systems dominated by all-year animal housing. Then ruminants including cattle, sheep and goats are presented, as systems which are complicated by the widespread practice of partial-year housing."

**Comment:** Section 3.2.2: I would add a reference to the section where the new poultry emissions are compared to the older version. This would help the reader to understand that there is a dedicated section for that comparison.

**Reply:** We added a reference to Section 4.5 that discusses the comparison between new and old model version at the end of Section 3.2.2 from L695

"… Since the current model version has updated processes for simulating $NH_3$ emissions from poultry agriculture, a comparison of results between the current model version as described in this section and the previous model version by Jiang et al. (2021) is discussed in Section 4.5."

**Comment:** The title of Section 3.3.2 could be a bit more precise: "Comparison of grazing $NH_3$ emissions with observations" ?

**Reply:** We changed the title of Section 3.3.2 to "Comparison of estimated grazing $NH_3$ emissions using AMCLIM with observations"

**Comment:** Section 3.4: For the N budget, I wonder if other N species were also taken into account. $N_2O$, $N_2$ and $NO_2$ emissions can also originate from manure. Even though these EFs are relatively small compared to $NH_3$, I think it is important to mention them for a consistent N budget. I invite the authors to have a look at studies from Sommer et al., 2019 and EMEP/EEA 2019 for more information.

**Reply:** The N budget shown in this manuscript does not include $N_2O$, $N_2$ and $NO_2$ emissions because the present version of AMCLIM does not simulate denitrification process (and $N_2O$ from nitrification), as presented in Fig. 2 in the companion pager (Jiang et al., 2024). These fluxes are considered to be implicitly represented by the total nitrification term. We clarified this in the revised manuscript. Here we made the following changes from L857

"Figure 13 summarizes the simulated N flows of global livestock farming for the reference year 2010 by AMCLIM, which are allocated to housing, manure management, and application to land, with a focus on $NH_3$ emissions. Other simulated nitrogen pathways include surface runoff, nitrification, leaching and diffusion to deeper soils, uptake by plants and amount left in soils. As specified in the description of soil processes in AMCLIM (Jiang et al., 2024), denitrification and emission of NO, $N_2O$ and $N_2$ are not explicitly included in this study. The flux of "nitrification" in the N budget simulated in this study can be seen as a sum of both nitrified and denitrified N, with the amount of all relevant species (NO, $N_2O$ and $N_2$) being included."

We agree that the studies of Sommer et al. and EMEP/EEA 2019 are useful on how to consider such interactions further.

**Comment:** L. 806: Please explain what is the GUANO model.

**Reply:** GUANO model is a process-based model designed for simulating and predicting NH₃ emissions from a source of seabird-derived uric acid (Riddick et al., 2017).

Accordingly, we have now revised the text to note, with new text added from L930:

"AMCLIM has been developed based on prior testing for chicken houses (Jiang et al., 2021), with the model principles building on the earlier simulation approach of the GUANO model (Generalisation of Uric Acid Nitrogen emissions), a process-based model designed for simulating and predicting NH₃ emissions from a source of seabird-derived uric acid, which has been tested in relation to measurements from seabird colonies (Riddick et al., 2017)"

**Comment:** Section 4.2: The GLEAM model and estimated EF from Yang et al would benefit from a little description. I wonder how the AMCLIM EF also compares with EFs from EMEP/EEA or Sommer et al., 2019 both specific to Europe and sometimes applied to the whole globe in several process-based models (Vira et al., Beaudor et al.,). In the study from Sommer et al., 2019, they also analyzed the EFs of the ALFAM2 process-based model (given as % of TAN content in the manure applied) by season and livestock type which could be interesting to compare.

**Reply:** As suggested by the reviewer, we present a new table (Table R.1.2 shown below, that is now included as new Table 4 in the revised manuscript. This table compares the simulated EF (expressed as % of TAN) by AMCLIM with EFs reported by EMEP/EEA and Sommer et al. (2019).

We have added the following text to the manuscript to clarify the main messages of this table from L975

[revised manuscript text omitted]

**Comment:** L.833: For the animal EFs taken from Yang et al., study, would you have any references for the data presented? I don't see any precise reference in their paper but instead this description: "NH$_3$ EFs from different animal types were collected by keyword searches of above several databases, including "animal

ammonia/NH$_3$ emission", "cattle", "buffaloes", "chickens", "ducks", "goats", "sheep", "livestock production" and "animal husbandry operations". "

**Reply:** We also do not have detailed references for the animal EFs as they were not provided by Yang et al. (2022). Only a summary of all collected EFs has been reported by the paper.

**Comment:** L.918: Please define FAN v2. In addition, a little description would be beneficial for the reader maybe in the introduction as suggested earlier.

**Reply:** We added the following text in the manuscript to briefly introduce both FANv2 and CAMEO model.

"...There are other process-based models, such as the 'Flow of Agricultural Nitrogen' model, version 2' (FANv2; Vira et al. 2020) that simulates agricultural NH$_3$ emissions interactively within the Community Earth System Model (CESM) with detailed soil processes for land application of fertilizers and ruminant grazing. Another is the 'Calculation of AMmonia Emissions' model (CAMEO) that includes several management modules for livestock feed, manure management and agricultural handling practices within the global land surface model ORCHIDEE (Beaudor et al., 2023)..."

**Comment:** L. 978: Please correct for "database".

**Reply:** We corrected the word "database".

**Comment:** L.982: I believe the sentence in parenthesis should not be there.

**Reply:** We removed the unnecessary sentence in the parenthesis and updated the text from L1232

"Other uncertainties in the land application have been discussed in the companion paper (Jiang et al., 2024), which also influence the grazing simulations, including: input data for soil characteristics (soil texture, pH and organic matter content), the representation of soil pH dynamic after urea deposition during grazing, and linear relationships used for calculating diffusive and drainage fluxes of N species."

---

## Author Comment (AC2)

**Response to Anonymous Referee #2**
**Overall response:** We would like to thank reviewer 2 for these critical and useful comments. These help improve the manuscript. Here we outline the point-by-point responses below in blue, and the relevant figures are attached.

**Comment:** The manuscript presents AMCLIM, a process-based, dynamic model for estimating $NH_3$ emissions from livestock. The model comprehensively accounts for $NH_3$ emission processes and nitrogen flow across housing, manure management, and land application. Figures 1–3 clearly show these processes, and the model simulations successfully capture $NH_3$ emissions observed from individual animal houses. AMCLIM represents a significant improvement in the estimation of $NH_3$ emissions from livestock. I believe the manuscript should be published, with only a few comments for the authors to address:

**Reply:** We appreciate that the reviewer recognizes the value of our study. We thank the reviewer for spending time reviewing the manuscript.

1. In the Introduction, the authors highlight challenges in previous process-based models when representing various management practices. I recommend a direct comparison between the processes considered in AMCLIM and those in previous studies to better illustrate the differences.

**Reply:** We added the following text to highlight the state-of-art of AMCLIM for simulating agricultural $NH_3$ emissions compared with other process-based models. From L60,

"A process-based, dynamical emission model, AMmonia–CLIMate (AMCLIM) has been specifically designed that incorporates the effect of both environmental conditions and management practice to simulate agricultural $NH_3$ emissions. Compared with existing process-based models, AMCLIM is thought to be the first model that simulates $NH_3$ emission from both synthetic fertilizer use and livestock farming using a consistent process-based modelling approach, with high levels of detail of the representation of agricultural practices. There are other process-based models, such as the 'Flow of Agricultural Nitrogen' model, version 2' (FANv2; Vira et al. 2020) that simulates agricultural $NH_3$ emissions interactively within the Community Earth System Model (CESM) with detailed soil processes for land application of fertilizers and ruminant grazing. Another is the 'Calculation of AMmonia Emissions' model (CAMEO), which includes several management modules for livestock feed, manure management and agricultural handling practices within the global land surface model ORCHIDEE (Beaudor et al., 2023). While these models still largely rely on emission factors (EFs) for estimating $NH_3$ emissions from livestock sectors, AMCLIM explicitly models the N flows

within the systems and includes several major N processes. AMCLIM uses an integrated approach to simulate how various N species are influenced by environmental factors in a sequence of the practices in the livestock sector, from livestock housing to manure management and ultimate application of manure to fields, as well as ruminant grazing. By following this sequence in AMCLIM, changes in emissions at an early stage of livestock agriculture influence the simulated N pools, and can thereby affect emission at a later stage of these activities. The simulations for global synthetic fertilizer use have been presented in the companion paper (Jiang et al, 2024)."

2. Equations (1)–(19) provide a detailed description of how the model calculates NH3 concentrations inside animal houses and nitrogen flow. However, it remains unclear how these nitrogen pools or $NH_3$ concentrations are ultimately converted into NH3 emissions. Could the authors clarify?

**Reply:** The volatilization of $NH_3$ from the land surface to the atmosphere is driven by the concentration difference at two heights and is constrained by a set of resistances. For indoor simulations such as livestock housing and indoor manure storage, the calculation of $NH_3$ is given by the first half on the right hand side of Eq. (2) in the manuscript. We updated the equation to explicitly show how $NH_3$ emission is calculated as follows:

$$\begin{cases} V_{house} \dfrac{d\chi_{in}}{dt} = \dfrac{(\chi_{srf} - \chi_{in})}{R_{G,house}} \cdot S_{house} - Q_{in}(\chi_{in} - \chi_{out}) \\ \qquad\qquad F_{NH_3} = \dfrac{(\chi_{srf} - \chi_{in})}{R_{G,house}} \end{cases}, \tag{2}$$

For the $NH_3$ volatilization processes of outdoor simulations like manure application to field, we refer to the paper that describes the first part of AMCLIM by Jiang et al. (2024), in "Volatilization of NH3" under Sect 2.2.1.

We have drawn attention to this cross reference in the present manuscript by adding the following sentence from L351

"Specifically, the volatilization processes of $NH_3$ have been described in Sect.2.2.1 in Jiang et al. (2024)."

3. In Figure 2, I suggest including the explicit equations for *f(Tgnd, pH, RH)* and *f(Tgnd, pH)* since the influence of pH on the calculations is discussed in the manuscript.

**Reply:** We found it quite difficult to explicitly include all the equations in Figure 2 as requested due to limited space in the figure. However, we have added key relationships in a revised version of Figure 2. When it is not feasible to add an equation, we refer to the equation numbers of the manuscript.

Here is the updated Figure 2, including added equations and references to specific equation numbers of the manuscript:

[Figure]

4. In Section 3.1, the indoor temperature, airflow, and RH for individual animal houses are derived from observations, which likely contribute to the good agreement between modeled and observed NH3 emissions. How are these indoor parameters determined when estimating NH3 emissions at the global scale?

**Reply:** We refer to Sect.S6 in the Supplementary Material, which describes the generalization of housing environments applied to the global simulations.

In AMCLIM–Housing, the indoor temperature and ventilation of animal houses are modelled using a set of empirically derived relationships in relation to the outdoor temperature. These relationships are based on data from the Animal Feeding Operations (AFOs) dataset of the US Environmental Protection Agency (EPA, 2012) and theoretical parameterizations of indoor environments by Gyldenkærne et al. (2005). These relationships can vary between livestock sectors and production systems as each production system of livestock has a corresponding housing system and house type in the global simulations. More details and equations that present the relationships

between indoor temperature, ventilation and outdoor temperature for different housing systems are given in Sect.S6.1. The RH of indoor environments is assumed to be equivalent to outdoor RH.

To help better guide readers to these points in the revised manuscript, we have added a summary of the above points to expand the existing mention of Section S6 in Section 2.5.2 from L487

"To estimate the environmental conditions in livestock houses, we applied empirical relationships between indoor temperature and ventilation of animal houses based on data from the Animal Feeding Operations (EPA, 2012) and theoretical parameterizations of indoor environments by Gyldenkærne et al. (2005). Equations that present the relationships between indoor temperature, ventilation and outdoor temperature for different housing systems are given in Sect.S6.1. The RH of indoor environments is assumed to be equivalent to outdoor RH."

5. The global livestock population data is based on FAOSTAT for 2010. How is this dataset extended to other years? Are the calculations for 2010 and 2018 based on the same livestock population and distribution, or do they incorporate changes over time?

**Reply:** Livestock inputs for 2010 and 2018 are different. Livestock and MMS data for the year 2010 were obtained from the FAO GLEAM model. We applied the interannual variability between 2005 to 2015 reported by Lu and Tian (2017) and extended to year 2018 by a linear interpolation. The meteorological inputs were from ERA5 reanalysis datasets for both years. We made the following changes for the revised manuscript, from from L455

"In the present paper, the combined AMCLIM model was applied for 2010 and 2018 to demonstrate full simulations for two different years, with activity data and meteorological variables varied between years, so that the inter-annual variability in both emissions and volatilization rates can be analysed."

and from L466

"…The reference year of these data is 2010. For simulations for the year 2018, livestock population and N excretion rates were extended by a linear interpolation based on the inter-annual variations between 2005 and 2015 suggested by Lu and Tian (2017). The MMS data that determines the fraction of a manure management system are assumed to be constant through the year."

6. Line 574: Do the volatilization rates refer to the ratio of NH3 emissions to total nitrogen excreted by livestock (Nr)? Please clarify.

**Reply:** We added the following text in the revised manuscript to improve clarity from L655

"For housing, the volatilization rates $(P_V)$ are expressed as percentage of total N by excreted by livestock in the animal houses. For manure management and application to land, the volatilization rates are expressed as percentage of the total remaining N from the previous stage that is volatilized as $NH_3$."

7. In the comparison of $NH_3$ emissions between 2010 and 2018, could the authors directly show the spatial distribution of the difference? Current spatial maps make it difficult to distinguish the changes clearly.

**Reply:** We followed suggestions from both reviewers to make a new figure (included as Figure 15 in the revised manuscript) showing the $NH_3$ emissions from each livestock group by geographical regions for both simulated years, along with comparisons of the volatilization rates. We added the following text from L906

"As shown in Fig. 15, $NH_3$ emissions and volatilization rates vary across different geographical regions and between the two simulated years, i.e., 2010 and 2018. The highest $NH_3$ emissions from livestock agriculture are estimated to occur in East and South Asia. In general, the volatilization rates for livestock are lower in 2018 than 2010, except for poultry. This is because a large fraction of poultry which are broiler and layer production systems are assumed in the model to be kept in houses with controlled temperature and ventilation, so the $P_V$ rates were less impacted by the differences in environmental conditions between years. By comparison, the $P_V$ rates from sheep and goats show the largest inter-annual variability among all livestock groups because sheep and goats typically graze outside and are housed in naturally ventilated barns. As a result, $NH_3$ volatilization from sheep and goats is more dependent on the environmental conditions than poultry. The differences in housing and grazing management also explain why $P_V$ rates from pigs showed the second smallest difference and cattle showed the second largest difference between the two simulated years."

We also generated six additional global maps to show the differences of $NH_3$ emissions and volatilization rates for the two simulated years, expressing as ratios between years. These maps are included as Figures A12 to A17, and we updated Sect.4.4. Please see the revised manuscript for changes.

[Figure]

**Figure R2-1. Estimated (a) NH₃ emissions from livestock farming in seven geographical regions for the two studied years, 2010 (left) and 2018 (right, hashed), and comparisons of volatilization rates $P_V$ between the two years for (b) cattle, (c) pigs, (d) poultry and (e) sheep and goats.**

8. Line 575: It would be helpful if the authors could explain the spatial variations in Pv.

**Reply:** Explanation of the major spatial differences in $P_v$ is given in Section 4.3. We added the following text from L1023

"Among cattle and buffaloes, the overall simulated volatilization rates for buffaloes are higher than other types of cattle. This is because buffaloes are predominantly reared in hot regions such as southern China, South Asia and southeast Asia compared with other cattle, which are widely distributed across the globe, resulting in higher $P_v$ for buffaloes due to generally hotter conditions. Also, the estimated volatilization rates for sheep and goat farming are higher than those of cattle farming, which is partly due to a higher N concentration in sheep and goat's urine compared with cattle. Another reason is that sheep and goat are more "concentrated" in the Middle East and South Asia where they tend to have higher volatilization rates due to warmer climates. In addition to temperature, soil pH plays an important role in $NH_3$ volatilization. As pointed out by Jiang et al. (2024), simulated high $Pv$ values have been found in regions with high soil pH, such as the western US, Namibia, Mongolia and part of northern China.

Various management practices can lead to very different volatilization rates. For housing, industrial pigs show higher volatilization compared to intermediate and backyard pigs because the industrial pigs are kept in buildings with heating systems and excreta are kept longer in the houses as in-situ storage is available. Moreover, the pits for manure storage provides an additional emitting surface of $NH_3$. The housing density assumed in AMCLIM is another factor that affects the volatilization rates. The volatilization rates of feedlot cattle housing are the second lowest among ruminants. This is partly because the feedlot cattle had the highest stocking density in the model. Increasing the stocking density results in a smaller source area for $NH_3$ emission, which leads to lower emissions.

For manure management, especially in warm climates, manure left on land without much management is identified to result in much higher $NH_3$ emissions than manure that is stored either as liquid or solid manure, leading to larger $P_v$ values. Such practice is common in Africa and some countries in South Asia like India and Myanmar, and these regions have hot climate (as reflected in high $P_v$ values). Conversely, manure storage under cover greatly reduces $NH_3$ emissions (Bittman et al., 2014). Although the effect of covering stored manure has not been the focus of the present study, the process-based nature of AMCLIM would lend itself to a future examination of such effects."